# SUBSPACE OPTIMIZATION FOR LARGE LANGUAGE MODELS WITH CONVERGENCE GUARANTEES

## ABSTRACT

Subspace optimization algorithms, with GaLore (Zhao et al., 2024) as a representative method, have gained popularity for pre-training or fine-tuning large language models (LLMs) due to their memory efficiency. However, their convergence guarantees remain unclear, particularly in stochastic settings. In this paper, we unexpectedly discover that GaLore does not always converge to the optimal solution and substantiate this finding with an explicit counter-example. We then investigate the conditions under which GaLore can achieve convergence, demonstrating that it does so either in deterministic scenarios or when using a sufficiently large mini-batch size. More significantly, we introduce **GoLore** (**G**radient rand**o**m **Lo**w-**r**ank proj**e**ction), a novel variant of GaLore that provably converges in stochastic settings, even with standard batch sizes. Our convergence analysis can be readily extended to other sparse subspace optimization algorithms. Finally, we conduct numerical experiments to validate our theoretical results and empirically explore the proposed mechanisms.

## 1 INTRODUCTION

Large Language Models (LLMs) have demonstrated impressive performance across a variety of tasks, including language processing, planning, and coding. However, LLMs require substantial computational resources and memory due to their large model size and the extensive amounts of training data. Consequently, recent advancements in stochastic optimization have focused on developing memory-efficient strategies to pre-train or fine-tune LLMs with significantly reduced computing resources. Most approaches (Vyas et al., 2024; Ramesh et al., 2024; Luo et al., 2023; Liu et al., 2024; Bini et al., 2024; Hao et al., 2024; Zhao et al., 2024; Muhamed et al., 2024; Pan et al., 2024; Loeschcke et al., 2024; Hayou et al., 2024; Lialin et al., 2023; Han et al., 2024; Song et al., 2023) concentrate on reducing the memory of optimizer states, which are critical components of overall training memory consumption. For instance, optimizers such as Adam (Kingma, 2014) and AdamW (Loshchilov, 2017) maintain first and second-order momentum terms for gradients as optimizer states, leading to significant memory overhead for large models.

Among the most popular memory-efficient fine-tuning algorithms is LoRA (Hu et al., 2021), which decreases the number of trainable parameters by employing low-rank model adapters. However, the low-rank constraint on weight updates can result in substantial performance degradation for tasks that require full-rank updates, particularly in the pre-training of LLMs. To address this issue, several LoRA variants have been proposed, including ReLoRA (Lialin et al., 2023) and SLTrain (Han et al., 2024). Recently, GaLore (Zhao et al., 2024) has emerged as an effective solution, significantly reducing optimizer states by projecting full-parameter gradients into periodically recomputed subspaces. By retaining optimizer states in low-rank subspaces, GaLore can reduce memory usage by over 60%, enabling the pre-training of a 7B model on an NVIDIA RTX 4090 with 24GB of memory. In contrast, the vanilla 8-bit Adam without low-rank projection requires over 40GB of memory.

### 1.1 FUNDAMENTAL OPEN QUESTIONS AND MAIN RESULTS

While GaLore's memory efficiency has been well established both theoretically and empirically, its convergence guarantees remain unclear. This raises the following fundamental open question:

*Q1. Can GaLore converge to stationary solutions, under regular assumptions?*

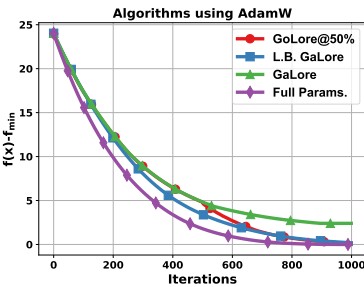 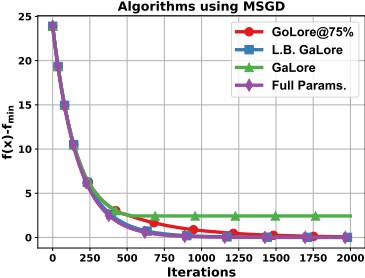

Figure 1: Loss curves of algorithms using AdamW (left) and Momentum SGD (right) on problem (1), where *L.B. GaLore* stands for large-batch GaLore, *GoLore@x%* applies GaLore for the beginning $(100 - x)\%$ iterations and GoLore for the last $x\%$ iterations.

By *stationary solutions*, we refer to first-order stationary points $x \in \mathbb{R}^d$ such that $\nabla f(x) = 0$ for objective function $f : \mathbb{R}^d \to \mathbb{R}$. By *regular assumptions*, we refer to standard conditions in non-convex smooth optimization, including lower boundedness, $L$-smoothness and unbiased stochastic gradients with bounded variances, as outlined in Assumptions 1-3 in Sec. 2.

Contrary to expectations, our investigation reveals that GaLore does **NOT** converge to stationary solutions under regular assumptions. The intuition behind this finding is straightforward: GaLore projects the stochastic gradient matrix onto a low-rank subspace spanned by the top $r$ singular vectors obtained via Singular Value Decomposition (SVD), effectively capturing the dominant components of the stochastic gradient matrix. However, the stochastic gradient comprises two components: the true gradient and gradient noise. When the true gradient dominates, the SVD-identified subspace primarily captures the gradient component. *In contrast, as the algorithm approaches a local minimum so that the true gradient diminishes while noise persists, the SVD-derived subspace captures only the noise component, rather than the true gradient, ultimately leading to non-convergence.* To validate this intuition, we construct a counter-example demonstrating that GaLore fails to converge to stationary solutions, see the illustration in Fig. 1. This leads us to a subsequent open question:

*Q2. Under what additional assumptions can GaLore converge to stationary solutions?*

Based on the preceding discussion, we conclude that the SVD-identified subspace in GaLore aligns well with the descent direction in scenarios where the true gradient component dominates the gradient noise component. This observation naturally leads to two additional assumptions under which GaLore can converge:

- **Noise-Free Assumption.** We theoretically establish that GaLore converges at a rate of $\mathcal{O}(1/T)$ in the deterministic and non-convex setting.

- **Large-Batch Assumption.** We theoretically demonstrate that GaLore converges at a rate of $\mathcal{O}(1/\sqrt{T})$ in the stochastic and non-convex setting, provided that the batch size is extremely large and increases with the number of iterations $T$, *e.g.*, a batch size of $\Theta(\sqrt{T})$.

However, neither the noise-free assumption nor the large-batch assumption applies to the practical pre-training and fine-tuning of LLMs. This leads to another fundamental open question:

*Q3. Under what modifications can GaLore provably converge in the LLM setting, in which gradient noise presents and the batch-size cannot be extremely large?*

It is evident that SVD-based projections cannot extract meaningful information from noise-dominant matrices. To address this issue, this paper proposes modifying the SVD projection to a **G**radient **R**ando**m** **Lo**w-**R**ank proje**c**tion, resulting in the **GoLore** algorithm for pre-training or fine-tuning LLMs. This random projection can effectively capture gradient information even when gradient noise predominates, allowing for convergence in the stochastic and non-convex setting with normal batch sizes. We establish that GoLore converges at a rate of $\mathcal{O}(1/\sqrt{T})$ under standard assumptions.

In our empirical experiments, we implement GaLore during the primary phases of pre-training or fine-tuning LLMs due to its efficacy in capturing the gradient component using SVD-based projection. In contrast, we employ GoLore in the final phase, leveraging its ability to extract the gradient

component from noise-dominant stochastic gradients using random projection. This approach enhances performance compared to employing GaLore throughout all stages.

While our analysis primarily focuses on GaLore, it also has significant connections to other memory-efficient algorithms. We demonstrate that a ReLoRA-like implementation is equivalent to GaLore, which is more computational efficient with little additional memory overhead. Furthermore, our theoretical results can be easily adapted to sparse subspace descent algorithms with minimal effort.

**Contributions.** Our contributions can be summarized as follows:

- We find that GaLore cannot converge to stationary solutions under regular assumptions. The key insight is that the SVD-derived subspace primarily captures the noise component rather than the true gradient in scenarios where gradient noise predominates. We validate the non-convergence of GaLore by providing an explicit counterexample. This addresses Question Q1.
- Inspired by the aforementioned insight, we propose different additional assumptions under which GaLore can provably converge to stationary solutions. Under the noise-free assumption, we establish that GaLore converges at a rate of $\mathcal{O}(1/T)$. Under the large-batch assumption or some additional isotropic noise assumptions, we demonstrate that GaLore converges at a rate of $\mathcal{O}(1/\sqrt{T})$. This addresses Question Q2.
- In settings where gradient noise persists and the batch size cannot be extremely large, we modify the SVD projection in GaLore to a random projection, resulting in the GoLore algorithm that provably converges to stationary solutions at a rate of $\mathcal{O}(1/\sqrt{T})$. This addresses Question Q3.
- We present an equivalent yet more computationally efficient, ReLoRA-like implementation of GaLore/GoLore, and extend our analysis to other sparse subspace descent algorithms.
- We conduct experiments across various tasks to validate our theoretical findings. In particular, by alternately using GaLore and GoLore during different phases in LLMs pre-training and fine-tuning, we achieve enhanced empirical performance.

## 1.2 RELATED WORK

**Memory-efficient training.** In LLM training, the primary memory consumption arises not only from the model parameters but also from activation values and optimizer states. Jiang et al. (2022) and Yu et al. (2024) have proposed methods to compress activation values into sparse vectors to alleviate memory usage. Other approaches primarily focus on reducing optimizer states. A notable work, LoRA (Hu et al., 2021) reparameterizes the weight matrix $W \in \mathbb{R}^{m \times n}$ as $W = W_0 + BA$, where $W_0 \in \mathbb{R}^{m \times n}$ remains frozen as the pre-trained weights, and $B \in \mathbb{R}^{m \times r}$ and $A \in \mathbb{R}^{r \times n}$ are learnable low-rank adapters. Variants of LoRA, such as those proposed by Liu et al. (2024) and Hayou et al. (2024), aim to enhance training performance. However, constrained to low-rank updates, LoRA and its variants are primarily effective for fine-tuning tasks and struggle with pre-training tasks that require high-rank updates. To address this limitation, ReLoRA (Lialin et al., 2023) enables high-rank updates by accumulating multiple LoRA updates, while LISA (Pan et al., 2024) learns full-parameter updates on dynamically selected trainable layers. GaLore (Zhao et al., 2024) and FLORA (Hao et al., 2024) achieve high-rank updates by accumulating low-rank updates in periodically recomputed subspaces, and SLTrain (Han et al., 2024) employs additional sparse adapters for high-rank updates. SIFT (Song et al., 2023) also utilizes sparse updates. Although these algorithms have demonstrated comparable empirical performance to full-parameter training methods, theoretical guarantees regarding their convergence have not been established. A recent study by Liang et al. (2024) provides a proof of continuous-time convergence for a class of online subspace descent algorithms, however, its analysis depends on the availability of true gradients rather than the stochastic gradients that are more practical in LLM training. To the best of our knowledge, this work offers the *first* analysis of the discrete-time convergence rate for memory-efficient LLM training algorithms in stochastic settings.

**Convergence for lossy algorithms.** Many optimization algorithms utilize lossy compression on training dynamics, such as gradients, particularly in the realm of distributed optimization with communication compression. Researchers have established convergence properties for these algorithms based on either unbiased (Li et al., 2020; Li & Richtárik, 2021; Condat et al., 2024; He et al., 2024b;a; Mishchenko et al., 2019; Gorbunov et al., 2021; Alistarh et al., 2017; He et al., 2023) or contractive (Richtárik et al., 2021; Xie et al., 2020; Fatkhullin et al., 2024; He et al., 2023) compressibility. Kozak et al. (2019) provides a convergence analysis for subspace compression under

Polyak-Lojasiewicz or convex conditions, where the subspace compression adheres contractive compressibility at each iteration. Despite these extensive findings, analyzing the convergence properties of subspace descent algorithms like GaLore remains challenging, as the compressions used can be neither unbiased nor contractive due to the reuse of projection matrices.

## 2 PRELIMINARIES AND ASSUMPTIONS

**Full-parameter training.** Training an $N_L$-layer neural network can be formulated as the following optimization problem:

$$\min_{\boldsymbol{x}} f(\boldsymbol{x}) := \mathbb{E}_{\xi \sim \mathcal{D}} F(\boldsymbol{x}; \xi).$$

Here, $\boldsymbol{x} = (\text{vec}(\boldsymbol{X}_1)^\top, \cdots, \text{vec}(\boldsymbol{X}_{N_L})^\top)^\top$ collects all trainable parameters in the model, where $N_L$ is the number of layers, $\boldsymbol{X}_\ell \in \mathbb{R}^{m_\ell \times n_\ell}$ denotes the weight matrix in the $\ell$-th layer, $\ell = 1, \cdots, N_L$. $F(\boldsymbol{x}; \xi)$ computes the loss with respective to data point $\xi$, $\mathcal{D}$ denotes the training data distribution. In full-parameter training, we directly apply the optimizer to the full-parameter $\boldsymbol{x}$:

$$\boldsymbol{G}_\ell^{(t)} = \nabla_\ell F(\boldsymbol{x}^{(t)}; \xi^{(t)}), \quad \boldsymbol{X}_\ell^{(t+1)} = \boldsymbol{X}_\ell^{(t)} + \rho_\ell^{(t)}(\boldsymbol{G}_\ell^{(t)}), \quad \ell = 1, \cdots, N_L;$$

where $\nabla_\ell$ computes the gradient with respective to the $\ell$-th weight matrix $\boldsymbol{X}_\ell$, superscript $(t)$ denotes the variable in the $t$-th iteration, and $\rho_\ell^{(t)}$ is an entry-wise stateful gradient operator, such as Adam or Momentum SGD (MSGD). Specifically, using MSGD leads to the following $\rho_\ell^{(t)}(\cdot)$:

$$\boldsymbol{M}_\ell^{(t)} = (1 - \beta_1)\boldsymbol{M}_\ell^{(t-1)} + \beta_1 \boldsymbol{G}_\ell^{(t)}; \quad \rho_\ell^{(t)}(\boldsymbol{G}_\ell^{(t)}) = -\eta \boldsymbol{M}_\ell^{(t)};$$

where $\eta$ is the learning rate, $\beta_1 \in (0, 1]$ is the momentum coefficient, and $\boldsymbol{M}_\ell^{(t)}$ is the momentum retained in the optimizer state. In full-parameter pre-training or fine-tuning of LLMs, the memory requirements for storing momentum in MSGD and the additional variance state in Adam are highly demanding. According to Zhao et al. (2024), pre-training a LLaMA 7B model with a single batch size requires 58 GB of memory, with 42 GB allocated to Adam optimizer states and weight gradients.

**GaLore algorithm.** To address the memory challenge, Zhao et al. (2024) proposes a Gradient Low-Rank Projection (GaLore) approach that allows full-parameter learning but is much more memory-efficient. The key idea is to project each stochastic gradient $\boldsymbol{G}_\ell \in \mathbb{R}^{m_\ell \times n_\ell}$ onto a low-rank subspace, yielding a low-dimensional gradient approximation. Specifically, GaLore performs SVD on $\boldsymbol{G}_\ell^{(t)} = \boldsymbol{U}\boldsymbol{\Sigma}\boldsymbol{V}^\top$ and obtains rank-$r_\ell$ projection matrices $\boldsymbol{P}_\ell^{(t)} = \boldsymbol{U}[:, : r_\ell] \in \mathbb{R}^{m_\ell \times r_\ell}$ and $\boldsymbol{Q}_\ell^{(t)} = \boldsymbol{V}[:, : r_\ell] \in \mathbb{R}^{n_\ell \times r_\ell}$, where $[:, : r]$ denotes the selection of the matrix's first $r$ columns. When $m_\ell \leq n_\ell$, GaLore projects $\boldsymbol{G}_\ell$ onto $\boldsymbol{P}_\ell$, yielding a low-rank gradient representation $(\boldsymbol{P}_\ell^{(t)})^\top \boldsymbol{G}_\ell^{(t)} \in \mathbb{R}^{r_\ell \times n_\ell}$. Conversely, when $m_\ell > n_\ell$, GaLore projects $\boldsymbol{G}_\ell$ onto $\boldsymbol{Q}_\ell$, resulting in $\boldsymbol{G}_\ell^{(t)}\boldsymbol{Q}_\ell^{(t)} \in \mathbb{R}^{m_\ell \times r_\ell}$. In either scenarios, the memory cost of optimizer states associated with these low-rank representations can be significantly reduced, leading to memory-efficient LLMs pre-training or fine-tuning:

$$\boldsymbol{X}_\ell^{(t+1)} = \begin{cases} \boldsymbol{X}_\ell^{(t)} + \boldsymbol{P}_\ell^{(t)} \rho_\ell^{(t)}((\boldsymbol{P}_\ell^{(t)})^\top \boldsymbol{G}_\ell^{(t)}), & \text{if } m_\ell \leq n_\ell; \\ \boldsymbol{X}_\ell^{(t)} + \rho_\ell^{(t)}(\boldsymbol{G}_\ell^{(t)}\boldsymbol{Q}_\ell^{(t)})(\boldsymbol{Q}_\ell^{(t)})^\top, & \text{if } m_\ell > n_\ell. \end{cases}$$

Typically, GaLore selects $\rho_\ell(\cdot)$ as the Adam gradient operator, as illustrated in Alg. 1. However, GaLore can also choose $\rho_\ell(\cdot)$ to be gradient operators in either vanilla SGD or MSGD. Since SVD decomposition is computationally expensive, GaLore updates $\boldsymbol{P}_\ell^{(t)}$ or $\boldsymbol{Q}_\ell^{(t)}$ periodically. In other words, GaLore computes $\boldsymbol{P}_\ell^{(t)}$ or $\boldsymbol{Q}_\ell^{(t)}$ when iteration step $t \not\equiv 0 \pmod{\tau}$ where $\tau > 0$ is the period, otherwise $\boldsymbol{P}_\ell^{(t)} = \boldsymbol{P}_\ell^{(t-1)}$ and $\boldsymbol{Q}_\ell^{(t)} = \boldsymbol{Q}_\ell^{(t-1)}$ remain unchanged. Both the gradient subspace projection and periodic switches between different low-rank subspaces pose significant challenges to the convergence analysis for GaLore-like algorithms.

**Stiefel manifold.** An $m \times r$ Stiefel manifold ($r \leq m$) is defined as

$$\text{St}_{m,r} = \{\boldsymbol{P} \in \mathbb{R}^{m \times r} \mid \boldsymbol{P}^\top \boldsymbol{P} = I_r\}.$$

Stiefel manifold is the set of low-rank projection matrices to use in subspace optimization. Typically, in GaLore we have $\boldsymbol{P}_\ell^{(t)} \in \text{St}_{m_\ell, r_\ell}$ and $\boldsymbol{Q}_\ell^{(t)} \in \text{St}_{n_\ell, r_\ell}$.

**Basic assumptions.** We introduce the basic assumptions used throughout our theoretical analysis. Each of these assumptions is standard for stochastic optimization.

**Assumption 1** (Lower boundedness). *The objective function $f : \mathbb{R}^d \to \mathbb{R}$ satisfies $\inf_{\boldsymbol{x} \in \mathbb{R}^d} f(\boldsymbol{x}) > -\infty$, where $d = \sum_{\ell=1}^{N_\ell} m_\ell n_\ell$ is the total number of parameters in the model.*

**Assumption 2** (L-smoothness). *The objective function $f : \mathbb{R}^d \to \mathbb{R}$ satisfies $\|\nabla f(\boldsymbol{x}) - \nabla f(\boldsymbol{y})\|_2 \leq L\|\boldsymbol{x} - \boldsymbol{y}\|_2$, for any $\boldsymbol{x}, \boldsymbol{y} \in \mathbb{R}^d$.*

**Assumption 3** (Stochastic gradient). *The gradient oracle $(F, \mathcal{D})$ satisfies*

$$\mathbb{E}_{\xi \sim \mathcal{D}}[\nabla_\ell F(\boldsymbol{x}; \xi)] = \nabla_\ell f(\boldsymbol{x}), \quad and \quad \mathbb{E}_{\xi \sim \mathcal{D}}[\|\nabla_\ell F(\boldsymbol{x}; \xi) - \nabla_\ell f(\boldsymbol{x})\|_F^2] \leq \sigma_\ell^2, \quad \forall \boldsymbol{x} \in \mathbb{R}^d,$$

*where $\sigma_\ell > 0$ is a scalar. Summing all weight matrices we obtain*

$$\mathbb{E}_{\xi \sim \mathcal{D}}[\nabla F(\boldsymbol{x}; \xi)] = \nabla f(\boldsymbol{x}), \quad and \quad \mathbb{E}_{\xi \sim \mathcal{D}}[\|\nabla F(\boldsymbol{x}; \xi) - \nabla f(\boldsymbol{x})\|_2^2] \leq \sigma^2, \quad \forall \boldsymbol{x} \in \mathbb{R}^d,$$

*where $\sigma = \sqrt{\sum_{\ell=1}^{N_\ell} \sigma_\ell^2}$.*

## 3 NON-CONVERGENCE OF GALORE: INTUITION AND COUNTER-EXAMPLE

In this section, we demonstrate why GaLore cannot guarantee exact convergence under Assumptions 1-3. We first illustrate the insight behind the result, then present its formal description.

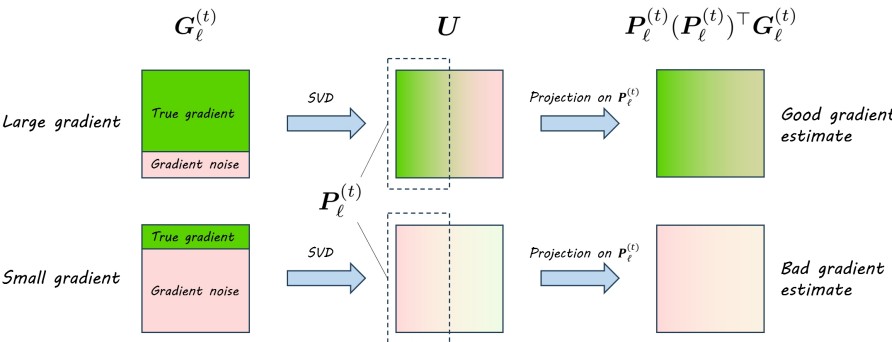

Figure 2: An illustration of the insight on why GaLore fails to converge in small-gradient scenarios. We use color green for true gradient and red for gradient noise.

**Insight behind non-convergence.** As reviewed in Sec. 2, GaLore performs SVD on stochastic gradient $\boldsymbol{G} = \boldsymbol{U}\boldsymbol{\Sigma}\boldsymbol{V}^\top$ and obtains rank-$r$ projection matrices $\boldsymbol{P} = \boldsymbol{U}[:, : r] \in \mathbb{R}^{m \times r}$. GaLore projects $\boldsymbol{G}$ onto $\boldsymbol{P}$, yielding a low-rank gradient representation $\boldsymbol{P}^\top \boldsymbol{G} \in \mathbb{R}^{r \times n}$. In other words, GaLore projects the stochastic gradient matrix onto a low-rank subspace spanned by the top $r$ singular vectors, capturing the dominant components of the stochastic gradient matrix. However, the stochastic gradient comprises two components: the true gradient and gradient noise, as shown in Fig. 2. When the true gradient significantly exceeds the gradient noise, typically at the start of training, the low-rank subspace obtained via SVD effectively preserves the true gradient information. As training progresses and the true gradient diminishes to zero, especially near a local minimum, the subspace may become increasingly influenced by gradient noise. In the extreme case, this noise-dominated subspace can become orthogonal to the true gradient subspace, leading to non-convergence.

**Counter-Example.** We consider the following quadratic problem with gradient noise:

$$f(\boldsymbol{X}) = \frac{1}{2}\|\boldsymbol{A}\boldsymbol{X}\|_F^2 + \langle \boldsymbol{B}, \boldsymbol{X} \rangle_F, \quad \nabla F(\boldsymbol{X}; \xi) = \nabla f(\boldsymbol{X}) + \xi\sigma\boldsymbol{C}, \tag{1}$$

where $\boldsymbol{A} = (\boldsymbol{I}_{n-r} \quad 0) \in \mathbb{R}^{(n-r) \times n}$, $\boldsymbol{B} = \begin{pmatrix} \boldsymbol{D} & 0 \\ 0 & 0 \end{pmatrix} \in \mathbb{R}^{n \times n}$ with $\boldsymbol{D} \in \mathbb{R}^{(n-r) \times (n-r)}$ generated randomly, $\boldsymbol{C} = \begin{pmatrix} 0 & 0 \\ 0 & \boldsymbol{I}_r \end{pmatrix} \in \mathbb{R}^{n \times n}$, $\xi$ is a random variable uniformly sampled from $\{1, -1\}$ per iteration, and $\sigma$ is used to control the gradient noise. It is straightforward to verify that problem (1) satisfies Assumptions 1-3. Moreover, as $\boldsymbol{X}$ approaches the global minimum of $f(\boldsymbol{X})$, the true gradient $\nabla f(X) \to 0$, while the gradient noise persists with a variance on the order of $\sigma^2$. Fig. 1

illustrates the performance of GaLore when solving problem (1). It is observed that GaLore fails to converge to the optimal solution, regardless of whether the AdamW or MSGD optimizer is used.

**Non-convergence of GaLore.** Based on the aforementioned insight, we establish the following theorem regarding the non-convergence of GaLore.

**Theorem 1** (Non-convergence of GaLore). *There exists an objective function $f : \mathbb{R}^d \to \mathbb{R}$ satisfying Assumptions 1, 2, a stochastic gradient oracle $(F, \mathcal{D})$ satisfying Assumption 3, an initial point $\boldsymbol{x}^{(0)} \in \mathbb{R}^d$, a constant $\epsilon_0 > 0$ such that for any rank $r_\ell < \min\{m_\ell, n_\ell\}$, subspace changing frequency $\tau$, any subspace optimizer $\rho$ inputting subspace gradient of shape $r_\ell \times n_\ell$ and outputting subspace update direction of shape $r_\ell \times n_\ell$ with arbitrary hyperparameters and any $t > 0$, it holds that*

$$\|\nabla f(\boldsymbol{x}^{(t)})\|_2^2 \geq \epsilon_0.$$

## 4 CONDITIONS UNDER WHICH GALORE CAN CONVERGE

**GaLore provably converges in the noise-free setting.** According to the insight presented in Sec. 3, GaLore fails to converge when gradient noise dominates the true gradient in magnitudes. This motivates us to examine the deterministic scenario where the true gradient $\nabla f(\boldsymbol{x})$ can be accessed without any gradient noise. The GaLore algorithm with noise-free gradients is presented in Alg. 1 (or Alg. 2 in Appendix B.3), where the true gradient oracle is highlighted with the label (deterministic).

Since no gradient noise exists, the projection matrix $\boldsymbol{P}_\ell^{(t)}$ obtained by SVD can effectively capture the true gradient even when the algorithm approaches a local minimum. For simplicity, we analyze GaLore with MSGD and the following momentum updating mechanism:

$$\boldsymbol{M}_\ell^{(t)} = \begin{cases} (1 - \beta_1)(\boldsymbol{P}_\ell^{(t)})^\top \boldsymbol{P}_\ell^{(t-1)} \boldsymbol{M}_\ell^{(t-1)} + \beta_1 (\boldsymbol{P}_\ell^{(t)})^\top \boldsymbol{G}_\ell^{(t)}, & \text{if } m_\ell \leq n_\ell, \\ (1 - \beta_1)\boldsymbol{M}_\ell^{(t-1)}(\boldsymbol{Q}_\ell^{(t-1)})^\top \boldsymbol{Q}_\ell^{(t)} + \beta_1 \boldsymbol{G}_\ell^{(t)} \boldsymbol{Q}_\ell^{(t)}, & \text{if } m_\ell > n_\ell. \end{cases} \tag{2}$$

If the subspace does not change at iteration $t$, $(\boldsymbol{P}_\ell^{(t)})^\top \boldsymbol{P}_\ell^{(t-1)} = (\boldsymbol{Q}_\ell^{(t-1)})^\top \boldsymbol{Q}_\ell^{(t)} = \boldsymbol{I}_{r_\ell}$ and (2) reduces to regular momentum updates. If the subspace changes at iteration $t$, we inherit $\boldsymbol{M}_\ell^{(t-1)}$ by first projecting back to the previous space and then to the new subspace. For convenience, we use *momentum projection (MP)* to refer to mechanism (2). When MP is used in the algorithm, we label the corresponding with (with MP) in Alg. 1 otherwise (without MP). The following theorem provides convergence guarantees for GaLore using deterministic gradients and MSGD with MP.

**Theorem 2** (Convergence rate of deterministic GaLore). *Under Assumptions 1-2, if the number of iterations $T \geq 64/(3\underline{\delta})$ and we choose*

$$\beta_1 = 1, \quad \tau = \left\lceil \frac{64}{3\underline{\delta}\beta_1} \right\rceil, \quad \text{and} \quad \eta = \left( 4L + \sqrt{\frac{80L^2}{3\underline{\delta}\beta_1^2}} + \sqrt{\frac{80\tau^2 L^2}{3\underline{\delta}}} + \sqrt{\frac{16\tau L^2}{3\beta_1}} \right)^{-1},$$

*GaLore using deterministic gradients and MSGD with MP converges as*

$$\frac{1}{T}\sum_{t=0}^{T-1} \|\nabla f(\boldsymbol{x}^{(t)})\|_2^2 = \mathcal{O}\left( \frac{L\Delta}{\underline{\delta}^{5/2}T} \right),$$

*where $\Delta = f(\boldsymbol{x}^{(0)}) - \inf_{\boldsymbol{x}} f(\boldsymbol{x})$ and $\underline{\delta} := \min_\ell \frac{r_\ell}{\min\{m_\ell, n_\ell\}}$.*

**Remark.** In fact, *MSGD* here reduces to momentum gradient descent by using deterministic gradients. Theorem 2 demonstrates that GaLore converges at a rate of $\mathcal{O}(1/T)$ in the deterministic scenario, which is on the same order as full-parameter training. A more detailed result is presented in Theorem 6 in Appendix B.3, where we established convergence for more general hyperparameter choices However, in deep learning tasks with exceptionally large training datasets, computing the true gradient becomes impractical due to significant computational and memory costs. Therefore, we will next focus on the stochastic setting.

**GaLore provably converges with large-batch stochastic gradients.** Inspired by the insight presented in Sec. 3, GaLore converges in cases where the true gradient dominates the gradient noise.

**Algorithm 1** GaLore / GoLore algorithm using stochastic / deterministic / large-batch gradients with / without momentum projection

---

**Input:** Initial point $\boldsymbol{x}^{(0)}$, data distribution $\mathcal{D}$, learning rate $\eta$, subspace changing frequency $\tau$, rank $\{r_\ell\}_{\ell=1}^{N_L}$, optimizer hyperparameters $\beta_1, \beta_2, \epsilon$, large batch size $\mathcal{B}$.

**Output:** $\{\boldsymbol{x}^{(t)}\}_{t=0}^T$.

    Initialize optimizer state $\{\boldsymbol{M}_\ell^{(-1)}\}_{\ell=1}^{N_L}$ and $\{\boldsymbol{V}_\ell^{(-1)}\}_{\ell=1}^{N_L}$ to zero;

    **for** $t = 0, 1, \cdots, T-1$ **do**

        **for** $\ell = 1, 2, \cdots, N_L$ **do**

            **if** $t \equiv 0 \pmod{\tau}$ **then**

                $\boldsymbol{G}_\ell^{(t)} \leftarrow \nabla_\ell F(\boldsymbol{x}^{(t)}; \xi^{(t)});$   (stochastic)

                $\boldsymbol{G}_\ell^{(t)} \leftarrow \nabla_\ell f(\boldsymbol{x}^{(t)});$   (deterministic)

                $\boldsymbol{G}_\ell^{(t)} \leftarrow \frac{1}{\mathcal{B}} \sum_{b=1}^{\mathcal{B}} \nabla_\ell F(\boldsymbol{x}^{(t)}; \xi^{(t,b)});$   (large-batch)

                $\boldsymbol{U}, \boldsymbol{\Sigma}, \boldsymbol{V} \leftarrow \text{SVD}(\boldsymbol{G}_\ell^{(t)}), \boldsymbol{P}_\ell^{(t)} \leftarrow \boldsymbol{U}[:, : r_\ell], \boldsymbol{Q}_\ell^{(t)} \leftarrow \boldsymbol{V}[:, : r_\ell];$   (GaLore)

                Sample $\boldsymbol{P}_\ell^{(t)} \sim \mathcal{U}(\text{St}_{m_\ell, r_\ell}),$   $\boldsymbol{Q}_\ell^{(t)} \sim \mathcal{U}(\text{St}_{n_\ell, r_\ell});$   (GoLore)

            **else**

                $\boldsymbol{G}_\ell^{(t)} \leftarrow \nabla_\ell F(\boldsymbol{x}^{(t)}; \xi^{(t)});$   (stochastic)

                $\boldsymbol{G}_\ell^{(t)} \leftarrow \nabla_\ell f(\boldsymbol{x}^{(t)});$   (deterministic)

                $\boldsymbol{G}_\ell^{(t)} \leftarrow \nabla_\ell F(\boldsymbol{x}^{(t)}; \xi^{(t)});$   (large-batch)

                $\boldsymbol{P}_\ell^{(t)} \leftarrow \boldsymbol{P}_\ell^{(t-1)}, \boldsymbol{Q}_\ell^{(t)} \leftarrow \boldsymbol{Q}_\ell^{(t-1)};$

            **end if**

            $\boldsymbol{R}_\ell^{(t)} \leftarrow \begin{cases} (\boldsymbol{P}_\ell^{(t)})^\top \boldsymbol{G}_\ell^{(t)}, & \text{if } m_\ell \leq n_\ell; \\ \boldsymbol{G}_\ell^{(t)} \boldsymbol{Q}_\ell^{(t)}, & \text{if } m_\ell > n_\ell; \end{cases}$

            $\boldsymbol{M}_\ell^{(t)} \leftarrow \begin{cases} (1 - \beta_1)(\boldsymbol{P}_\ell^{(t)})^\top \boldsymbol{P}_\ell^{(t-1)} \boldsymbol{M}_\ell^{(t-1)} + \beta_1 \boldsymbol{R}_\ell^{(t)}, & \text{if } m_\ell \leq n_\ell; \\ (1 - \beta_1) \boldsymbol{M}_\ell^{(t-1)} (\boldsymbol{Q}_\ell^{(t-1)})^\top \boldsymbol{Q}_\ell^{(t)} + \beta_1 \boldsymbol{R}_\ell^{(t)}, & \text{if } m_\ell > n_\ell; \end{cases}$   (with MP)

            $\boldsymbol{M}_\ell^{(t)} \leftarrow (1 - \beta_1) \boldsymbol{M}_\ell^{(t-1)} + \beta_1 \boldsymbol{R}_\ell^{(t)};$   (without MP)

            $\boldsymbol{V}_\ell^{(t)} \leftarrow (1 - \beta_2) \boldsymbol{V}_\ell^{(t-1)} + \beta_2 \boldsymbol{R}_\ell^{(t)} \odot \boldsymbol{R}_\ell^{(t)};$

            **if** using Adam **then**

                $\boldsymbol{M}_\ell^{(t)} \leftarrow \boldsymbol{M}_\ell^{(t)}/(1 - \beta_1^t), \quad \boldsymbol{V}_\ell^{(t)} \leftarrow \boldsymbol{V}_\ell^{(t)}/(1 - \beta_2^t), \quad \boldsymbol{N}_\ell^{(t)} \leftarrow \boldsymbol{M}_\ell^{(t)}/(\sqrt{\boldsymbol{V}_\ell^{(t)}} + \epsilon);$

            **else if** using MSGD **then**

                $\boldsymbol{N}_\ell^{(t)} \leftarrow \boldsymbol{M}_\ell^{(t)};$

            **end if**

            $\boldsymbol{X}_\ell^{(t+1)} \leftarrow \begin{cases} \boldsymbol{X}_\ell^{(t)} - \eta \boldsymbol{P}_\ell^{(t)} \boldsymbol{N}_\ell^{(t)}, & \text{if } m_\ell \leq n_\ell; \\ \boldsymbol{X}_\ell^{(t)} - \eta \boldsymbol{N}_\ell^{(t)} (\boldsymbol{Q}_\ell^{(t)})^\top, & \text{if } m_\ell > n_\ell; \end{cases}$

        **end for**

    **end for**

---

This convergence can be ensured by reducing the gradient noise through an increased batch size, particularly as the algorithm approaches a local minimum. Specifically, we replace the stochastic gradient $\boldsymbol{G}_\ell^{(t)} = \nabla_\ell F(\boldsymbol{x}^{(t)}; \xi^{(t)})$ with large-batch gradient $\boldsymbol{G}_\ell^{(t)} = \frac{1}{\mathcal{B}} \sum_{b=1}^{\mathcal{B}} \nabla_\ell F(\boldsymbol{x}^{(t)}; \xi^{(t,b)})$, which reduces the variance of gradient noise by $\mathcal{B}$ times. The GaLore algorithm with large-batch stochastic gradients is presented in Alg. 1 (or Alg. 3 in Appendix B.4), where the large-batch stochastic gradient oracle is highlighted with the label (large-batch) . It is worth noting that the non-convergence of GaLore primarily stems from the erroneous subspace dominated by gradient noise. Therefore, we compute a large-batch gradient only for the SVD step while maintaining a smaller batch size for

other computations, see Alg. 1. As the batch size $\mathcal{B}$ increases with iteration $T$, GaLore provably converge to stationary solutions, as established in the following theorem:

**Theorem 3** (Convergence rate of large-batch GaLore). *Under Assumptions 1-3, if $T \geq 2 + 256/(3\underline{\delta}) + (256\sigma)^2/(9\sqrt{\underline{\delta}}L\Delta)$ and we choose $\tau = \lceil 128/(3\underline{\delta}\beta_1) \rceil$, $\mathcal{B} = \lceil 1/(\underline{\delta}\beta_1) \rceil$,*

$$\beta_1 = \left(1 + \sqrt{\frac{\underline{\delta}^{3/2}\sigma^2 T}{L\Delta}}\right)^{-1}, \quad and \quad \eta = \left(4L + \sqrt{\frac{80L^2}{3\underline{\delta}\beta_1^2}} + \sqrt{\frac{40\tau^2 L^2}{\underline{\delta}}} + \sqrt{\frac{32\tau L^2}{3\beta_1}}\right)^{-1},$$

*GaLore using large-batch gradients and MSGD with MP converges as*

$$\frac{1}{T}\sum_{t=0}^{T-1} \mathbb{E}[\|\nabla f(\boldsymbol{x}^{(t)})\|_2^2] = \mathcal{O}\left(\frac{L\Delta}{\underline{\delta}^{5/2}T} + \sqrt{\frac{L\Delta\sigma^2}{\underline{\delta}^{7/2}T}}\right),$$

*where $\Delta = f(\boldsymbol{x}^{(0)}) - \inf_{\boldsymbol{x}} f(\boldsymbol{x})$ and $\underline{\delta} := \min_\ell \frac{r_\ell}{\min\{m_\ell, n_\ell\}}$.*

**Remark.** A more detailed result is presented in Theorem 7 in Appendix B.4, where we established convergence for more general hyperparameter choices. The batch size $\mathcal{B} = \Theta(\sqrt{T})$ in large-batch GaLore grows with iteration $T$, leading to increased memory overhead, making it less practical than small-batch GaLore. With gradient accumulation, an additional variable is needed to track the gradient, complicating compatibility with per-layer weight updates. Otherwise, larger batch sizes raise the memory required for activation values. Therefore, exploring algorithms that can converge with standard small-batch stochastic gradients becomes essential.

**Empirical validation.** Fig. 1 illustrates the convergence of large-batch GaLore (blue curve) in solving problem (1). It demonstrates that large-batch GaLore effectively corrects the bias present in small-batch stochastic GaLore (green curve), achieving convergence to stationary solutions.

**GaLore provably converges with isotropic noise assumptions.** In Appendix G, we further prove that under some additional isotropic noise assumptions, GaLore with small-batch stochastic gradients can also be guaranteed to converge at a rate of $\mathcal{O}(1/\sqrt{T})$.

## 5 GOLORE: GRADIENT RANDOM LOW-RANK PROJECTION

**GoLore algorithm.** The main issue with SVD-based projection in GaLore is that it aims to capture the dominant component in the stochastic gradient matrix. Consequently, when gradient noise overshadows the true gradient as the algorithm approaches a local minimum, the SVD-based projection fails to identify valuable gradient information.

To address this, we propose replacing the SVD-based projection with a random projection, which captures components of the stochastic gradient matrix randomly without any preference. This results in the GoLore algorithm presented in Alg. 1 (or Alg. 4 in Appendix B.5). In Alg. 1, the GaLore method highlighted with the label (GaLore) samples the projection matrix $\boldsymbol{P}_\ell^{(t)}$ via SVD decomposition. In contrast, the GoLore method highlighted with the label (GoLore) samples $\boldsymbol{P}_\ell^{(t)}$ from $\mathcal{U}(\mathrm{St}_{m_\ell, r_\ell})$, a uniform distribution on the $m_\ell \times r_\ell$ Stiefel manifold. The following proposition provides a practical strategy to sample from distribution $\mathcal{U}(\mathrm{St}_{m,r})$.

**Proposition 1** (Chikuse (2012), Theorem 2.2.1). *A random matrix $\boldsymbol{X}$ uniformly distributed on $\mathrm{St}_{m,r}$ is expressed as $\boldsymbol{X} = \boldsymbol{Z}(\boldsymbol{Z}^\top \boldsymbol{Z})^{-1/2}$, where the elements of an $m \times r$ random matrix $\boldsymbol{Z}$ are independent and identically distributed as normal $\mathcal{N}(0,1)$.*

**Convergence guarantee.** Unlike SVD used in GaLore, the random sampling strategy in GoLore prevents the subspace from being dominated by gradient noise. The theorem below provides convergence guarantees for GoLore when using small-batch stochastic gradients and MSGD with MP.

**Theorem 4** (Convergence rate of GoLore). *Under Assumptions 1-3, for any $T \geq 2 + 128/(3\underline{\delta}) + (128\sigma)^2/(9\sqrt{\underline{\delta}}L\Delta)$, if we choose $\tau = \lceil 64/(3\underline{\delta}\beta_1) \rceil$,*

$$\beta_1 = \left(1 + \sqrt{\frac{\underline{\delta}^{3/2}\sigma^2 T}{L\Delta}}\right)^{-1}, \quad and \quad \eta = \left(4L + \sqrt{\frac{80L^2}{3\underline{\delta}\beta_1^2}} + \sqrt{\frac{80\tau^2 L^2}{3\underline{\delta}}} + \sqrt{\frac{16\tau L^2}{3\beta_1}}\right)^{-1},$$

Table 1: Memory and computation comparison between GaLore's original implementation and our ReLoRA-like version, both utilizing MSGD with batch size $b$. We assume the weight $\boldsymbol{W} \in \mathbb{R}^{m \times n}$ satisfies $m \leq n$.

| GaLore Implementation | Memory | Computation |
|---|---|---|
| (Zhao et al., 2024) | $mn + rm + rn + bm$ | $6bmn + 4rmn + 2mn + 3rn$ |
| Our ReLoRA-like version | $mn + rm + 2rn + bm + br$ | $4bmn + 4brm + 6brn + 5rn$ |

*GoLore using small-batch stochastic gradients and MSGD with MP converges as*

$$\frac{1}{T} \sum_{t=0}^{T-1} \mathbb{E}[\|\nabla f(\boldsymbol{x}^{(t)})\|_2^2] = \mathcal{O}\left(\frac{L\Delta}{\underline{\delta}^{5/2}T} + \sqrt{\frac{L\Delta\sigma^2}{\underline{\delta}^{7/2}T}}\right),$$

*where $\Delta = f(\boldsymbol{x}^{(0)}) - \inf_{\boldsymbol{x}} f(\boldsymbol{x})$ and $\underline{\delta} := \min_\ell \frac{r_\ell}{\min\{m_\ell, n_\ell\}}$.*

**Remark.** Theorem 4 demonstrates that GaLore converges at a rate of $\mathcal{O}(1/\sqrt{T})$, which is consistent with the convergence rate of full-parameter pre-training using standard MSGD. A more detailed result is presented in Theorem 8 in Appendix B.5, where we established convergence for more general hyperparameter choices. Unlike deterministic GaLore and low-rank GaLore discussed in Sec. 4, the newly-proposed GoLore algorithm converges in the non-convex stochastic setting with regular batch sizes, making it far more suitable for LLM pre-training and fine-tuning.

**Practical application of GoLore in LLMs.** While GoLore have theoretical convergence guarantees, directly applying GoLore in LLM tasks may not be ideal. The advantage of using randomly sampled projection matrices becomes evident in the later stages of training, where stochastic gradients are primarily dominated by gradient noise. However, in the early stages, projection matrices derived from SVD retain more gradient information, leading to more effective subspaces. Therefore, we recommend a *hybrid* approach: initially using GaLore to converge toward the neighborhood of the solution, then switching to GoLore for refinement and achieving more accurate results.

**Empirical validation.** Fig. 1 shows the convergence of the hybrid algorithm (red curve) applied to problem (1), which employs GaLore during the early training phase and switches to GoLore in the later stage. It is observed that the hybrid algorithm successfully converges to stationary solutions.

## 6 CONNECTION WITH OTHER SUBSPACE OPTIMIZATION METHODS

**Connection with ReLoRA.** Algorithms like GaLore/GoLore that optimizes in periodically recomputed subspaces can be implemented in an equivalent yet potentially more computational efficient, ReloRA-like way. Consider a linear layer $\boldsymbol{y} = \boldsymbol{W}\boldsymbol{x}$ with $\boldsymbol{W} \in \mathbb{R}^{m \times n}$, where $m \leq n$, GaLore first computes the full-parameter gradient $\nabla_{\boldsymbol{W}}\mathcal{L} = (\nabla_{\boldsymbol{y}}\mathcal{L})\boldsymbol{x}^\top$ via back propagation and update $\boldsymbol{W}$ in the subspace as $\boldsymbol{W} \leftarrow \boldsymbol{W} + \boldsymbol{P}\rho(\boldsymbol{P}^\top(\nabla_{\boldsymbol{W}}\mathcal{L}))$, where $\boldsymbol{P} \in \mathbb{R}^{m \times r}$ is a low-rank projection matrix. If we use LoRA adaptation $\boldsymbol{W} = \boldsymbol{W}_0 + \boldsymbol{B}\boldsymbol{A}$ with $\boldsymbol{B} \in \mathbb{R}^{m \times r}$ and $\boldsymbol{A} \in \mathbb{R}^{r \times n}$, we compute $\boldsymbol{A}$'s gradient $\nabla_{\boldsymbol{A}}\mathcal{L} = (\nabla_{\boldsymbol{z}}\mathcal{L})\boldsymbol{x}^\top = \boldsymbol{B}^\top(\nabla_{\boldsymbol{y}}\mathcal{L})\boldsymbol{x}^\top$, where $\boldsymbol{z} = \boldsymbol{B}\boldsymbol{x}$ is the additional activation. If we fix $\boldsymbol{B} = \boldsymbol{P}$, update $\boldsymbol{A} \leftarrow \boldsymbol{A} + \rho(\nabla_{\boldsymbol{A}}\mathcal{L})$ is equivalent to $\boldsymbol{W} \leftarrow \boldsymbol{W} + \boldsymbol{P}\rho(\boldsymbol{P}^\top(\nabla_{\boldsymbol{W}}\mathcal{L}))$. The memory and computational costs of the two implementations are compared in Table 1, showing the potential of our ReLoRA-like implementation to reduce computation with little memory overhead. Detailed algorithm descriptions and calculations are in Appendix D.

**Connection with FLORA.** Aware of the equivalence of the two (GaLore/ReLoRA-like) implementations, the main difference between GoLore and FLORA lies in the choice of projection matrices. Though both algorithms sample $\boldsymbol{P} \in \mathbb{R}^{m \times r}$ randomly, GoLore uses a uniform distribution on the Stiefel manifold $\mathcal{U}(\mathrm{St}_{m,r})$, while FLORA uses a random Gaussian distribution where each element in $\boldsymbol{P}$ is independently sampled from $\mathcal{N}(0, 1/r)$, and thus $\boldsymbol{P}$ may not belongs to $\mathrm{St}_{m,r}$.

**Connection with SIFT.** SIFT fine-tunes LLMs with sparsified gradients, which can also be viewed as subspace descent. While GaLore projects gradient $\boldsymbol{G}$ to $\boldsymbol{P}^\top\boldsymbol{G}$ via a projection matrix $\boldsymbol{P}$, SIFT projects gradient $\boldsymbol{G}$ to $\boldsymbol{S} \odot \boldsymbol{G}$ via a sparse mask matrix $\boldsymbol{S}$. Our theoretical analysis can be directly transferred to sparse subspace descent with little effort, implying similar results as in low-rank subspace descent, see Appendix C.

## 7 EXPERIMENTS

We evaluate GaLore and GoLore on several different tasks, including solving a counter-example problem (1), pre-training and fine-tuning LLMs with real benchmarks. Throughout our experi-

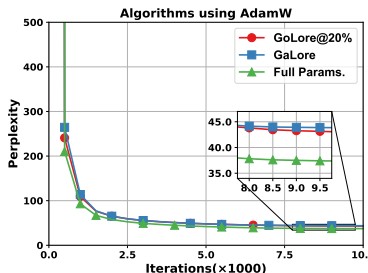 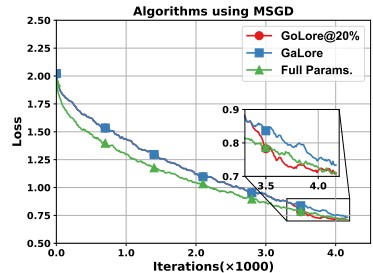

Figure 3: Pre-training curves of various approaches using AdamW with BF16 precision.

Figure 4: Fine-tuning curves of various approaches using MSGD with BF16 precision.

Table 2: Fine-tuning results on GLUE benchmark using pre-trained RoBERTa-Base.

| Algorithm | CoLA | STS-B | MRPC | RTE | SST2 | MNLI | QNLI | QQP | Avg |
|---|---|---|---|---|---|---|---|---|---|
| Full Params. | 62.07 | 90.18 | 92.25 | 78.34 | 94.38 | 87.59 | 92.46 | 91.90 | 86.15 |
| GaLore | 61.32 | 90.24 | 92.55 | 77.62 | **94.61** | 86.92 | 92.06 | 90.84 | 85.77 |
| FLORA | 57.71 | 89.59 | 91.96 | 76.17 | 94.50 | 85.42 | 91.93 | 90.49 | 84.72 |
| GoLore@20% | **61.66** | **90.55** | **92.93** | **78.34** | **94.61** | 87.02 | 92.20 | 90.91 | **86.03** |

ments, *GoLore@x%* uses GaLore in the first $(100 - x)\%$ iterations and GoLore in the last $x\%$ iterations, *L.B. GaLore* denotes large-batch GaLore, and *Full Params.* denotes full-parameter training. Further results and detailed experimental specifications including the hyperparameter choices and computing resources are deferred to Appendix E.

**GaLore's non-convergence.** To validate the non-convergence of GaLore and the convergence properties of GoLore and large-batch GaLore, we compare them with full-parameter training on the constructed quadratic problem defined in (1). Fig. 1 shows that, regardless of whether AdamW or MSGD is employed as the subspace optimizer, GaLore does not converge to the desired solution. In contrast, both GoLore and large-batch GaLore, along with full-parameter training, achieve exact convergence, thereby validating our theoretical results.

**Pre-training.** To validate the efficiency of GoLore in LLM pre-training tasks, we pre-trained LLaMA-60M on the C4 (Raffel et al., 2020) dataset for 10,000 iterations using various algorithms, including GaLore, GoLore and full-parameter training. All implementations utilized the AdamW optimizer in BF16 format. As illustrated in Fig. 3, there is a noticeable performance gap between GaLore/GoLore and full-parameter training, indicating that the parameters are away from local minima. However, GoLore still demonstrates slightly better training performance compared to GaLore.

**Fine-tuning.** To validate the efficiency of GoLore in LLM fine-tuning tasks, we fine-tuned pre-trained LLaMA2-7B models (Touvron et al., 2023) on the WinoGrande dataset (Sakaguchi et al., 2021) and pre-trained RoBERTa models (Liu, 2019) on the GLUE benchmark (Wang, 2018) with AdamW optimizers. Fig. 4 displays the fine-tuning loss curves for GaLore and GoLore with rank 1024, while Table 2 presents the task scores for GaLore/GoLore and FLORA with rank 4. In both experiments, GoLore outperforms GaLore.

## 8 CONCLUSION AND LIMITATIONS

This paper investigates subspace optimization approaches for LLM pre-training and fine-tuning. We demonstrate that GaLore fails to converge to the desired solution under regular assumptions, as the SVD-based projection often generates noise-dominated subspaces when the true gradient is relatively small. However, we establish that GaLore can achieve exact convergence when using deterministic or large-batch stochastic gradients. We further introduce GoLore—a variant of Ga-Lore employing randomly sampled projection matrices—and establish its convergence rate even with small-batch stochastic gradients. A limitation of this paper is that convergence guarantees for GoLore are currently provided only when using MSGD as the subspace optimizer. Although GoLore with AdamW performs well empirically, as shown in Table 2, its theoretical convergence guarantees remain unknown and will be addressed in future work.

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

# APPENDIX

## A  CHALLENGES IN THEORETICAL ANALYSIS

Gradient projection onto a low-rank subspace poses two significant challenges for the convergence analysis of (momentum) stochastic gradient descent:

- **Neither unbiased nor contractive compression.** gradient projection onto this subspace can be viewed as gradient compression. Traditional analyses of optimization algorithms with lossy compression typically rely on either unbiased (Li et al., 2020; Li & Richtárik, 2021; Huang & Pu, 2023; He et al., 2024a;b; Condat et al., 2024) compressibility, *i.e.*, the compressor $\mathcal{C}$ satisfies

$$\mathbb{E}[\mathcal{C}(\boldsymbol{x})] = \boldsymbol{x}, \quad \mathbb{E}[\|\mathcal{C}(\boldsymbol{x}) - \boldsymbol{x}\|_2^2] \leq \omega\|\boldsymbol{x}\|_2^2, \quad \forall \boldsymbol{x} \in \mathbb{R}^d,$$

  for some $\omega \geq 0$, or contractive (Richtárik et al., 2021; Xie et al., 2020; Fatkhullin et al., 2024; He et al., 2023) compressibility, *i.e.*,

$$\mathbb{E}[\|\mathcal{C}(\boldsymbol{x}) - \boldsymbol{x}\|_2^2] \leq (1 - \delta)\|\boldsymbol{x}\|_2^2, \quad \forall \boldsymbol{x} \in \mathbb{R}^d,$$

  for some $\delta \in (0, 1]$. However, GaLore's subspace compression is neither unbiased nor contractive due to the reuse of projection matrices. For example, consider a pre-computed projection matrix $\boldsymbol{P} \in \mathbb{R}^{m \times r}$. There exists a full-parameter gradient $\boldsymbol{G} \in \mathbb{R}^{m \times n}$ such that $\boldsymbol{G} \neq 0$ and $\mathcal{C}(\boldsymbol{G}) := \boldsymbol{P}\boldsymbol{P}^\top\boldsymbol{G} = 0$, violating both unbiased and contractive compressibility.

- **Periodically projected optimizer states.** When GaLore changes the subspace, the retained momentum terms must be adjusted to track the gradients in the new subspace. Since these momentum terms were initially aligned with the gradients in the original subspace, such adjustments inevitably introduce additional errors, especially when the two subspaces differ significantly. In the extreme case where the two subspaces are entirely orthogonal, the momentum from the previous subspace becomes largely irrelevant for optimization in the new one.

## B  THEORETICAL PROOFS

### B.1  NOTATIONS AND USEFUL LEMMAS

We assume the model parameters consist of $N_L$ weight matrices. We use $\boldsymbol{X}_\ell \in \mathbb{R}^{m_\ell \times n_\ell}$ to denote the $\ell$-th weight matrix and $\boldsymbol{x} \in \mathbb{R}^d = (\text{vec}(\boldsymbol{X}_1)^\top, \cdots, \text{vec}(\boldsymbol{X}_{N_L})^\top)^\top$ to denote the vector collecting all the parameters, $d = \sum_{\ell=1}^{N_L} m_\ell n_\ell$. We assume GaLore/GoLore applies rank-$r_\ell$ projection to the $\ell$-th weight matrix and denote

$$\delta_\ell = \frac{r_\ell}{\min\{m_\ell, n_\ell\}}, \quad \underline{\delta} = \min_{1 \leq \ell \leq N_L} \delta_\ell, \quad \overline{\delta} = \max_{1 \leq \ell \leq N_l} \delta_\ell.$$

We define $\tilde{\boldsymbol{M}}_\ell^{(t)}$ as

$$\tilde{\boldsymbol{M}}_\ell^{(t)} = \begin{cases} \boldsymbol{P}_\ell^{(t)}\boldsymbol{M}_\ell^{(t)}, & \text{if } m_\ell \leq n_\ell, \\ \boldsymbol{M}_\ell^{(t)}(\boldsymbol{Q}_\ell^{(t)})^\top, & \text{if } m_\ell > n_\ell, \end{cases}$$

and $\tilde{\boldsymbol{m}} = (\text{vec}(\tilde{\boldsymbol{M}}_1)^\top, \cdots, \text{vec}(\tilde{\boldsymbol{M}}_{N_L})^\top)^\top$. While using Alg. 1 with MSGD and MP, it holds for $m_\ell \leq n_\ell$ that

$$\tilde{\boldsymbol{M}}_\ell^{(t)} = \begin{cases} \beta_1 \boldsymbol{P}_\ell^{(0)}(\boldsymbol{P}_\ell^{(0)})^\top\boldsymbol{G}_\ell^{(0)}, & t = 0; \\ \boldsymbol{P}_\ell^{(t)}(\boldsymbol{P}_\ell^{(t)})^\top \left((1 - \beta_1)\tilde{\boldsymbol{M}}_\ell^{(t-1)} + \beta_1\boldsymbol{G}_\ell^{(t)}\right), & t = k\tau, \ k \in \mathbb{N}^*; \\ (1 - \beta_1)\tilde{\boldsymbol{M}}_\ell^{(t-1)} + \beta_1\boldsymbol{P}_\ell^{(t)}(\boldsymbol{P}_\ell^{(t)})^\top\boldsymbol{G}_\ell^{(t)}, & t = k\tau + r, \ k \in \mathbb{N}, \ 1 \leq r < \tau; \end{cases}$$

for $m_\ell > n_\ell$ that

$$\tilde{\boldsymbol{M}}_\ell^{(t)} = \begin{cases} \beta_1 \boldsymbol{G}_\ell^{(0)}\boldsymbol{Q}_\ell^{(0)}(\boldsymbol{Q}_\ell^{(0)})^\top, & t = 0; \\ \left((1 - \beta_1)\tilde{\boldsymbol{M}}_\ell^{(t-1)} + \beta_1\boldsymbol{G}_\ell^{(t)}\right)\boldsymbol{Q}_\ell^{(t)}(\boldsymbol{Q}_\ell^{(t)})^\top, & t = k\tau, \ k \in \mathbb{N}^*; \\ (1 - \beta_1)\tilde{\boldsymbol{M}}_\ell^{(t-1)} + \beta_1\boldsymbol{G}_\ell^{(t)}\boldsymbol{Q}_\ell^{(t)}(\boldsymbol{Q}_\ell^{(t)})^\top, & t = k\tau + r, \ k \in \mathbb{N}, \ 1 \leq r < \tau; \end{cases}$$

and for both cases that

$$\boldsymbol{X}_\ell^{(t+1)} = \boldsymbol{X}_\ell^{(t)} - \eta \tilde{\boldsymbol{M}}_\ell^{(t)}.$$

**Lemma 1** (Error of GaLore's projection). *Let $\boldsymbol{G} = \boldsymbol{U}\boldsymbol{\Sigma}\boldsymbol{V}^\top$ be the SVD of $\boldsymbol{G} \in \mathbb{R}^{m \times n}$, projection matrix $\boldsymbol{P} = \boldsymbol{U}[:,:r]$, $\boldsymbol{Q} = \boldsymbol{V}[:,:r]$, $r < \min\{m,n\}$. It holds for $m \le n$ that*

$$\|\boldsymbol{P}\boldsymbol{P}^\top\boldsymbol{G} - \boldsymbol{G}\|_F^2 \le \left(1 - \frac{r}{m}\right)\|\boldsymbol{G}\|_F^2,$$

*and for $m > n$ that*

$$\|\boldsymbol{G}\boldsymbol{Q}\boldsymbol{Q}^\top - \boldsymbol{G}\|_F^2 \le \left(1 - \frac{r}{n}\right)\|\boldsymbol{G}\|_F^2.$$

*Proof.* Without loss of generality assume $m \le n$ (the other case can be proved similarly). Let $\boldsymbol{Q} = \boldsymbol{U}[:,(r+1):]$, It holds that $\boldsymbol{I} = \boldsymbol{U}\boldsymbol{U}^\top = \boldsymbol{P}\boldsymbol{P}^\top + \boldsymbol{Q}\boldsymbol{Q}^\top$. Thus,

$$\begin{aligned}
\|\boldsymbol{P}\boldsymbol{P}^\top\boldsymbol{G} - \boldsymbol{G}\|_F^2 &= \|(\boldsymbol{I} - \boldsymbol{P}\boldsymbol{P}^\top)\boldsymbol{U}\boldsymbol{\Sigma}\boldsymbol{V}^\top\|_F^2 \\
&= \mathrm{tr}(\boldsymbol{V}\boldsymbol{\Sigma}^\top\boldsymbol{U}^\top(\boldsymbol{I} - \boldsymbol{P}\boldsymbol{P}^\top)^2\boldsymbol{U}\boldsymbol{\Sigma}\boldsymbol{V}^\top) \\
&= \mathrm{tr}(\boldsymbol{\Sigma}^\top\boldsymbol{U}^\top\boldsymbol{Q}\boldsymbol{Q}^\top\boldsymbol{U}\boldsymbol{\Sigma}),
\end{aligned} \tag{3}$$

where the second equation uses $\|\boldsymbol{X}\|_F^2 = \mathrm{tr}(\boldsymbol{X}^\top\boldsymbol{X})$ and the last equation uses $\mathrm{tr}(\boldsymbol{A}\boldsymbol{B}) = \mathrm{tr}(\boldsymbol{B}\boldsymbol{A})$, $\boldsymbol{V}^\top\boldsymbol{V} = \boldsymbol{I}$ and $\boldsymbol{Q}^\top\boldsymbol{Q} = \boldsymbol{I}$. By $\boldsymbol{Q}^\top\boldsymbol{P} = 0$ and $\boldsymbol{P}^\top\boldsymbol{Q} = 0$, we have

$$\boldsymbol{U}^\top\boldsymbol{Q}\boldsymbol{Q}^\top\boldsymbol{U} = \begin{pmatrix} \boldsymbol{P}^\top \\ \boldsymbol{Q}^\top \end{pmatrix}\boldsymbol{Q}\boldsymbol{Q}^\top \begin{pmatrix} \boldsymbol{P} & \boldsymbol{Q} \end{pmatrix} = \begin{pmatrix} 0_{r\times r} & 0_{r\times(m-r)} \\ 0_{(m-r)\times r} & \boldsymbol{I}_{m-r} \end{pmatrix}. \tag{4}$$

Let $\sigma_1 \ge \sigma_2 \ge \cdots \ge \sigma_m \ge 0$ denote the eigenvalues of $\boldsymbol{G}$, (4) implies

$$\boldsymbol{\Sigma}^\top\boldsymbol{U}^\top\boldsymbol{Q}\boldsymbol{Q}^\top\boldsymbol{U}\boldsymbol{\Sigma} = \begin{pmatrix} 0_{r\times r} & 0_{r\times(m-r)} & 0_{r\times(n-m)} \\ 0_{(m-r)\times r} & \mathrm{diag}(\sigma_{r+1},\cdots,\sigma_m) & 0_{(m-r)\times(n-m)} \\ 0_{(n-m)\times r} & 0_{(n-m)\times(m-r)} & 0_{(n-m)\times(n-m)} \end{pmatrix}. \tag{5}$$

Applying (5) to (3) yields

$$\|\boldsymbol{P}\boldsymbol{P}^\top\boldsymbol{G} - \boldsymbol{G}\|_F^2 = \mathrm{tr}(\boldsymbol{\Sigma}^\top\boldsymbol{U}^\top\boldsymbol{Q}\boldsymbol{Q}^\top\boldsymbol{U}\boldsymbol{\Sigma}) = \sum_{i=r+1}^m \sigma_i^2 \le \frac{m-r}{m}\|\boldsymbol{G}\|_F^2,$$

where the inequality uses $\|\boldsymbol{G}\|_F^2 = \mathrm{tr}(\boldsymbol{G}^\top\boldsymbol{G}) = \mathrm{tr}(\boldsymbol{\Sigma}^\top\boldsymbol{\Sigma}) = \sum_{i=1}^m \sigma_i^2$. $\qquad\square$

**Lemma 2** (Gradient connections). *It holds for any $t$, $\tau > 0$ that*

$$\|\nabla_\ell f(\boldsymbol{x}^{(0)})\|_F^2 \le \frac{2}{\tau}\sum_{r=0}^{\tau-1}\|\nabla_\ell f(\boldsymbol{x}^{(t+r)})\|_F^2 + (\tau-1)\sum_{r=0}^{\tau-2}\|\nabla_\ell f(\boldsymbol{x}^{(t+r+1)}) - \nabla_\ell f(\boldsymbol{x}^{(t+r)})\|_F^2. \tag{6}$$

*Proof.* For any $r = 1, \cdots, \tau-1$, it holds that

$$\begin{aligned}
\|\nabla_\ell f(\boldsymbol{x}^{(t)})\|_F^2 &= \|\nabla_\ell f(\boldsymbol{x}^{(t+r)}) - (\nabla_\ell f(\boldsymbol{x}^{(t+r)}) - \nabla_\ell f(\boldsymbol{x}^{(t)}))\|_F^2 \\
&\le 2\|\nabla_\ell f(\boldsymbol{x}^{(t+r)})\|_F^2 + 2\|\nabla_\ell f(\boldsymbol{x}^{(t+r)}) - \nabla_\ell f(\boldsymbol{x}^{(t)})\|_F^2.
\end{aligned} \tag{7}$$

For any $r = 2, \cdots, \tau-1$, it holds that

$$\begin{aligned}
\|\nabla_\ell f(\boldsymbol{x}^{(t+r)}) - \nabla_\ell f(\boldsymbol{x}^{(t)})\|_F^2 &= \left\|\sum_{i=1}^r \nabla_\ell f(\boldsymbol{x}^{(t+i)}) - \nabla_\ell f(\boldsymbol{x}^{(t+i-1)})\right\|_F^2 \\
&\le r\sum_{i=1}^r \|\nabla_\ell f(\boldsymbol{x}^{(t+i)}) - \nabla_\ell f(\boldsymbol{x}^{(t+i-1)})\|_F^2,
\end{aligned} \tag{8}$$

where the inequality uses Cauchy's inequality. Summing (7) from $r = 1$ to $\tau - 1$ and applying (8) yields

$$
\begin{aligned}
\tau\|\nabla_\ell f(\boldsymbol{x}^{(t)})\|_F^2 \leq& 2\sum_{r=0}^{\tau-1}\|\nabla_\ell f(\boldsymbol{x}^{(t+r)})\|_F^2 + 2\sum_{i=1}^{\tau-1}\sum_{j=1}^{i} i\|\nabla_\ell f(\boldsymbol{x}^{(t+j)}) - \nabla_\ell f(\boldsymbol{x}^{(t+j-1)})\|_F^2 \\
\leq& 2\sum_{r=0}^{\tau-1}\|\nabla_\ell f(\boldsymbol{x}^{(t+r)})\|_F^2 + 2\sum_{j=1}^{\tau-1}\sum_{i=1}^{\tau-1} i\|\nabla_\ell f(\boldsymbol{x}^{(t+j)}) - \nabla_\ell f(\boldsymbol{x}^{(t+j-1)})\|_F^2 \\
=& 2\sum_{r=0}^{\tau-1}\|\nabla_\ell f(\boldsymbol{x}^{(t+r)})\|_F^2 + \tau(\tau-1)\sum_{j=1}^{\tau-1}\|\nabla_\ell f(\boldsymbol{x}^{(t+j)}) - \nabla_\ell f(\boldsymbol{x}^{(t+j-1)})\|_F^2,
\end{aligned}
$$

which is exactly (6). $\qquad\square$

**Lemma 3** (Projection orthogonality). *If $\boldsymbol{P} \in \mathrm{St}_{m,r}$, it holds for any $\boldsymbol{A}, \boldsymbol{B} \in \mathbb{R}^{m \times n}$ that*

$$\|\boldsymbol{P}\boldsymbol{P}^\top \boldsymbol{A} + (\boldsymbol{I} - \boldsymbol{P}\boldsymbol{P}^\top)\boldsymbol{B}\|_F^2 = \|\boldsymbol{P}\boldsymbol{P}^\top \boldsymbol{A}\|_F^2 + \|(\boldsymbol{I} - \boldsymbol{P}\boldsymbol{P}^\top)\boldsymbol{B}\|_F^2. \tag{9}$$

*Proof.* By definition we have $\boldsymbol{P}^\top \boldsymbol{P} = \boldsymbol{I}$. It suffices to note that

$$\langle \boldsymbol{P}\boldsymbol{P}^\top \boldsymbol{A}, (\boldsymbol{I} - \boldsymbol{P}\boldsymbol{P}^\top)\boldsymbol{B}\rangle_F = \mathrm{tr}(\boldsymbol{A}^\top \boldsymbol{P}\boldsymbol{P}^\top(\boldsymbol{I} - \boldsymbol{P}\boldsymbol{P}^\top)\boldsymbol{B}) = \mathrm{tr}(0) = 0.$$

$\qquad\square$

**Lemma 4** (Descent lemma). *Under Assumption 2, for update*

$$\boldsymbol{x}^{(t+1)} = \boldsymbol{x}^{(t)} - \eta\tilde{\boldsymbol{m}}^{(t)},$$

*it holds that*

$$
\begin{aligned}
f(\boldsymbol{x}^{(t+1)}) \leq& f(\boldsymbol{x}^{(t)}) - \left(\frac{1}{2\eta} - \frac{L}{2}\right)\|\boldsymbol{x}^{(t+1)} - \boldsymbol{x}^{(t)}\|_2^2 + \frac{\eta}{2}\|\tilde{\boldsymbol{m}}^{(t)} - \nabla f(\boldsymbol{x}^{(t)})\|_2^2 \\
& - \frac{\eta}{2}\|\nabla f(\boldsymbol{x}^{(t)})\|_2^2.
\end{aligned} \tag{10}
$$

*Proof.* By $L$-smoothness of $f$ (Assumption 2) we have

$$
\begin{aligned}
& f(\boldsymbol{x}^{(t+1)}) - f(\boldsymbol{x}^{(t)}) \\
\leq& \langle \nabla f(\boldsymbol{x}^{(t)}), \boldsymbol{x}^{(t+1)} - \boldsymbol{x}^{(t)}\rangle + \frac{L}{2}\|\boldsymbol{x}^{(t+1)} - \boldsymbol{x}^{(t)}\|_2^2 \\
=& \left\langle \frac{\tilde{\boldsymbol{m}}^{(t)}}{2}, \boldsymbol{x}^{(t+1)} - \boldsymbol{x}^{(t)}\right\rangle + \left\langle \nabla f(\boldsymbol{x}^{(t)}) - \frac{\tilde{\boldsymbol{m}}^{(t)}}{2}, \boldsymbol{x}^{(t+1)} - \boldsymbol{x}^{(t)}\right\rangle + \frac{L}{2}\|\boldsymbol{x}^{(t+1)} - \boldsymbol{x}^{(t)}\|_2^2 \\
=& -\left(\frac{1}{2\eta} - \frac{L}{2}\right)\|\boldsymbol{x}^{(t+1)} - \boldsymbol{x}^{(t)}\|_2^2 + \frac{\eta}{2}\|\nabla f(\boldsymbol{x}^{(t)}) - \tilde{\boldsymbol{m}}^{(t)}\|_2^2 - \frac{\eta}{2}\|\nabla f(\boldsymbol{x}^{(t)})\|_2^2,
\end{aligned}
$$

which is exactly (10). $\qquad\square$

**Lemma 5** (Error of GoLore's projection). *Let $\boldsymbol{P} \sim \mathcal{U}(\mathrm{St}_{m,r})$, $\boldsymbol{Q} \sim \mathcal{U}(\mathrm{St}_{n,r})$, it holds for all $\boldsymbol{G} \in \mathbb{R}^{m \times n}$ that*

$$\mathbb{E}[\boldsymbol{P}\boldsymbol{P}^\top] = \frac{r}{m} \cdot \boldsymbol{I}, \quad \mathbb{E}[\boldsymbol{Q}\boldsymbol{Q}^\top] = \frac{r}{n} \cdot \boldsymbol{I}, \tag{11}$$

*and*

$$\mathbb{E}[\|\boldsymbol{P}\boldsymbol{P}^\top \boldsymbol{G} - \boldsymbol{G}\|_F^2] = \left(1 - \frac{r}{m}\right)\|\boldsymbol{G}\|_F^2, \quad \mathbb{E}[\|\boldsymbol{G}\boldsymbol{Q}\boldsymbol{Q}^\top - \boldsymbol{G}\|_F^2] = \left(1 - \frac{r}{n}\right)\|\boldsymbol{G}\|_F^2. \tag{12}$$

*Proof.* We refer the proof of (11) to Theorem 2.2.2 in Chikuse (2012). By $\boldsymbol{P}^\top \boldsymbol{P} = \boldsymbol{I}$, we have

$$
\begin{aligned}
\mathbb{E}[\|\boldsymbol{P}\boldsymbol{P}^\top \boldsymbol{G} - \boldsymbol{G}\|_F^2] =& \mathbb{E}[\mathrm{tr}(\boldsymbol{G}^\top(\boldsymbol{I} - \boldsymbol{P}\boldsymbol{P}^\top)^2\boldsymbol{G})] \\
=& \mathbb{E}[\mathrm{tr}(\boldsymbol{G}^\top(\boldsymbol{I} - \boldsymbol{P}\boldsymbol{P}^\top)\boldsymbol{G})] \\
=& \mathrm{tr}(\boldsymbol{G}^\top(\boldsymbol{I} - \mathbb{E}[\boldsymbol{P}\boldsymbol{P}^\top])\boldsymbol{G}).
\end{aligned} \tag{13}
$$

Applying (11) to (13) yields

$$\mathbb{E}[\|\boldsymbol{P}\boldsymbol{P}^\top\boldsymbol{G} - \boldsymbol{G}\|_F^2] = \mathrm{tr}\left(\boldsymbol{G}^\top\left(\boldsymbol{I} - \frac{r}{m}\boldsymbol{I}\right)\boldsymbol{G}\right)$$
$$= \left(1 - \frac{r}{m}\right)\mathrm{tr}(\boldsymbol{G}^\top\boldsymbol{G})$$
$$= \left(1 - \frac{r}{m}\right)\|\boldsymbol{G}\|_F^2.$$

The other part of (12) can be proved similarly. □

### B.2 Non-convergence of GaLore

In this subsection, we present the proof for Theorem 1. We first restate Theorem 1 as follows:

**Theorem 5** (Non-convergence of GaLore). *There exists an objective function $f : \mathbb{R}^d \to \mathbb{R}$ satisfying Assumptions 1, 2, a stochastic gradient oracle $(F, \mathcal{D})$ satisfying Assumption 3, an initial point $\boldsymbol{x}^{(0)} \in \mathbb{R}^d$, a constant $\epsilon_0 > 0$ such that for GaLore with any rank $r_\ell < \min\{m_\ell, n_\ell\}$, subspace changing frequency $\tau$, any subspace optimizer $\rho$ with arbitrary hyperparameters and any $t > 0$, it holds that*

$$\|\nabla f(\boldsymbol{x}^{(t)})\|_2^2 \geq \epsilon_0.$$

*Proof.* Consider target function $f(\boldsymbol{X}) = \frac{L}{2}\mathrm{tr}(\boldsymbol{X}^\top\boldsymbol{p}\boldsymbol{p}^\top\boldsymbol{X})$ where $L > 0$, $\boldsymbol{X} \in \mathbb{R}^{n\times n}$ with $n > 1$ and $\boldsymbol{p} = (1, 0, \cdots, 0)^\top \in \mathbb{R}^n$. It holds that

$$f(\boldsymbol{X}) = \frac{L}{2}\|\boldsymbol{p}^\top\boldsymbol{X}\|_2^2 \geq 0,$$

thus $f$ satisfies Assumption 1. Since $\nabla f(\boldsymbol{X}) = L\boldsymbol{p}\boldsymbol{p}^\top\boldsymbol{X}$, it holds that

$$\|\nabla f(\boldsymbol{X}) - \nabla f(\boldsymbol{Y})\|_F = L\|\boldsymbol{p}\boldsymbol{p}^\top(\boldsymbol{X} - \boldsymbol{Y})\|_F \leq L\|\boldsymbol{p}\boldsymbol{p}^\top\|_2\|\boldsymbol{X} - \boldsymbol{Y}\|_F = L\|\boldsymbol{X} - \boldsymbol{Y}\|_F,$$

thus $f$ satisfies Assumption 2.

Consider the following stochastic gradient oracle:

$$F(\boldsymbol{X}; \xi) = f(\boldsymbol{X}) + \xi\tilde{\sigma} \cdot \mathrm{tr}(\boldsymbol{Q}\boldsymbol{Q}^\top\boldsymbol{X}), \quad \text{and} \quad \mathbb{P}_{\xi\sim\mathcal{D}}[\xi = 1] = \mathbb{P}_{\xi\sim\mathcal{D}}[\xi = -1] = 0.5,$$

where $\tilde{\sigma} = \sigma/\sqrt{(n-1)n/2}$ and

$$\boldsymbol{Q} = \begin{pmatrix} 0 \\ \mathrm{diag}\left(1, \sqrt[4]{2}, \cdots, \sqrt[4]{n-1}\right) \end{pmatrix} \in \mathbb{R}^{n\times(n-1)}.$$

Note that $\nabla F(\boldsymbol{X}; \xi) = \nabla f(\boldsymbol{X}) + \xi\tilde{\sigma}\boldsymbol{Q}\boldsymbol{Q}^\top$, it holds for any $\boldsymbol{X} \in \mathbb{R}^{n\times n}$ that

$$\mathbb{E}_{\xi\sim\mathcal{D}}[\nabla F(\boldsymbol{X}; \xi)] = \nabla f(\boldsymbol{X})$$

$$\mathbb{E}_{\xi\sim\mathcal{D}}[\|\nabla F(\boldsymbol{X}; \xi) - \nabla f(\boldsymbol{X})\|_F^2] = \tilde{\sigma}^2\|\boldsymbol{Q}\boldsymbol{Q}^\top\|_F^2 = \frac{\sigma^2}{(n-1)n/2} \cdot \sum_{i=1}^{n-1} i = \sigma^2,$$

thus oracle $(F, \mathcal{D})$ satisfies Assumption 3.

Consider the following initial point:

$$\boldsymbol{X}^{(0)} = \begin{pmatrix} \lambda\boldsymbol{p}^\top \\ \boldsymbol{\Lambda} \end{pmatrix},$$

where $0 < \lambda < \tilde{\sigma}/L$ is a scalar and $\boldsymbol{\Lambda} \in \mathbb{R}^{(n-1)\times n}$ is an arbitrary matrix. We show that GaLore with the above objective function $f$, stochastic gradient oracle $(F, \mathcal{D})$, initial point $\boldsymbol{X}^{(0)}$, arbitrary rank $0 < r < n$, arbitrary subspace changing frequency $\tau$ and arbitrary subspace optimizer $\rho$, can only output points $\boldsymbol{X}^{(t)}$ with $\|\nabla f(\boldsymbol{X}^{(t)})\|_F^2 \geq \epsilon_0$ for $\epsilon_0 = L^2\lambda^2 > 0$.

When $\tau \mid t$, GaLore recomputes the subspace projection matrix at iteration $t$. If the first row of $\boldsymbol{X}^{(t)}$ equals $\lambda\boldsymbol{p}^\top$, *i.e.*, $\boldsymbol{X}^{(t)}[1, :] = \lambda\boldsymbol{p}^\top$, the stochastic gradient is given by

$$\boldsymbol{G}^{(t)} = L\boldsymbol{p}\boldsymbol{p}^\top\boldsymbol{X} + \xi^{(t)}\tilde{\sigma}\boldsymbol{Q}\boldsymbol{Q}^\top = \mathrm{diag}\left(L\lambda, \xi^{(t)}\tilde{\sigma}, \sqrt{2}\xi^{(t)}\tilde{\sigma}, \cdots, \sqrt{n-1}\xi^{(t)}\tilde{\sigma}\right).$$

since $L\lambda < \tilde{\sigma}$, computing SVD yields

$$\boldsymbol{G}^{(t)} = \begin{pmatrix} L\lambda & 0 & \cdots & 0 \\ 0 & \xi^{(t)}\tilde{\sigma} & \cdots & 0 \\ \vdots & \vdots & \ddots & \vdots \\ 0 & 0 & \cdots & \sqrt{n-1}\xi^{(t)}\tilde{\sigma} \end{pmatrix}$$

$$= \underbrace{\begin{pmatrix} 0 & \cdots & 0 & \zeta_1 \\ 0 & \cdots & \zeta_2 & 0 \\ \vdots & \ddots & \vdots & \vdots \\ \zeta_n & \cdots & 0 & 0 \end{pmatrix}}_{:=\boldsymbol{U}} \underbrace{\begin{pmatrix} \sqrt{n-1}\tilde{\sigma} & \cdots & 0 & 0 \\ \vdots & \ddots & \vdots & \vdots \\ 0 & \cdots & \tilde{\sigma} & 0 \\ 0 & \cdots & 0 & L\lambda \end{pmatrix}}_{:=\boldsymbol{\Sigma}} \underbrace{\begin{pmatrix} 0 & 0 & \cdots & \zeta_n\xi^{(t)} \\ \vdots & \vdots & \ddots & \vdots \\ 0 & \zeta_2\xi^{(t)} & \cdots & 0 \\ \zeta_1 & 0 & \cdots & 0 \end{pmatrix}}_{:=\boldsymbol{V}^\top},$$

where $\zeta_1, \cdots, \zeta_n \in \{-1, 1\}$. For any rank $r < n$, the projection matrix is thus

$$\boldsymbol{P}^{(t)} = \begin{pmatrix} 0 & 0 & \cdots & 0 \\ \vdots & \vdots & \ddots & \vdots \\ 0 & 0 & \cdots & 0 \\ 0 & 0 & \cdots & \zeta_{n-r+1} \\ \vdots & \vdots & \ddots & \vdots \\ 0 & \zeta_{n-1} & \cdots & 0 \\ \zeta_n & 0 & \cdots & 0 \end{pmatrix} \in \mathbb{R}^{n \times r}.$$

Using this projection matrix, the subspace updates in the following $\tau$ iterations is as

$$\boldsymbol{X}^{(t+\Delta_t)} = \boldsymbol{X}^{(t)} + \boldsymbol{P}^{(t)} \sum_{s=0}^{\Delta_t-1} \rho^{(t+s)}((\boldsymbol{P}^{(t)})^\top \boldsymbol{G}^{(t)}) \quad \Rightarrow \quad \boldsymbol{X}^{(t+\Delta_t)}[1,:] = \boldsymbol{X}^{(t)}[1,:] = \lambda \boldsymbol{p}^\top,$$

for $\Delta_t = 1, 2, \cdots, \tau$. Since $\boldsymbol{X}^{(0)}[1,:] = \lambda \boldsymbol{p}^\top$, it holds for all $t > 0$ that $\boldsymbol{X}^{(t)}[1,:] = \lambda \boldsymbol{p}^\top$ and thus

$$\|\nabla f(\boldsymbol{X}^{(t)})\|_F^2 = L^2\lambda^2 = \epsilon_0.$$

$\square$

**Remark.** When setting $\boldsymbol{B} = 0$ in the quadratic problem setting (Sec. 7), the quadratic problem is equivalent to the counter-example we construct in the proof of Theorem 5. The illustration in Fig. 5 displays the loss curves for this problem.

### B.3 CONVERGENCE OF DETERMINISTIC GALORE

In this subsection, we present the proof for Theorem 2. GaLore using deterministic gradients and MSGD with MP is specified as Alg. 2.

**Lemma 6** (Momentum contraction). *In deterministic GaLore using MSGD with MP (Alg. 2), if $0 < \beta_1 \leq 1$, term $\tilde{\boldsymbol{M}}_\ell^{(t)}$ has the following contraction properties:*

- *When $t = 0$, it holds that*

$$\|\tilde{\boldsymbol{M}}_\ell^{(0)} - \nabla_\ell f(\boldsymbol{X}^{(0)})\|_F^2 \leq (\tau-1)(1-\delta_\ell\beta_1) \sum_{r=0}^{\tau-2} \|\nabla_\ell f(\boldsymbol{x}^{(r+1)}) - \nabla_\ell f(\boldsymbol{x}^{(r)})\|_F^2$$

$$+ \frac{2(1-\delta_\ell\beta_1)}{\tau} \sum_{r=0}^{\tau-1} \|\nabla_\ell f(\boldsymbol{x}^{(r)})\|_F^2; \quad (14)$$

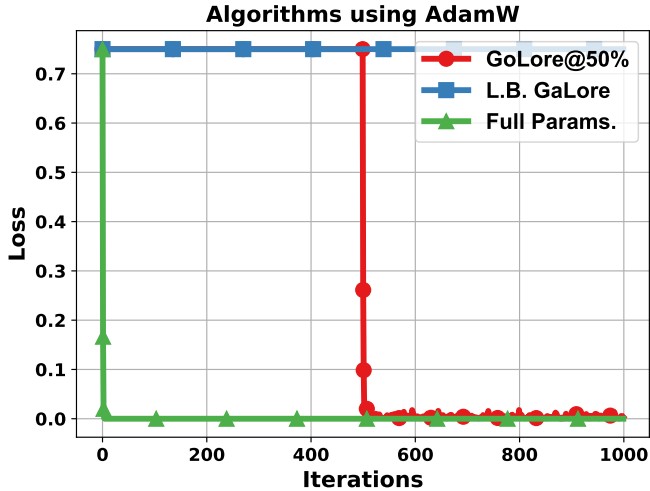

Figure 5: Loss curves of algorithms using AdamW. *GoLore@50%* uses GaLore in the first half and shifts to GoLore in the last half, *Full Params.* denotes full-parameter training.

- *When $t = k\tau$, $k \in \mathbb{N}^*$, it holds that*

$$\|\tilde{\boldsymbol{M}}_\ell^{(t)} - \nabla_\ell f(\boldsymbol{x}^{(t)})\|_F^2 - \left(1 - \left(1 - \frac{\delta_\ell}{4}\right)\beta_1\right)\|\tilde{\boldsymbol{M}}_\ell^{(t-1)} - \nabla_\ell f(\boldsymbol{x}^{(t-1)})\|_F^2$$

$$\leq \frac{2(1-\delta_\ell)}{\tau}\sum_{r=0}^{\tau-1}\|\nabla_l f(\boldsymbol{x}^{(k\tau+r)})\|_F^2 + \frac{5(1-\beta_1)}{\delta_\ell\beta_1}\|\nabla_\ell f(\boldsymbol{x}^{(t)}) - \nabla_\ell f(\boldsymbol{x}^{(t-1)})\|_F^2$$

$$+ (\tau-1)(1-\delta_\ell)\sum_{r=0}^{\tau-2}\|\nabla_\ell f(\boldsymbol{x}^{(k\tau+r+1)}) - \nabla_\ell f(\boldsymbol{x}^{(k\tau+r)})\|_F^2; \tag{15}$$

- *When $t = k\tau + r$, $k \in \mathbb{N}$, $1 \leq r < \tau$, it holds that*

$$\|\tilde{\boldsymbol{M}}_\ell^{(t)} - \nabla_\ell f(\boldsymbol{x}^{(t)})\|_F^2 - \left(1 - \left(1 - \frac{\delta_\ell}{4}\right)\beta_1\right)\|\tilde{\boldsymbol{M}}_\ell^{(t-1)} - \nabla_\ell f(\boldsymbol{x}^{(t-1)})\|_F^2$$

$$\leq \left(1 - \frac{\delta_\ell}{2}\right)\beta_1\|\nabla_\ell f(\boldsymbol{x}^{(t)})\|_F^2 + \frac{5(1-\beta_1)}{\delta_\ell\beta_1}\|\nabla_\ell f(\boldsymbol{x}^{(t)}) - \nabla_\ell f(\boldsymbol{x}^{(t-1)})\|_F^2$$

$$+ \frac{10r\beta_1}{\delta_\ell}\sum_{i=1}^{r}\|\nabla_\ell f(\boldsymbol{x}^{(k\tau+i)}) - \nabla_\ell f(\boldsymbol{x}^{(k\tau+i-1)})\|_F^2. \tag{16}$$

*Proof.* Without loss of generality assume $m_\ell \leq n_\ell$ (the other case can be proved similarly). When $t = 0$, we have

$$\|\tilde{\boldsymbol{M}}_\ell^{(0)} - \nabla_\ell f(\boldsymbol{x}^{(0)})\|_F^2 = \|\beta_1(\boldsymbol{P}_\ell^{(0)}(\boldsymbol{P}_\ell^{(0)})^\top - \boldsymbol{I})\nabla_\ell f(\boldsymbol{x}^{(0)}) - (1-\beta_1)\nabla_\ell f(\boldsymbol{x}^{(0)})\|_F^2$$

$$\leq \beta_1(1-\delta_\ell)\|\nabla_\ell f(\boldsymbol{x}^{(0)})\|_F^2 + (1-\beta_1)\|\nabla_\ell f(\boldsymbol{x}^{(0)})\|_F^2$$

$$= (1 - \delta_\ell\beta_1)\|\nabla_\ell f(\boldsymbol{x}^{(0)})\|_F^2, \tag{17}$$

where the inequality uses Lemma 1 and Jensen's inequality. Applying Lemma 2 to (17) yields (14).

When $t = k\tau$, $k \in \mathbb{N}^*$, we have

$$\|\tilde{\boldsymbol{M}}_\ell^{(t)} - \nabla_\ell f(\boldsymbol{x}^{(t)})\|_F^2$$

$$= \|\boldsymbol{P}_\ell^{(t)}(\boldsymbol{P}_\ell^{(t)})^\top[(1-\beta_1)\tilde{\boldsymbol{M}}_\ell^{(t-1)} + \beta_1\boldsymbol{G}_\ell^{(t)} - \nabla_\ell f(\boldsymbol{x}^{(t)})] - (\boldsymbol{I} - \boldsymbol{P}_\ell^{(t)}(\boldsymbol{P}_\ell^{(t)})^\top)\nabla_\ell f(\boldsymbol{x}^{(t)})\|_F^2$$

$$= \|\boldsymbol{P}_\ell^{(t)}(\boldsymbol{P}_\ell^{(t)})^\top[(1-\beta_1)(\tilde{\boldsymbol{M}}_\ell^{(t-1)} - \nabla_\ell f(\boldsymbol{x}^{(t)}))]\|_F^2 + \|(\boldsymbol{I} - \boldsymbol{P}_\ell^{(t)}(\boldsymbol{P}_\ell^{(t)})^\top)\nabla_\ell f(\boldsymbol{x}^{(t)})\|_F^2$$

$$\leq \|(1-\beta_1)(\tilde{\boldsymbol{M}}_\ell^{(t-1)} - \nabla_\ell f(\boldsymbol{x}^{(t)}))\|_F^2 + (1-\delta_\ell)\|\nabla_\ell f(\boldsymbol{x}^{(t)})\|_F^2, \tag{18}$$

---

**Algorithm 2** GaLore using deterministic gradients and MSGD with MP

---

**Input:** Initial point $\boldsymbol{x}^{(0)}$, learning rate $\eta$, subspace changing frequency $\tau$, rank $\{r_\ell\}_{\ell=1}^{N_L}$, momentum parameter $\beta_1$.

**Output:** $\{\boldsymbol{x}^{(t)}\}_{t=0}^T$.

  Initialize optimizer state $\{\boldsymbol{M}_\ell^{(-1)}\}_{\ell=1}^{N_L}$ to zero;

  **for** $t = 0, 1, \cdots, T-1$ **do**

    **for** $\ell = 1, 2, \cdots, N_L$ **do**

      $\boldsymbol{G}_\ell^{(t)} \leftarrow \nabla_\ell f(\boldsymbol{x}^{(t)})$;

      **if** $t \equiv 0 \pmod{\tau}$ **then**

        $\boldsymbol{U}, \boldsymbol{\Sigma}, \boldsymbol{V} \leftarrow \text{SVD}(\boldsymbol{G}_\ell^{(t)})$;

        **if** $m_\ell \leq n_\ell$ **then**

          $\boldsymbol{P}_\ell^{(t)} \leftarrow \boldsymbol{U}[:, : r_\ell]$;

          $\boldsymbol{M}_\ell^{(t)} \leftarrow (1 - \beta_1)(\boldsymbol{P}_\ell^{(t)})^\top \boldsymbol{P}_\ell^{(t-1)} \boldsymbol{M}_\ell^{(t-1)} + \beta_1 (\boldsymbol{P}_\ell^{(t)})^\top \boldsymbol{G}_\ell^{(t)}$;

          $\boldsymbol{X}_\ell^{(t+1)} \leftarrow \boldsymbol{X}_\ell^{(t)} - \eta \boldsymbol{P}_\ell^{(t)} \boldsymbol{M}_\ell^{(t)}$;

        **else**

          $\boldsymbol{Q}_\ell^{(t)} \leftarrow \boldsymbol{V}[:, : r_\ell]$;

          $\boldsymbol{M}_\ell^{(t)} \leftarrow (1 - \beta_1)\boldsymbol{M}_\ell^{(t-1)}(\boldsymbol{Q}_\ell^{(t-1)})^\top \boldsymbol{Q}_\ell^{(t)} + \beta_1 \boldsymbol{G}_\ell^{(t)} \boldsymbol{Q}_\ell^{(t)}$;

          $\boldsymbol{X}_\ell^{(t+1)} \leftarrow \boldsymbol{X}_\ell^{(t)} - \eta \boldsymbol{M}_\ell^{(t)}(\boldsymbol{Q}_\ell^{(t)})^\top$;

        **end if**

      **else**

        **if** $m_\ell \leq n_\ell$ **then**

          $\boldsymbol{P}_\ell^{(t)} \leftarrow \boldsymbol{P}_\ell^{(t-1)}$;

          $\boldsymbol{M}_\ell^{(t)} \leftarrow (1 - \beta_1)\boldsymbol{M}_\ell^{(t-1)} + \beta_1 (\boldsymbol{P}_\ell^{(t)})^\top \boldsymbol{G}_\ell^{(t)}$;

          $\boldsymbol{X}_\ell^{(t+1)} \leftarrow \boldsymbol{X}_\ell^{(t)} - \eta \boldsymbol{P}_\ell^{(t)} \boldsymbol{M}_\ell^{(t)}$;

        **else**

          $\boldsymbol{Q}_\ell^{(t)} \leftarrow \boldsymbol{Q}_\ell^{(t-1)}$;

          $\boldsymbol{M}_\ell^{(t)} \leftarrow (1 - \beta_1)\boldsymbol{M}_\ell^{(t-1)} + \beta_1 \boldsymbol{G}_\ell^{(t)} \boldsymbol{Q}_\ell^{(t)}$;

          $\boldsymbol{X}_\ell^{(t+1)} \leftarrow \boldsymbol{X}_\ell^{(t)} - \eta \boldsymbol{M}_\ell^{(t)}(\boldsymbol{Q}_\ell^{(t)})^\top$;

        **end if**

      **end if**

    **end for**

  **end for**

---

where the second equality uses Lemma 3 and $\boldsymbol{G}_\ell^{(t)} = \nabla_\ell f(\boldsymbol{x}^{(t)})$, the inequality uses Lemma 1 and $\|\boldsymbol{P}_\ell^{(t)}(\boldsymbol{P}_\ell^{(t)})^\top\|_2 = 1$. By Young's inequality, we have

$$
\begin{aligned}
&\|\tilde{\boldsymbol{M}}_\ell^{(t-1)} - \nabla_\ell f(\boldsymbol{x}^{(t)})\|_F^2 \\
&= \|(\tilde{\boldsymbol{M}}_\ell^{(t-1)} - \nabla_\ell f(\boldsymbol{x}^{(t-1)})) - (\nabla_\ell f(\boldsymbol{x}^{(t)}) - \nabla_\ell f(\boldsymbol{x}^{(t-1)})\|_F^2 \\
&\leq \left(1 + \frac{\delta_\ell \beta_1}{4}\right)\|\tilde{\boldsymbol{M}}_\ell^{(t-1)} - \nabla_\ell f(\boldsymbol{x}^{(t-1)})\|_F^2 + \left(1 + \frac{4}{\delta_\ell \beta_1}\right)\|\nabla_\ell f(\boldsymbol{x}^{(t)}) - \nabla_\ell f(\boldsymbol{x}^{(t-1)})\|_F^2.
\end{aligned}
\tag{19}
$$

Applying Lemma 2 and (19) to (18) yields (15).

When $t = k\tau + r$, $k \in \mathbb{N}$, $1 \leq r < \tau$, we have

$$
\begin{aligned}
&\|\tilde{\boldsymbol{M}}_\ell^{(t)} - \nabla_\ell f(\boldsymbol{x}^{(t)})\|_F^2 \\
&= \|(1 - \beta_1)(\tilde{\boldsymbol{M}}_\ell^{(t-1)} - \nabla_\ell f(\boldsymbol{x}^{(t)})) + \beta_1(\boldsymbol{P}_\ell^{(t)}(\boldsymbol{P}_\ell^{(t)})^\top - \boldsymbol{I})\nabla_\ell f(\boldsymbol{x}^{(t)})\|_F^2 \\
&\leq (1 - \beta_1)\|\tilde{\boldsymbol{M}}_\ell^{(t-1)} - \nabla_\ell f(\boldsymbol{x}^{(t)})\|_F^2 + \beta_1\|(\boldsymbol{I} - \boldsymbol{P}_\ell^{(k\tau)}(\boldsymbol{P}_\ell^{(k\tau)})^\top)\nabla_\ell f(\boldsymbol{x}^{(t)})\|_F^2,
\end{aligned}
\tag{20}
$$

where the inequality uses Jensen's inequality and $\boldsymbol{P}_\ell^{(t)} = \boldsymbol{P}_\ell^{(t-1)} = \cdots = \boldsymbol{P}_\ell^{(k\tau)}$. The first term can be similarly upper bounded as (19). For the second term, we have

$$
(\boldsymbol{I} - \boldsymbol{P}_\ell^{(k\tau)}(\boldsymbol{P}_\ell^{(k\tau)})^\top)\nabla_\ell f(\boldsymbol{x}^{(t)})\|_F^2
$$

$$
\leq \left(1 + \frac{\delta_\ell}{4}\right)\|(\boldsymbol{I} - \boldsymbol{P}_\ell^{(k\tau)}(\boldsymbol{P}_\ell^{(k\tau)})^\top)\nabla_\ell f(\boldsymbol{x}^{(k\tau)})\|_F^2
$$

$$
+ \left(1 + \frac{4}{\delta_\ell}\right)\|(\boldsymbol{I} - \boldsymbol{P}_\ell^{(k\tau)}(\boldsymbol{P}_\ell^{(k\tau)})^\top)(\nabla_\ell f(\boldsymbol{x}^{(t)}) - \nabla_\ell f(\boldsymbol{x}^{(k\tau)})\|_F^2
$$

$$
\leq \left(1 + \frac{\delta_\ell}{4}\right)(1 - \delta_\ell)\|\nabla_\ell f(\boldsymbol{x}^{(k\tau)})\|_F^2 + \frac{5}{\delta_\ell}\|\nabla_\ell f(\boldsymbol{x}^{(t)}) - \nabla_\ell f(\boldsymbol{x}^{(k\tau)})\|_F^2, \tag{21}
$$

where the first inequality uses Young's inequality and the second inequality uses Lemma 1. By Young's inequality, we have

$$
\|\nabla_\ell f(\boldsymbol{x}^{(k\tau)})\|_F^2 \leq \left(1 + \frac{\delta_\ell}{4}\right)\|\nabla_\ell f(\boldsymbol{x}^{(t)})\|_F^2 + \left(1 + \frac{4}{\delta_\ell}\right)\|\nabla_\ell f(\boldsymbol{x}^{(t)}) - \nabla_\ell f(\boldsymbol{x}^{(k\tau)})\|_F^2. \tag{22}
$$

Note that $t = k\tau + r$, we further have

$$
\|\nabla_\ell f(\boldsymbol{x}^{(t)}) - \nabla_\ell f(\boldsymbol{x}^{(k\tau)})\|_F^2 = \left\|\sum_{i=1}^r \nabla_\ell f(\boldsymbol{x}^{(k\tau+i)}) - \nabla_\ell f(\boldsymbol{x}^{(k\tau+i-1)})\right\|_F^2
$$

$$
\leq r\sum_{i=1}^r \|\nabla_\ell f(\boldsymbol{x}^{(k\tau+i)}) - \nabla_\ell f(\boldsymbol{x}^{(k\tau+i-1)})\|_F^2, \tag{23}
$$

where the inequality uses Cauchy's inequality. Applying (22)(23) to (21) yields

$$
(\boldsymbol{I} - \boldsymbol{P}_\ell^{(k\tau)}(\boldsymbol{P}_\ell^{(k\tau)})^\top)\nabla_\ell f(\boldsymbol{x}^{(t)})\|_F^2
$$

$$
\leq \left(1 - \frac{\delta_\ell}{2}\right)\|\nabla_\ell f(\boldsymbol{x}^{(t)})\|_F^2 + \frac{10r}{\delta_\ell}\sum_{i=1}^r \|\nabla_\ell f(\boldsymbol{x}^{(k\tau+i)}) - \nabla_\ell f(\boldsymbol{x}^{(k\tau+i-1)})\|_F^2. \tag{24}
$$

Applying (19)(24) to (20) yields (16). $\qquad\square$

**Lemma 7** (Momentum error). *Under Assumption 2, if $0 < \beta_1 \leq 1$ in deterministic GaLore using MSGD and MP (Alg. 2), it holds for any $K \geq 1$ that*

$$
\sum_{t=0}^{K\tau-1} \|\tilde{\boldsymbol{m}}^{(t)} - \nabla f(\boldsymbol{x}^{(t)})\|_2^2
$$

$$
\leq \left(\frac{5(1-\beta_1)}{(1-\underline{\delta}/4)\underline{\delta}\beta_1^2} + \frac{5\tau(\tau-1)}{(1-\underline{\delta}/4)\underline{\delta}} + \frac{\tau-1}{(1-\overline{\delta}/4)\beta_1}\right)L^2\sum_{t=0}^{K\tau-2} \|\boldsymbol{x}^{(t+1)} - \boldsymbol{x}^{(t)}\|_2^2
$$

$$
+ \left(\frac{1-\underline{\delta}/2}{1-\underline{\delta}/4} + \frac{2}{(1-\overline{\delta}/4)\tau\beta_1}\right)\sum_{t=0}^{K\tau-1} \|\nabla f(\boldsymbol{x}^{(t)})\|_2^2. \tag{25}
$$

*Proof.* By Lemma 6 we have

$$
\sum_{t=0}^{K\tau-1} \|\tilde{\boldsymbol{M}}_\ell^{(t)} - \nabla_\ell f(\boldsymbol{x}^{(t)})\|_F^2 - \left(1 - \left(1 - \frac{\delta_\ell}{4}\right)\beta_1\right)\sum_{t=0}^{K\tau-2} \|\tilde{\boldsymbol{M}}_\ell^{(t)} - \nabla_\ell f(\boldsymbol{x}^{(t)})\|_F^2
$$

$$
\leq \left(\frac{5(1-\beta_1)}{\delta_\ell\beta_1} + \frac{5\tau(\tau-1)\beta_1}{\delta_\ell} + (\tau-1)\right)\sum_{t=0}^{K\tau-2} \|\nabla_\ell f(\boldsymbol{x}^{(t+1)}) - \nabla_\ell f(\boldsymbol{x}^{(t)})\|_F^2
$$

$$
+ \left(\frac{2}{\tau} + \left(1 - \frac{\delta_\ell}{2}\right)\beta_1\right)\sum_{t=0}^{K\tau-1} \|\nabla_\ell f(\boldsymbol{x}^{(t)})\|_F^2,
$$

which implies

$$
\sum_{t=0}^{K\tau-1} \|\tilde{\boldsymbol{M}}_\ell^{(t)} - \nabla_\ell f(\boldsymbol{x}^{(t)})\|_F^2
$$

$$
\leq \left( \frac{5(1-\beta_1)}{(1-\delta_\ell/4)\delta_\ell\beta_1^2} + \frac{5\tau(\tau-1)}{(1-\delta_\ell/4)\delta_\ell} + \frac{\tau-1}{(1-\delta_\ell/4)\beta_1} \right) \sum_{t=0}^{K\tau-2} \|\nabla_\ell f(\boldsymbol{x}^{(t+1)}) - \nabla_\ell f(\boldsymbol{x}^{(t)})\|_F^2
$$

$$
+ \left( \frac{1-\delta_\ell/2}{1-\delta_\ell/4} + \frac{2}{(1-\delta_\ell/4)\tau\beta_1} \right) \sum_{t=0}^{K\tau-1} \|\nabla_\ell f(\boldsymbol{x}^{(t)})\|_F^2. \tag{26}
$$

Summing (26) for $\ell = 1, \cdots, N_L$ and applying Assumption 2 yields (25). $\qquad\square$

Now we are ready to prove the convergence of Alg. 2.

**Theorem 6** (Convergence of deterministic GaLore). *Under Assumptions 1-2, if hyperparameters*

$$
0 < \beta_1 \leq 1, \quad \tau \geq \frac{64}{3\beta_1\underline{\delta}}, \quad 0 < \eta \leq \min\left\{ \frac{1}{4L}, \sqrt{\frac{3\underline{\delta}\beta_1^2}{80L^2}}, \sqrt{\frac{3\underline{\delta}}{80\tau^2 L^2}}, \sqrt{\frac{3\beta_1}{16\tau L^2}} \right\}, \tag{27}
$$

*GaLore using deterministic gradients and MSGD with MP (Alg. 2) converges as*

$$
\frac{1}{K\tau} \sum_{t=0}^{K\tau-1} \|\nabla f(\boldsymbol{x}^{(t)})\|_2^2 \leq \frac{16\Delta}{\underline{\delta}\eta K\tau} \tag{28}
$$

*for any $K \geq 1$, where $\Delta = f(\boldsymbol{x}^{(0)}) - \inf_{\boldsymbol{x}} f(\boldsymbol{x})$.*

*Proof.* By Lemma 4 we have

$$
\sum_{t=0}^{K\tau-1} \|\nabla f(\boldsymbol{x}^{(t)})\|_2^2 \leq \frac{2[f(\boldsymbol{x}^{(0)}) - f(\boldsymbol{x}^{(K\tau)})]}{\eta} + \sum_{t=0}^{K\tau-1} \|\tilde{\boldsymbol{m}}^{(t)} - \nabla f(\boldsymbol{x}^{(t)})\|_2^2
$$

$$
- \left( \frac{1}{\eta^2} - \frac{L}{\eta} \right) \sum_{t=0}^{K\tau-1} \|\boldsymbol{x}^{(t+1)} - \boldsymbol{x}^{(t)}\|_2^2. \tag{29}
$$

Applying Lemma 7 to (29) and using $\underline{\delta} \leq \overline{\delta} < 1$ yields

$$
\left( \frac{\delta}{4} - \frac{8}{3\tau\beta_1} \right) \sum_{t=0}^{K\tau-1} \|\nabla f(\boldsymbol{x}^{(t)})\|_2^2
$$

$$
\leq \frac{2}{\eta} f(\boldsymbol{x}^{(0)}) - f(\boldsymbol{x}^{(K\tau)})
$$

$$
- \left( \frac{1}{\eta^2} - \frac{L}{\eta} - \frac{20(1-\beta_1)L^2}{3\underline{\delta}\beta_1^2} - \frac{20\tau(\tau-1)L^2}{3\underline{\delta}} - \frac{4(\tau-1)L^2}{3\beta_1} \right) \sum_{t=0}^{K\tau-1} \|\boldsymbol{x}^{(t+1)} - \boldsymbol{x}^{(t)}\|_2^2. \tag{30}
$$

By (27) we have

$$
\frac{\delta}{4} - \frac{8}{3\tau\beta_1} \geq \frac{\delta}{8}, \quad \text{and} \quad \frac{1}{4\eta^2} \geq \max\left\{ \frac{L}{\eta}, \frac{20(1-\beta_1)L^2}{3\underline{\delta}\beta_1^2}, \frac{20\tau(\tau-1)L^2}{3\underline{\delta}}, \frac{4(\tau-1)L^2}{3\beta_1} \right\}. \tag{31}
$$

Applying (31) to (30) yields (28). $\qquad\square$

We now prove Theorem 2, which is restated as follows.

**Corollary 1** (Convergence complexity of deterministic GaLore). *Under Assumptions 1-2, if $T \geq 64/(3\underline{\delta})$ and we choose*

$$
\beta_1 = 1
$$

$$
\tau = \left\lceil \frac{64}{3\underline{\delta}\beta_1} \right\rceil
$$

$$
\eta = \left( 4L + \sqrt{\frac{80L^2}{3\underline{\delta}\beta_1^2}} + \sqrt{\frac{80\tau^2 L^2}{3\underline{\delta}}} + \sqrt{\frac{16\tau L^2}{3\beta_1}} \right)^{-1},
$$

*GaLore using deterministic gradients and MSGD with MP (Alg. 2) converges as*

$$\frac{1}{T} \sum_{t=0}^{T-1} \|\nabla f(\boldsymbol{x}^{(t)})\|_2^2 = \mathcal{O}\left(\frac{L\Delta}{\underline{\delta}^{5/2}T}\right), \tag{32}$$

*where $\Delta = f(\boldsymbol{x}^{(0)}) - \inf_{\boldsymbol{x}} f(\boldsymbol{x})$. Consequently, the computation complexity to reach an $\varepsilon$-accurate solution $\boldsymbol{x}$ such that $\|\nabla f(\boldsymbol{x})\|_2^2 \leq \varepsilon$ is $\mathcal{O}\left(\frac{L\Delta}{\underline{\delta}^{5/2}\varepsilon} + \frac{1}{\underline{\delta}}\right)$.*

*Proof.* $T \geq 1 + 64/(3\underline{\delta})$ guarantees $T \geq \tau$. Let $T = K\tau + r$, where $K \in \mathbb{N}^*$ and $0 \leq r < \tau$. If $r = 0$, (32) is a direct result of Theorem 6. If $r > 0$, applying Theorem 6 to $\tilde{K} := K + 1$ yields

$$\frac{1}{T} \sum_{t=0}^{T-1} \|\nabla f(\boldsymbol{x}^{(t)})\|_2^2 \leq \frac{\tilde{K}\tau}{T} \cdot \frac{1}{\tilde{K}\tau} \sum_{t=0}^{\tilde{K}\tau-1} \|\nabla f(\boldsymbol{x}^{(t)})\|_2^2 = \mathcal{O}\left(\frac{L\Delta}{\underline{\delta}^{5/2}T}\right).$$

$\square$

### B.4 CONVERGENCE OF LARGE-BATCH GALORE

In this subsection, we present the proof for Theorem 3. GaLore using large-batch stochastic gradients and MSGD with MP is specified as Alg. 3.

**Lemma 8** (Momentum contraction). *Under Assumption 3, in large-batch GaLore using MSGD with MP (Alg. 3), if $0 < \beta_1 \leq 1$, term $\tilde{\boldsymbol{M}}_\ell^{(t)}$ has the following contraction properties:*

- *When $t = 0$, it holds that*

$$\mathbb{E}[\|\tilde{\boldsymbol{M}}_\ell^{(0)} - \nabla_\ell f(\boldsymbol{X}^{(0)})\|_F^2] \leq 2(\tau-1)(1-\delta_\ell\beta_1)\sum_{r=0}^{\tau-2}\mathbb{E}[\|\nabla_\ell f(\boldsymbol{x}^{(r+1)}) - \nabla_\ell f(\boldsymbol{x}^{(r)})\|_F^2]$$

$$+ \frac{4(1-\delta_\ell\beta_1)}{\tau}\sum_{r=0}^{\tau-1}\mathbb{E}[\|\nabla_\ell f(\boldsymbol{x}^{(r)})\|_F^2] + \frac{4\beta_1\sigma_\ell^2}{\mathcal{B}}; \tag{33}$$

- *When $t = k\tau$, $k \in \mathbb{N}^*$, it holds that*

$$\mathbb{E}[\|\tilde{\boldsymbol{M}}_\ell^{(t)} - \nabla_\ell f(\boldsymbol{x}^{(t)})\|_F^2] - \left(1 - \left(1 - \frac{\delta_\ell}{4}\right)\beta_1\right)\mathbb{E}[\|\tilde{\boldsymbol{M}}_\ell^{(t-1)} - \nabla_\ell f(\boldsymbol{x}^{(t-1)})\|_F^2]$$

$$\leq \frac{4(1-\delta_\ell)}{\tau}\sum_{r=0}^{\tau-1}\mathbb{E}[\|\nabla_l f(\boldsymbol{x}^{(k\tau+r)})\|_F^2] + \frac{5(1-\beta_1)}{\delta_\ell\beta_1}\mathbb{E}[\|\nabla_\ell f(\boldsymbol{x}^{(t)}) - \nabla_\ell f(\boldsymbol{x}^{(t-1)})\|_F^2]$$

$$+ 2(\tau-1)(1-\delta_\ell)\sum_{r=0}^{\tau-2}\mathbb{E}[\|\nabla_\ell f(\boldsymbol{x}^{(k\tau+r+1)}) - \nabla_\ell f(\boldsymbol{x}^{(k\tau+r)})\|_F^2] + \frac{5\sigma_\ell^2}{\mathcal{B}}; \tag{34}$$

- *When $t = k\tau + r$, $k \in \mathbb{N}$, $1 \leq r < \tau$, it holds that*

$$\mathbb{E}[\|\tilde{\boldsymbol{M}}_\ell^{(t)} - \nabla_\ell f(\boldsymbol{x}^{(t)})\|_F^2] - \left(1 - \left(1 - \frac{\delta_\ell}{4}\right)\beta_1\right)\mathbb{E}[\|\tilde{\boldsymbol{M}}_\ell^{(t-1)} - \nabla_\ell f(\boldsymbol{x}^{(t-1)})\|_F^2]$$

$$\leq \left(1 - \frac{\delta_\ell}{2}\right)\beta_1\mathbb{E}[\|\nabla_\ell f(\boldsymbol{x}^{(t)})\|_F^2] + \frac{5(1-\beta_1)}{\delta_\ell\beta_1}\mathbb{E}[\|\nabla_\ell f(\boldsymbol{x}^{(t)}) - \nabla_\ell f(\boldsymbol{x}^{(t-1)})\|_F^2]$$

$$+ \frac{15r\beta_1}{\delta_\ell}\sum_{i=1}^{r}\mathbb{E}[\|\nabla_\ell f(\boldsymbol{x}^{(k\tau+i)}) - \nabla_\ell f(\boldsymbol{x}^{(k\tau+i-1)})\|_F^2] + \left(\frac{11\beta_1}{\delta_\ell\mathcal{B}} + \beta_1^2\right)\sigma_\ell^2. \tag{35}$$

---

**Algorithm 3** GaLore using large-batch stochastic gradients and MSGD with MP

---

**Input:** Initial point $\boldsymbol{x}^{(0)}$, data distribution $\mathcal{D}$, learning rate $\eta$, subspace changing frequency $\tau$, rank $\{r_\ell\}_{\ell=1}^{N_L}$, momentum parameter $\beta_1$, large batch size $\mathcal{B}$.
**Output:** $\{\boldsymbol{x}^{(t)}\}_{t=0}^T$.
  Initialize optimizer state $\{\boldsymbol{M}_\ell^{(-1)}\}_{\ell=1}^{N_L}$ to zero;
  **for** $t = 0, 1, \cdots, T-1$ **do**
    **if** $t \equiv 0 \pmod{\tau}$ **then**
      Sample $\{\xi^{(t,b)}\}_{b=1}^{\mathcal{B}} \overset{\text{i.i.d.}}{\sim} \mathcal{D}$;
    **else**
      Sample $\xi^{(t)} \sim \mathcal{D}$;
    **end if**
    **for** $\ell = 1, 2, \cdots, N_L$ **do**
      **if** $t \equiv 0 \pmod{\tau}$ **then**
        $\boldsymbol{G}_\ell^{(t)} = \frac{1}{\mathcal{B}} \sum_{b=1}^{\mathcal{B}} \nabla_\ell F(\boldsymbol{x}^{(t)}; \xi^{(t,b)})$;
        $\boldsymbol{U}, \boldsymbol{\Sigma}, \boldsymbol{V} \leftarrow \text{SVD}(\boldsymbol{G}_\ell^{(t)})$;
        **if** $m_\ell \leq n_\ell$ **then**
          $\boldsymbol{P}_\ell^{(t)} \leftarrow \boldsymbol{U}[:, :r_\ell]$;
          $\boldsymbol{M}_\ell^{(t)} \leftarrow (1 - \beta_1)(\boldsymbol{P}_\ell^{(t)})^\top \boldsymbol{P}_\ell^{(t-1)} \boldsymbol{M}_\ell^{(t-1)} + \beta_1 (\boldsymbol{P}_\ell^{(t)})^\top \boldsymbol{G}_\ell^{(t)}$;
          $\boldsymbol{X}_\ell^{(t+1)} \leftarrow \boldsymbol{X}_\ell^{(t)} - \eta \boldsymbol{P}_\ell^{(t)} \boldsymbol{M}_\ell^{(t)}$;
        **else**
          $\boldsymbol{Q}_\ell^{(t)} \leftarrow \boldsymbol{V}[:, :r_\ell]$;
          $\boldsymbol{M}_\ell^{(t)} \leftarrow (1 - \beta_1) \boldsymbol{M}_\ell^{(t-1)} (\boldsymbol{Q}_\ell^{(t-1)})^\top \boldsymbol{Q}_\ell^{(t)} + \beta_1 \boldsymbol{G}_\ell^{(t)} \boldsymbol{Q}_\ell^{(t)}$;
          $\boldsymbol{X}_\ell^{(t+1)} \leftarrow \boldsymbol{X}_\ell^{(t)} - \eta \boldsymbol{M}_\ell^{(t)} (\boldsymbol{Q}_\ell^{(t)})^\top$;
        **end if**
      **else**
        $\boldsymbol{G}_\ell^{(t)} = \nabla_\ell F(\boldsymbol{x}^{(t)}; \xi^{(t)})$;
        **if** $m_\ell \leq n_\ell$ **then**
          $\boldsymbol{P}_\ell^{(t)} \leftarrow \boldsymbol{P}_\ell^{(t-1)}$;
          $\boldsymbol{M}_\ell^{(t)} \leftarrow (1 - \beta_1) \boldsymbol{M}_\ell^{(t-1)} + \beta_1 (\boldsymbol{P}_\ell^{(t)})^\top \boldsymbol{G}_\ell^{(t)}$;
          $\boldsymbol{X}_\ell^{(t+1)} \leftarrow \boldsymbol{X}_\ell^{(t)} - \eta \boldsymbol{P}_\ell^{(t)} \boldsymbol{M}_\ell^{(t)}$;
        **else**
          $\boldsymbol{Q}_\ell^{(t)} \leftarrow \boldsymbol{Q}_\ell^{(t-1)}$;
          $\boldsymbol{M}_\ell^{(t)} \leftarrow (1 - \beta_1) \boldsymbol{M}_\ell^{(t-1)} + \beta_1 \boldsymbol{G}_\ell^{(t)} \boldsymbol{Q}_\ell^{(t)}$;
          $\boldsymbol{X}_\ell^{(t+1)} \leftarrow \boldsymbol{X}_\ell^{(t)} - \eta \boldsymbol{M}_\ell^{(t)} (\boldsymbol{Q}_\ell^{(t)})^\top$;
        **end if**
      **end if**
    **end for**
  **end for**

---

*Proof.* Without loss of generality assume $m_\ell \leq n_\ell$ (the other case can be proved similarly). When $t = 0$, we have

$$\mathbb{E}[\|\tilde{\boldsymbol{M}}_\ell^{(0)} - \nabla_\ell f(\boldsymbol{x}^{(0)})\|_F^2]$$

$$= \mathbb{E}[\|\beta_1 \boldsymbol{P}_\ell^{(0)} (\boldsymbol{P}_\ell^{(0)})^\top \boldsymbol{G}_\ell^{(0)} - \nabla_\ell f(\boldsymbol{x}^{(0)})\|_F^2]$$

$$= \mathbb{E}[\|\beta_1 (\boldsymbol{P}_\ell^{(0)} (\boldsymbol{P}_\ell^{(0)})^\top - \boldsymbol{I}) \boldsymbol{G}_\ell^{(0)} + \beta_1 (\boldsymbol{G}_\ell^{(0)} - \nabla_\ell f(\boldsymbol{x}^{(0)})) - (1 - \beta_1) \nabla_\ell f(\boldsymbol{x}^{(0)})\|_F^2]$$

$$\leq \beta_1 \mathbb{E}[\|(\boldsymbol{P}_\ell^{(0)} (\boldsymbol{P}_\ell^{(0)})^\top - \boldsymbol{I}) \boldsymbol{G}_\ell^{(0)} + \boldsymbol{G}_\ell^{(0)} - \nabla_\ell f(\boldsymbol{x}^{(0)})\|_F^2] + (1 - \beta_1) \|\nabla_\ell f(\boldsymbol{x}^{(0)})\|_F^2, \qquad (36)$$

where the inequality uses Jensen's inequality. For the first term we have

$$\mathbb{E}[\|(\boldsymbol{P}_\ell^{(0)}(\boldsymbol{P}_\ell^{(0)})^\top - \boldsymbol{I})\boldsymbol{G}_\ell^{(0)} + \boldsymbol{G}_\ell^{(0)} - \nabla_\ell f(\boldsymbol{x}^{(0)})\|_F^2]$$

$$\leq 2\mathbb{E}[\|(\boldsymbol{I} - \boldsymbol{P}_\ell^{(0)}(\boldsymbol{P}_\ell^{(0)})^\top)\boldsymbol{G}_\ell^{(0)}\|_F^2] + 2\mathbb{E}[\|\boldsymbol{G}_\ell^{(0)} - \nabla_\ell f(\boldsymbol{x}^{(0)})\|_F^2]$$

$$\leq 2(1 - \delta_\ell)\mathbb{E}[\|\boldsymbol{G}_\ell\|_F^2] + 2\mathbb{E}[\|\boldsymbol{G}_\ell^{(0)} - \nabla_\ell f(\boldsymbol{x}^{(0)})\|_F^2]$$

$$\leq 2(1 - \delta_\ell)\|\nabla_\ell f(\boldsymbol{x}^{(0)})\|_F^2 + \frac{(4 - 2\delta_\ell)\sigma_\ell^2}{\mathcal{B}}, \tag{37}$$

where the first inequality uses Cauchy's inequality, the second inequality uses Lemma 1, the third inequality uses $\mathbb{E}[\|\boldsymbol{G}_\ell^{(0)} - \nabla_\ell f(\boldsymbol{x}^{(0)})\|_F^2] \leq \sigma_\ell^2/\mathcal{B}$ (Assumption 3). Applying (37) and Lemma 2 to (36) yields (33).

When $t = k\tau$, $k \in \mathbb{N}^*$, we have

$$\mathbb{E}[\|\tilde{\boldsymbol{M}}_\ell^{(t)} - \nabla_\ell f(\boldsymbol{x}^{(t)})\|_F^2]$$

$$=\mathbb{E}[\|\boldsymbol{P}_\ell^{(t)}(\boldsymbol{P}_\ell^{(t)})^\top[(1 - \beta_1)\tilde{\boldsymbol{M}}_\ell^{(t-1)} + \beta_1\boldsymbol{G}_\ell^{(t)} - \nabla_\ell f(\boldsymbol{x}^{(t)})] - (\boldsymbol{I} - \boldsymbol{P}_\ell^{(t)}(\boldsymbol{P}_\ell^{(t)})^\top)\nabla_\ell f(\boldsymbol{x}^{(t)})\|_F^2]$$

$$=\mathbb{E}[\|\boldsymbol{P}_\ell^{(t)}(\boldsymbol{P}_\ell^{(t)})^\top[(1 - \beta_1)\tilde{\boldsymbol{M}}_\ell^{(t-1)} + \beta_1\boldsymbol{G}_\ell^{(t)} - \nabla_\ell f(\boldsymbol{x}^{(t)})]\|_F^2]$$

$$+ \mathbb{E}[\|(\boldsymbol{I} - \boldsymbol{P}_\ell^{(t)}(\boldsymbol{P}_\ell^{(t)})^\top)\nabla_\ell f(\boldsymbol{x}^{(t)})\|_F^2], \tag{38}$$

where the second equality uses Lemma 3. By $\|\boldsymbol{P}_\ell^{(t)}(\boldsymbol{P}_\ell^{(t)})^\top\|_2 = 1$, we have

$$\mathbb{E}[\|\boldsymbol{P}_\ell^{(t)}(\boldsymbol{P}_\ell^{(t)})^\top[(1 - \beta_1)\tilde{\boldsymbol{M}}_\ell^{(t-1)} + \beta_1\boldsymbol{G}_\ell^{(t)} - \nabla_\ell f(\boldsymbol{x}^{(t)})]\|_F^2]$$

$$\leq\mathbb{E}[\|(1 - \beta_1)\tilde{\boldsymbol{M}}_\ell^{(t-1)} + \beta_1\boldsymbol{G}_\ell^{(t)} - \nabla_\ell f(\boldsymbol{x}^{(t)})\|_F^2]$$

$$=\mathbb{E}[\|(1 - \beta_1)(\tilde{\boldsymbol{M}}_\ell^{(t-1)} - \nabla_\ell f(\boldsymbol{x}^{(t)})) + \beta_1(\boldsymbol{G}_\ell^{(t)} - \nabla_\ell f(\boldsymbol{x}^{(t)}))\|_F^2]$$

$$\leq\mathbb{E}[\|(1 - \beta_1)(\tilde{\boldsymbol{M}}_\ell^{(t-1)} - \nabla_\ell f(\boldsymbol{x}^{(t)}))\|_F^2] + \beta_1^2\mathbb{E}[\|\boldsymbol{G}_\ell^{(t)} - \nabla_\ell f(\boldsymbol{x}^{(t)})\|_F^2], \tag{39}$$

where the last inequality uses the unbiasedness of $\boldsymbol{G}_\ell^{(t)}$ (Assumption 3). By Young's inequality, we have

$$\mathbb{E}[\|\tilde{\boldsymbol{M}}_\ell^{(t-1)} - \nabla_\ell f(\boldsymbol{x}^{(t)})\|_F^2]$$

$$=\mathbb{E}[\|(\tilde{\boldsymbol{M}}_\ell^{(t-1)} - \nabla_\ell f(\boldsymbol{x}^{(t-1)})) - (\nabla_\ell f(\boldsymbol{x}^{(t)}) - \nabla_\ell f(\boldsymbol{x}^{(t-1)})\|_F^2]$$

$$\leq \left(1 + \frac{\delta_\ell\beta_1}{4}\right)\mathbb{E}[\|\tilde{\boldsymbol{M}}_\ell^{(t-1)} - \nabla_\ell f(\boldsymbol{x}^{(t-1)})\|_F^2] + \left(1 + \frac{4}{\delta_\ell\beta_1}\right)\mathbb{E}[\|\nabla_\ell f(\boldsymbol{x}^{(t)}) - \nabla_\ell f(\boldsymbol{x}^{(t-1)})\|_F^2]. \tag{40}$$

Applying (40) to (39) yields

$$\mathbb{E}[\|\boldsymbol{P}_\ell^{(t)}(\boldsymbol{P}_\ell^{(t)})^\top[(1 - \beta_1)\tilde{\boldsymbol{M}}_\ell^{(t-1)} + \beta_1\boldsymbol{G}_\ell^{(t)} - \nabla_\ell f(\boldsymbol{x}^{(t)})]\|_F^2]$$

$$\leq \left(1 - \left(1 - \frac{\delta_\ell}{4}\right)\beta_1\right)\mathbb{E}[\|\tilde{\boldsymbol{M}}_\ell^{(t-1)} - \nabla_\ell f(\boldsymbol{x}^{(t-1)})\|_F^2] + \frac{\beta_1^2\sigma^2}{\mathcal{B}}$$

$$+ \frac{5(1 - \beta_1)}{\delta_\ell\beta_1}\mathbb{E}[\|\nabla_\ell f(\boldsymbol{x}^{(t)}) - \nabla_\ell f(\boldsymbol{x}^{(t-1)})\|_F^2]. \tag{41}$$

For the second term in (38), we have

$$\mathbb{E}[\|(\boldsymbol{I} - \boldsymbol{P}_\ell^{(t)}(\boldsymbol{P}_\ell^{(t)})^\top)\nabla_\ell f(\boldsymbol{x}^{(t)})\|_F^2]$$

$$\leq 2\mathbb{E}[\|(\boldsymbol{I} - \boldsymbol{P}_\ell^{(t)}(\boldsymbol{P}_\ell^{(t)})^\top)\boldsymbol{G}_\ell^{(t)}\|_F^2] + 2\mathbb{E}[\|(\boldsymbol{I} - \boldsymbol{P}_\ell^{(t)}(\boldsymbol{P}_\ell^{(t)})^\top)(\boldsymbol{G}_\ell^{(t)} - \nabla_\ell f(\boldsymbol{x}^{(t)}))\|_F^2]$$

$$\leq 2(1 - \delta_\ell)\mathbb{E}[\|\boldsymbol{G}_\ell^{(t)}\|_F^2] + 2\mathbb{E}[\|\boldsymbol{G}_\ell^{(t)} - \nabla_\ell f(\boldsymbol{x}^{(t)})\|_F^2]$$

$$\leq 2(1 - \delta_\ell)\mathbb{E}[\|\nabla_\ell f(\boldsymbol{x}^{(t)})\|_F^2] + \frac{4\sigma_\ell^2}{\mathcal{B}}, \tag{42}$$

where the first inequality uses Cauchy's inequality, the second inequality uses Lemma 1 and $\|\boldsymbol{I} - \boldsymbol{P}_\ell^{(t)}(\boldsymbol{P}_\ell^{(t)})^\top\|_2 = 1$, the third inequality uses Assumption 3. Applying (41)(42) to (38) and using Lemma 2 yields (34).

When $t = k\tau + r$, $k \in \mathbb{N}$, $1 \le r < \tau$, we have

$$\mathbb{E}[\|\tilde{\boldsymbol{M}}_\ell^{(t)} - \nabla_\ell f(\boldsymbol{x}^{(t)})\|_F^2]$$

$$= \mathbb{E}[\|(1 - \beta_1)(\tilde{\boldsymbol{M}}_\ell^{(t-1)} - \nabla_\ell f(\boldsymbol{x}^{(t)})) + \beta_1 (\boldsymbol{P}_\ell^{(t)}(\boldsymbol{P}_\ell^{(t)})^\top \boldsymbol{G}_\ell^{(t)} - \nabla_\ell f(\boldsymbol{x}^{(t)}))\|_F^2]$$

$$= \mathbb{E}[\|(1 - \beta_1)(\tilde{\boldsymbol{M}}_\ell^{(t-1)} - \nabla_\ell f(\boldsymbol{x}^{(t)})) + \beta_1 (\boldsymbol{P}_\ell^{(t)}(\boldsymbol{P}_\ell^{(t)})^\top - \boldsymbol{I})\nabla_\ell f(\boldsymbol{x}^{(t)})\|_F^2]$$

$$+ \beta_1^2 \mathbb{E}[\|\boldsymbol{P}_\ell^{(t)}(\boldsymbol{P}_\ell^{(t)})^\top (\boldsymbol{G}_\ell^{(t)} - \nabla_\ell f(\boldsymbol{x}^{(t)}))\|_F^2]$$

$$\le (1 - \beta_1)\mathbb{E}[\|\tilde{\boldsymbol{M}}_\ell^{(t-1)} - \nabla_\ell f(\boldsymbol{x}^{(t)})\|_F^2] + \beta_1 \mathbb{E}[\|(\boldsymbol{I} - \boldsymbol{P}_\ell^{(t)}(\boldsymbol{P}_\ell^{(t)})^\top)\nabla_\ell f(\boldsymbol{x}^{(t)})\|_F^2$$

$$+ \beta_1^2 \mathbb{E}[\|\boldsymbol{P}_\ell^{(t)}(\boldsymbol{P}_\ell^{(t)})^\top (\boldsymbol{G}_\ell^{(t)} - \nabla_\ell f(\boldsymbol{x}^{(t)}))\|_F^2], \tag{43}$$

where the second equality uses the unbiasedness of $\boldsymbol{G}_\ell^{(t)}$ and the independence implied by $\boldsymbol{P}_\ell^{(t)} = \boldsymbol{P}_\ell^{(t-1)}$, the inequality uses Jensen's inequality. The first term is similarly bounded as (40). For the second term, we have

$$\mathbb{E}[\|(\boldsymbol{I} - \boldsymbol{P}_\ell^{(k\tau)}(\boldsymbol{P}_\ell^{(k\tau)})^\top)\nabla_\ell f(\boldsymbol{x}^{(t)})\|_F^2]$$

$$\le \left(1 + \frac{\delta_\ell}{4}\right) \mathbb{E}[\|(\boldsymbol{I} - \boldsymbol{P}_\ell^{(k\tau)}(\boldsymbol{P}_\ell^{(k\tau)})^\top)\boldsymbol{G}_\ell^{(k\tau)}\|_F^2]$$

$$+ \left(1 + \frac{4}{\delta_\ell}\right) \mathbb{E}[\|(\boldsymbol{I} - \boldsymbol{P}_\ell^{(k\tau)}(\boldsymbol{P}_\ell^{(k\tau)})^\top)(\nabla_\ell f(\boldsymbol{x}^{(t)}) - \boldsymbol{G}_\ell^{(k\tau)})\|_F^2]$$

$$\le \left(1 - \frac{3\delta_\ell}{4}\right) \mathbb{E}[\|\boldsymbol{G}_\ell^{(k\tau)}\|_F^2] + 2\left(1 + \frac{4}{\delta_\ell}\right) \mathbb{E}[\|\boldsymbol{G}_\ell^{(k\tau)} - \nabla_\ell f(\boldsymbol{x}^{(k\tau)})\|_F^2]$$

$$+ 2\left(1 + \frac{4}{\delta_\ell}\right) \mathbb{E}[\|\nabla_\ell f(\boldsymbol{x}^{(t)}) - \nabla_\ell f(\boldsymbol{x}^{(k\tau)})\|_F^2], \tag{44}$$

where the first inequality uses Young's inequality, the second inequality uses Lemma 1 and Cauchy's inequality. We further have

$$\left(1 - \frac{3\delta_\ell}{4}\right) \mathbb{E}[\|\boldsymbol{G}_\ell^{(k\tau)}\|_F^2] + 2\left(1 + \frac{4}{\delta_\ell}\right) \mathbb{E}[\|\boldsymbol{G}_\ell^{(k\tau)} - \nabla_\ell f(\boldsymbol{x}^{(k\tau)})\|_F^2]$$

$$\le \left(1 - \frac{3\delta_\ell}{4}\right) \mathbb{E}[\|\nabla_\ell f(\boldsymbol{x}^{(k\tau)})\|_F^2] + \frac{11}{\delta_\ell} \mathbb{E}[\|\boldsymbol{G}_\ell^{(k\tau)} - \nabla_\ell f(\boldsymbol{x}^{(k\tau)})\|_F^2]$$

$$\le \left(1 - \frac{3\delta_\ell}{4}\right) \mathbb{E}[\|\nabla_\ell f(\boldsymbol{x}^{(k\tau)})\|_F^2] + \frac{11\sigma_\ell^2}{\delta_\ell \mathcal{B}}$$

$$\le \left(1 - \frac{\delta_\ell}{2}\right) \mathbb{E}[\|\nabla_\ell f(\boldsymbol{x}^{(t)})\|_F^2] + \left(1 + \frac{4}{\delta_\ell}\right) \mathbb{E}[\|\nabla_\ell f(\boldsymbol{x}^{(t)}) - \nabla_\ell f(\boldsymbol{x}^{(k\tau)})\|_F^2] + \frac{11\sigma_\ell^2}{\delta_\ell \mathcal{B}}, \tag{45}$$

where the first inequality uses unbiasedness of $\boldsymbol{G}_\ell^{(k\tau)}$, the second inequality uses Assumption 3, the third inequality uses Young's inequality.

Applying (45) to (44) and applying Cauchy's inequality yields

$$\mathbb{E}[\|(\boldsymbol{I} - \boldsymbol{P}_\ell^{(k\tau)}(\boldsymbol{P}_\ell^{(k\tau)})^\top)\nabla_\ell f(\boldsymbol{x}^{(t)})\|_F^2]$$

$$\le \left(1 - \frac{\delta_\ell}{2}\right) \mathbb{E}[\|\nabla_\ell f(\boldsymbol{x}^{(t)})\|_F^2] + \frac{11\sigma_\ell^2}{\delta_\ell \mathcal{B}} + \frac{15r}{\delta_\ell} \sum_{i=1}^r \mathbb{E}[\|\nabla_\ell f(\boldsymbol{x}^{(k\tau+i)}) - \nabla_\ell f(\boldsymbol{x}^{(k\tau+i-1)})\|_F^2]. \tag{46}$$

For the third term, we have

$$\mathbb{E}[\|\boldsymbol{P}_\ell^{(k\tau)}(\boldsymbol{P}_\ell^{(k\tau)})^\top (\boldsymbol{G}_\ell^{(t)} - \nabla_\ell f(\boldsymbol{x}^{(t)}))\|_F^2] \le \mathbb{E}[\|\boldsymbol{G}_\ell^{(t)} - \nabla_\ell f(\boldsymbol{x}^{(t)})\|_F^2] \le \sigma_\ell^2, \tag{47}$$

where the first inequality uses $\|\boldsymbol{P}_\ell^{(k\tau)}(\boldsymbol{P}_\ell^{(k\tau)})^\top\|_2 = 1$, the second inequality uses Assumption 3.

Applying (40)(46)(47) to (43) yields (35). $\qquad\square$

**Lemma 9** (Momentum error). *Under Assumption 2-3, if $0 < \beta_1 \leq 1$ in large-batch GaLore using MSGD and MP (Alg. 3), it holds for any $K \geq 1$ that*

$$\sum_{t=0}^{K\tau-1} \mathbb{E}[\|\tilde{\boldsymbol{m}}^{(t)} - \nabla f(\boldsymbol{x}^{(t)})\|_2^2]$$

$$\leq \left( \frac{5(1-\beta_1)}{(1-\underline{\delta}/4)\underline{\delta}\beta_1^2} + \frac{15\tau(\tau-1)}{2(1-\underline{\delta}/4)\underline{\delta}} + \frac{2(\tau-1)}{(1-\overline{\delta}/4)\beta_1} \right) L^2 \sum_{t=0}^{K\tau-2} \mathbb{E}[\|\boldsymbol{x}^{(t+1)} - \boldsymbol{x}^{(t)}\|_2^2]$$

$$+ \left( \frac{1-\underline{\delta}/2}{1-\underline{\delta}/4} + \frac{4}{(1-\overline{\delta}/4)\tau\beta_1} \right) \sum_{t=0}^{K\tau-1} \mathbb{E}[\|\nabla f(\boldsymbol{x}^{(t)})\|_2^2]$$

$$+ \left( \frac{5K}{(1-\overline{\delta}/4)\beta_1\mathcal{B}} + \frac{11K\tau}{(1-\underline{\delta}/4)\underline{\delta}\mathcal{B}} + \frac{K\tau\beta_1}{1-\overline{\delta}/4} \right) \sigma^2. \tag{48}$$

*Proof.* By Lemma 8 we have

$$\sum_{t=0}^{K\tau-1} \mathbb{E}[\|\tilde{\boldsymbol{M}}_\ell^{(t)} - \nabla_\ell f(\boldsymbol{x}^{(t)})\|_F^2] - \left( 1 - \left( 1 - \frac{\delta_\ell}{4} \right) \beta_1 \right) \sum_{t=0}^{K\tau-2} \mathbb{E}[\|\tilde{\boldsymbol{M}}_\ell^{(t)} - \nabla_\ell f(\boldsymbol{x}^{(t)})\|_F^2]$$

$$\leq \left( \frac{5(1-\beta_1)}{\delta_\ell\beta_1} + \frac{15\tau(\tau-1)\beta_1}{2\delta_\ell} + 2(\tau-1) \right) \sum_{t=0}^{K\tau-2} \mathbb{E}[\|\nabla_\ell f(\boldsymbol{x}^{(t+1)}) - \nabla_\ell f(\boldsymbol{x}^{(t)})\|_F^2]$$

$$+ \left( \frac{4}{\tau} + \left( 1 - \frac{\delta_\ell}{2} \right) \beta_1 \right) \sum_{t=0}^{K\tau-1} \mathbb{E}[\|\nabla_\ell f(\boldsymbol{x}^{(t)})\|_F^2] + \left( \frac{5K}{\mathcal{B}} + \frac{11K\tau\beta_1}{\delta_\ell\mathcal{B}} + K\tau\beta_1^2 \right) \sigma_\ell^2,$$

which implies

$$\sum_{t=0}^{K\tau-1} \mathbb{E}[\|\tilde{\boldsymbol{M}}_\ell^{(t)} - \nabla_\ell f(\boldsymbol{x}^{(t)})\|_F^2]$$

$$\leq \left( \frac{5(1-\beta_1)}{(1-\delta_\ell/4)\delta_\ell\beta_1^2} + \frac{15\tau(\tau-1)}{2(1-\delta_\ell/4)\delta_\ell} + \frac{2(\tau-1)}{(1-\delta_\ell/4)\beta_1} \right) \sum_{t=0}^{K\tau-2} \mathbb{E}[\|\nabla_\ell f(\boldsymbol{x}^{(t+1)}) - \nabla_\ell f(\boldsymbol{x}^{(t)})\|_F^2]$$

$$+ \left( \frac{1-\delta_\ell/2}{1-\delta_\ell/4} + \frac{4}{(1-\delta_\ell/4)\tau\beta_1} \right) \sum_{t=0}^{K\tau-1} \mathbb{E}[\|\nabla_\ell f(\boldsymbol{x}^{(t)})\|_F^2]$$

$$+ \left( \frac{5K}{(1-\delta_\ell/4)\beta_1\mathcal{B}} + \frac{11K\tau}{(1-\delta_\ell/4)\delta_\ell\mathcal{B}} + \frac{K\tau\beta_1}{1-\delta_\ell/4} \right) \sigma_\ell^2. \tag{49}$$

Summing (49) for $\ell = 1, \cdots, N_L$ and applying Assumption 2-3 yields (48). $\square$

Now we are ready to prove the convergence of Alg. 3.

**Theorem 7** (Convergence of large-batch GaLore). *Under Assumptions 1-3, if hyperparameters*

$$0 < \beta_1 \leq 1, \quad \tau \geq \frac{128}{3\beta_1\underline{\delta}}, \quad 0 < \eta \leq \min\left\{ \frac{1}{4L}, \sqrt{\frac{3\underline{\delta}\beta_1^2}{80L^2}}, \sqrt{\frac{\underline{\delta}}{40\tau^2 L^2}}, \sqrt{\frac{3\beta_1}{32\tau L^2}} \right\}, \tag{50}$$

*GaLore using large-batch stochastic gradients and MSGD with MP (Alg. 3) converges as*

$$\frac{1}{K\tau} \sum_{t=0}^{K\tau-1} \mathbb{E}\|\nabla f(\boldsymbol{x}^{(t)})\|_2^2 \leq \frac{16\Delta}{\underline{\delta}\eta K\tau} + \left( \frac{160}{3\beta_1\underline{\delta}\tau\mathcal{B}} + \frac{352}{3\underline{\delta}^2\mathcal{B}} + \frac{32\beta_1}{3\underline{\delta}} \right) \sigma^2 \tag{51}$$

*for any $K \geq 1$, where $\Delta = f(\boldsymbol{x}^{(0)}) - \inf_{\boldsymbol{x}} f(\boldsymbol{x})$.*

*Proof.* By Lemma 4 we have

$$\sum_{t=0}^{K\tau-1} \mathbb{E}[\|\nabla f(\boldsymbol{x}^{(t)})\|_2^2] \leq \frac{2[f(\boldsymbol{x}^{(0)}) - \mathbb{E}[f(\boldsymbol{x}^{(K\tau)})]]}{\eta} + \sum_{t=0}^{K\tau-1} \mathbb{E}[\|\tilde{\boldsymbol{m}}^{(t)} - \nabla f(\boldsymbol{x}^{(t)})\|_2^2]$$

$$- \left(\frac{1}{\eta^2} - \frac{L}{\eta}\right) \sum_{t=0}^{K\tau-1} \mathbb{E}[\|\boldsymbol{x}^{(t+1)} - \boldsymbol{x}^{(t)}\|_2^2]. \tag{52}$$

Applying Lemma 9 to (52) and using $\underline{\delta} \leq \overline{\delta} < 1$ yields

$$\left(\frac{\delta}{4} - \frac{16}{3\tau\beta_1}\right) \sum_{t=0}^{K\tau-1} \mathbb{E}[\|\nabla f(\boldsymbol{x}^{(t)})\|_2^2]$$

$$\leq \frac{2}{\eta} \mathbb{E}[f(\boldsymbol{x}^{(0)}) - f(\boldsymbol{x}^{(K\tau)})] + \left(\frac{20K}{3\beta_1\mathcal{B}} + \frac{44K\tau}{3\underline{\delta}\mathcal{B}} + \frac{4K\tau\beta_1}{3}\right)\sigma^2$$

$$- \left(\frac{1}{\eta^2} - \frac{L}{\eta} - \frac{20(1-\beta_1)L^2}{3\underline{\delta}\beta_1^2} - \frac{10\tau(\tau-1)L^2}{\underline{\delta}} - \frac{8(\tau-1)L^2}{3\beta_1}\right) \sum_{t=0}^{K\tau-1} \mathbb{E}[\|\boldsymbol{x}^{(t+1)} - \boldsymbol{x}^{(t)}\|_2^2]. \tag{53}$$

By (50) we have

$$\frac{\delta}{4} - \frac{16}{3\tau\beta_1} \geq \frac{\delta}{8}, \quad \text{and} \quad \frac{1}{4\eta^2} \geq \max\left\{\frac{L}{\eta}, \frac{20(1-\beta_1)L^2}{3\underline{\delta}\beta_1^2}, \frac{10\tau(\tau-1)L^2}{\underline{\delta}}, \frac{8(\tau-1)L^2}{3\beta_1}\right\}. \tag{54}$$

Applying (54) to (53) yields (51). $\qquad\square$

We now prove Theorem 3, which is restated as follows.

**Corollary 2** (Convergence complexity of large-batch GaLore). *Under Assumptions 1-3, if $T \geq 2 + 256/(3\underline{\delta}) + (256\sigma)^2/(9\sqrt{\underline{\delta}}L\Delta)$ and we choose*

$$\beta_1 = \left(1 + \sqrt{\frac{\underline{\delta}^{3/2}\sigma^2 T}{L\Delta}}\right)^{-1},$$

$$\tau = \left\lceil \frac{128}{3\underline{\delta}\beta_1} \right\rceil,$$

$$\eta = \left(4L + \sqrt{\frac{80L^2}{3\underline{\delta}\beta_1^2}} + \sqrt{\frac{40\tau^2 L^2}{\underline{\delta}}} + \sqrt{\frac{32\tau L^2}{3\beta_1}}\right)^{-1},$$

$$\mathcal{B} = \left\lceil \frac{1}{\underline{\delta}\beta_1} \right\rceil,$$

*GaLore using large-batch stochastic gradients and MSGD with MP (Alg. 3) converges as*

$$\frac{1}{T}\sum_{t=0}^{T-1} \mathbb{E}[\|\nabla f(\boldsymbol{x}^{(t)})\|_2^2] = \mathcal{O}\left(\frac{L\Delta}{\underline{\delta}^{5/2}T} + \sqrt{\frac{L\Delta\sigma^2}{\underline{\delta}^{7/2}T}}\right), \tag{55}$$

*where $\Delta = f(\boldsymbol{x}^{(0)}) - \inf_{\boldsymbol{x}} f(\boldsymbol{x})$. Consequently, the computation complexity to reach an $\varepsilon$-accurate solution $\boldsymbol{x}$ such that $\|\nabla f(\boldsymbol{x})\|_2^2 \leq \varepsilon$ is $\mathcal{O}\left(\frac{L\Delta\sigma^2}{\underline{\delta}^{7/2}\varepsilon^2} + \frac{L\Delta}{\underline{\delta}^{5/2}\varepsilon} + \frac{\sigma^2}{\underline{\delta}^{1/2}L\Delta} + \frac{1}{\underline{\delta}}\right)$.*

*Proof.* $T \geq 2 + 128/(3\underline{\delta}) + (128\sigma)^2/(9\sqrt{\underline{\delta}}L\Delta)$ guarantees $T \geq \tau$. Let $T = K\tau + r$, where $K \in \mathbb{N}^*$ and $0 \leq r < \tau$. If $r = 0$, (55) is a direct result of Theorem 7. If $r > 0$, applying Theorem 7 to $\tilde{K} := K + 1$ yields

$$\frac{1}{T}\sum_{t=0}^{T-1} \mathbb{E}[\|\nabla f(\boldsymbol{x}^{(t)})\|_2^2] \leq \frac{\tilde{K}\tau}{T} \cdot \frac{1}{\tilde{K}\tau}\sum_{t=0}^{\tilde{K}\tau-1} \mathbb{E}[\|\nabla f(\boldsymbol{x}^{(t)})\|_2^2] = \mathcal{O}\left(\frac{L\Delta}{\underline{\delta}^{5/2}T} + \sqrt{\frac{L\Delta\sigma^2}{\underline{\delta}^{7/2}T}}\right).$$

$\qquad\square$

---

**Algorithm 4** GoLore using small-batch stochastic gradients and MSGD with MP

---

**Input:** Initial point $\boldsymbol{x}^{(0)}$, data distribution $\mathcal{D}$, learning rate $\eta$, subspace changing frequency $\tau$, rank $\{r_\ell\}_{\ell=1}^{N_L}$, momentum parameter $\beta_1$.

**Output:** $\{\boldsymbol{x}^{(t)}\}_{t=0}^T$.

  Initialize optimizer state $\{\boldsymbol{M}_\ell^{(-1)}\}_{\ell=1}^{N_L}$ to zero;

  **for** $t = 0, 1, \cdots, T-1$ **do**

    Sample $\xi^{(t)} \sim \mathcal{D}$;

    $\boldsymbol{G}_\ell^{(t)} = \nabla_\ell F(\boldsymbol{x}^{(t)}; \xi^{(t)})$;

    **for** $\ell = 1, 2, \cdots, N_L$ **do**

      **if** $t \equiv 0 \pmod{\tau}$ **then**

        **if** $m_\ell \le n_\ell$ **then**

          Sample $\boldsymbol{P}_\ell^{(t)} \sim \mathcal{U}(\mathrm{St}_{m_\ell, r_\ell})$;

          $\boldsymbol{M}_\ell^{(t)} \leftarrow (1-\beta_1)(\boldsymbol{P}_\ell^{(t)})^\top \boldsymbol{P}_\ell^{(t-1)} \boldsymbol{M}_\ell^{(t-1)} + \beta_1 (\boldsymbol{P}_\ell^{(t)})^\top \boldsymbol{G}_\ell^{(t)}$;

          $\boldsymbol{X}_\ell^{(t+1)} \leftarrow \boldsymbol{X}_\ell^{(t)} - \eta \boldsymbol{P}_\ell^{(t)} \boldsymbol{M}_\ell^{(t)}$;

        **else**

          Sample $\boldsymbol{Q}_\ell^{(t)} \sim \mathcal{U}(\mathrm{St}_{n_\ell, r_\ell})$;

          $\boldsymbol{M}_\ell^{(t)} \leftarrow (1-\beta_1)\boldsymbol{M}_\ell^{(t-1)}(\boldsymbol{Q}_\ell^{(t-1)})^\top \boldsymbol{Q}_\ell^{(t)} + \beta_1 \boldsymbol{G}_\ell^{(t)} \boldsymbol{Q}_\ell^{(t)}$;

          $\boldsymbol{X}_\ell^{(t+1)} \leftarrow \boldsymbol{X}_\ell^{(t)} - \eta \boldsymbol{M}_\ell^{(t)}(\boldsymbol{Q}_\ell^{(t)})^\top$;

        **end if**

      **else**

        **if** $m_\ell \le n_\ell$ **then**

          $\boldsymbol{P}_\ell^{(t)} \leftarrow \boldsymbol{P}_\ell^{(t-1)}$;

          $\boldsymbol{M}_\ell^{(t)} \leftarrow (1-\beta_1)\boldsymbol{M}_\ell^{(t-1)} + \beta_1 (\boldsymbol{P}_\ell^{(t)})^\top \boldsymbol{G}_\ell^{(t)}$;

          $\boldsymbol{X}_\ell^{(t+1)} \leftarrow \boldsymbol{X}_\ell^{(t)} - \eta \boldsymbol{P}_\ell^{(t)} \boldsymbol{M}_\ell^{(t)}$;

        **else**

          $\boldsymbol{Q}_\ell^{(t)} \leftarrow \boldsymbol{Q}_\ell^{(t-1)}$;

          $\boldsymbol{M}_\ell^{(t)} \leftarrow (1-\beta_1)\boldsymbol{M}_\ell^{(t-1)} + \beta_1 \boldsymbol{G}_\ell^{(t)} \boldsymbol{Q}_\ell^{(t)}$;

          $\boldsymbol{X}_\ell^{(t+1)} \leftarrow \boldsymbol{X}_\ell^{(t)} - \eta \boldsymbol{M}_\ell^{(t)}(\boldsymbol{Q}_\ell^{(t)})^\top$;

        **end if**

      **end if**

    **end for**

  **end for**

---

### B.5   Convergence of GoLore

In this subsection, we present the proof for Theorem 4. GoLore using small-batch stochastic gradients and MSGD with MP is specified as Alg. 4.

**Lemma 10** (Momentum contraction). *Under Assumption 3, in large-batch GoLore using MSGD with MP (Alg. 4), if $0 < \beta_1 \le 1$, term $\tilde{\boldsymbol{M}}_\ell^{(t)}$ has the following contraction properties:*

- *When $t = 0$, it holds that*

$$
\mathbb{E}[\|\tilde{\boldsymbol{M}}_\ell^{(0)} - \nabla_\ell f(\boldsymbol{X}^{(0)})\|_F^2] \le (\tau-1)(1-\delta_\ell \beta_1) \sum_{r=0}^{\tau-2} \mathbb{E}[\|\nabla_\ell f(\boldsymbol{x}^{(r+1)}) - \nabla_\ell f(\boldsymbol{x}^{(r)})\|_F^2]
$$

$$
+ \frac{2(1-\delta_\ell \beta_1)}{\tau} \sum_{r=0}^{\tau-1} \mathbb{E}[\|\nabla_\ell f(\boldsymbol{x}^{(r)})\|_F^2] + \delta_\ell \beta_1^2 \sigma_\ell^2; \qquad (56)
$$

- *When $t = k\tau$, $k \in \mathbb{N}^*$, it holds that*

$$\mathbb{E}[\|\tilde{M}_\ell^{(t)} - \nabla_\ell f(x^{(t)})\|_F^2] - \delta_\ell \left(1 - \left(1 - \frac{\delta_\ell}{4}\right)\beta_1\right)\mathbb{E}[\|\tilde{M}_\ell^{(t-1)} - \nabla_\ell f(x^{(t-1)})\|_F^2]$$

$$\leq \frac{2(1-\delta_\ell)}{\tau}\sum_{r=0}^{\tau-1}\mathbb{E}[\|\nabla_l f(x^{(k\tau+r)})\|_F^2] + \frac{5(1-\beta_1)}{\beta_1}\mathbb{E}[\|\nabla_\ell f(x^{(t)}) - \nabla_\ell f(x^{(t-1)})\|_F^2]$$

$$+ (\tau-1)(1-\delta_\ell)\sum_{r=0}^{\tau-2}\mathbb{E}[\|\nabla_\ell f(x^{(k\tau+r+1)}) - \nabla_\ell f(x^{(k\tau+r)})\|_F^2] + \delta_\ell\beta_1^2\sigma_\ell^2; \qquad (57)$$

- *When $t = k\tau + r$, $k \in \mathbb{N}$, $1 \leq r < \tau$, it holds that*

$$\mathbb{E}[\|\tilde{M}_\ell^{(t)} - \nabla_\ell f(x^{(t)})\|_F^2] - \left(1 - \left(1 - \frac{\delta_\ell}{4}\right)\beta_1\right)\mathbb{E}[\|\tilde{M}_\ell^{(t-1)} - \nabla_\ell f(x^{(t-1)})\|_F^2]$$

$$\leq \left(1 - \frac{\delta_\ell}{2}\right)\beta_1\mathbb{E}[\|\nabla_\ell f(x^{(t)})\|_F^2] + \frac{5(1-\beta_1)}{\delta_\ell\beta_1}\mathbb{E}[\|\nabla_\ell f(x^{(t)}) - \nabla_\ell f(x^{(t-1)})\|_F^2]$$

$$+ \frac{10r\beta_1}{\delta_\ell}\sum_{i=1}^{r}\mathbb{E}[\|\nabla_\ell f(x^{(k\tau+i)}) - \nabla_\ell f(x^{(k\tau+i-1)})\|_F^2] + \beta_1^2\sigma_\ell^2. \qquad (58)$$

*Proof.* Without loss of generality assume $m_\ell \leq n_\ell$ (the other case can be proved similarly). When $t = 0$, we have

$$\mathbb{E}[\|\tilde{M}_\ell^{(0)} - \nabla_\ell f(x^{(0)})\|_F^2]$$

$$= \mathbb{E}[\|\beta_1 P_\ell^{(0)}(P_\ell^{(0)})^\top G_\ell^{(0)} - \nabla_\ell f(x^{(0)})\|_F^2]$$

$$= \mathbb{E}[\|(\beta_1 P_\ell^{(0)}(P_\ell^{(0)})^\top - I)\nabla_\ell f(x^{(0)})\|_F^2] + \beta_1^2\mathbb{E}[\|P_\ell^{(0)}(P_\ell^{(0)})^\top (G_\ell^{(0)} - \nabla_\ell f(x^{(0)}))\|_F^2]$$

$$= \text{tr}((\nabla_\ell f(x^{(0)}))^\top \mathbb{E}[(\beta_1 P_\ell^{(0)}(P_\ell^{(0)})^\top - I)^2]\nabla_\ell f(x^{(0)}))$$

$$+ \beta_1^2 \text{tr}(\mathbb{E}_{\xi^{(0)}\sim\mathcal{D}}[(G_\ell^{(0)} - \nabla_\ell f(x^{(0)}))^\top \mathbb{E}_{P\sim\mathcal{U}(\text{St}_{m_\ell,r_\ell})}[(PP^\top)^2](G_\ell^{(0)} - \nabla_\ell f(x^{(0)}))]), \quad (59)$$

where the second equality uses unbiasedness of $G_\ell^{(0)}$. By Lemma 5 we have

$$\mathbb{E}[(\beta P_\ell^{(0)}(P_\ell^{(0)})^\top - I)^2] = I - (2\beta_1 - \beta_1^2)\mathbb{E}[P_\ell^{(0)}(P_\ell^{(0)})^\top]$$

$$= I - (2\beta_1 - \beta_1^2)\delta_\ell I,$$

thus

$$\text{tr}((\nabla_\ell f(x^{(0)}))^\top \mathbb{E}[(\beta_1 P_\ell^{(0)}(P_\ell^{(0)})^\top - I)^2]\nabla_\ell f(x^{(0)})) = (1 - \delta_\ell(2\beta_1 - \beta_1^2))\|\nabla_\ell f(x^{(0)})\|_F^2$$

$$\leq (1 - \delta_\ell\beta_1)\|\nabla_\ell f(x^{(0)})\|_F^2. \qquad (60)$$

Similarly, by Lemma 5 we have

$$\text{tr}(\mathbb{E}_{\xi^{(0)}\sim\mathcal{D}}[(G_\ell^{(0)} - \nabla_\ell f(x^{(0)}))^\top \mathbb{E}_{P\sim\mathcal{U}(\text{St}_{m_\ell,r_\ell})}[(PP^\top)^2](G_\ell^{(0)} - \nabla_\ell f(x^{(0)}))])$$

$$= \text{tr}\left(\mathbb{E}\left[(G_\ell^{(0)} - \nabla_\ell f(x^{(0)}))^\top \left(\frac{r_\ell}{m_\ell}\cdot I\right)(G_\ell^{(0)} - \nabla_\ell f(x^{(0)}))\right]\right)$$

$$= \delta_\ell\mathbb{E}[\|G_\ell^{(0)} - \nabla_\ell f(x^{(0)})\|_F^2]$$

$$\leq \delta_\ell\sigma_\ell^2, \qquad (61)$$

where the inequality uses Assumption 3. Applying (60)(61) and Lemma 2 to (59) yields (56).

When $t = k\tau$, $k \in \mathbb{N}^*$, we have

$$\mathbb{E}[\|\tilde{M}_\ell^{(t)} - \nabla_\ell f(x^{(t)})\|_F^2]$$

$$= \mathbb{E}[\|P_\ell^{(t)}(P_\ell^{(t)})^\top[(1-\beta_1)\tilde{M}_\ell^{(t-1)} + \beta_1 G_\ell^{(t)} - \nabla_\ell f(x^{(t)})] - (I - P_\ell^{(t)}(P_\ell^{(t)})^\top)\nabla_\ell f(x^{(t)})\|_F^2]$$

$$= \delta_\ell\mathbb{E}[\|(1-\beta_1)\tilde{M}_\ell^{(t-1)} + \beta_1 G_\ell^{(t)} - \nabla_\ell f(x^{(t)})\|_F^2] + (1-\delta_\ell)\mathbb{E}[\|\nabla_\ell f(x^{(t)})\|_F^2], \qquad (62)$$

where the second equality uses Lemma 3 and Lemma 5. For the first term, we have

$$\mathbb{E}[\|(1-\beta_1)\tilde{\boldsymbol{M}}_\ell^{(t-1)} + \beta_1\boldsymbol{G}_\ell^{(t)} - \nabla_\ell f(\boldsymbol{x}^{(t)})\|_F^2]$$

$$=\mathbb{E}[\|(1-\beta_1)(\tilde{\boldsymbol{M}}_\ell^{(t-1)} - \nabla_\ell f(\boldsymbol{x}^{(t)})) + \beta_1(\boldsymbol{G}_\ell^{(t)} - \nabla_\ell f(\boldsymbol{x}^{(t)}))\|_F^2]$$

$$\leq\mathbb{E}[\|(1-\beta_1)(\tilde{\boldsymbol{M}}_\ell^{(t-1)} - \nabla_\ell f(\boldsymbol{x}^{(t)}))\|_F^2] + \beta_1^2\mathbb{E}[\|\boldsymbol{G}_\ell^{(t)} - \nabla_\ell f(\boldsymbol{x}^{(t)})\|_F^2]$$

$$\leq(1-\beta_1)\mathbb{E}[\|\tilde{\boldsymbol{M}}_\ell^{(t-1)} - \nabla_\ell f(\boldsymbol{x}^{(t)})\|_F^2] + \beta_1^2\sigma_\ell^2, \tag{63}$$

where both inequalities use Assumption 3. By Young's inequality, we have

$$\mathbb{E}[\|\tilde{\boldsymbol{M}}_\ell^{(t-1)} - \nabla_\ell f(\boldsymbol{x}^{(t)})\|_F^2]$$

$$=\mathbb{E}[\|(\tilde{\boldsymbol{M}}_\ell^{(t-1)} - \nabla_\ell f(\boldsymbol{x}^{(t-1)})) - (\nabla_\ell f(\boldsymbol{x}^{(t)}) - \nabla_\ell f(\boldsymbol{x}^{(t-1)})\|_F^2]$$

$$\leq\left(1+\frac{\delta_\ell\beta_1}{4}\right)\mathbb{E}[\|\tilde{\boldsymbol{M}}_\ell^{(t-1)} - \nabla_\ell f(\boldsymbol{x}^{(t-1)})\|_F^2] + \left(1+\frac{4}{\delta_\ell\beta_1}\right)\mathbb{E}[\|\nabla_\ell f(\boldsymbol{x}^{(t)}) - \nabla_\ell f(\boldsymbol{x}^{(t-1)})\|_F^2]. \tag{64}$$

Applying (63)(64) and Lemma 2 to (62) yields (57).

When $t = k\tau + r, k \in \mathbb{N}, 1 \leq r < \tau$, we have

$$\mathbb{E}[\|\tilde{\boldsymbol{M}}_\ell^{(t)} - \nabla_\ell f(\boldsymbol{x}^{(t)})\|_F^2]$$

$$=\mathbb{E}[\|(1-\beta_1)(\tilde{\boldsymbol{M}}_\ell^{(t-1)} - \nabla_\ell f(\boldsymbol{x}^{(t)})) + \beta_1(\boldsymbol{P}_\ell^{(t)}(\boldsymbol{P}_\ell^{(t)})^\top\boldsymbol{G}_\ell^{(t)} - \nabla_\ell f(\boldsymbol{x}^{(t)}))\|_F^2]$$

$$=\mathbb{E}[\|(1-\beta_1)(\tilde{\boldsymbol{M}}_\ell^{(t-1)} - \nabla_\ell f(\boldsymbol{x}^{(t)})) + \beta_1(\boldsymbol{P}_\ell^{(t)}(\boldsymbol{P}_\ell^{(t)})^\top - \boldsymbol{I})\nabla_\ell f(\boldsymbol{x}^{(t)})\|_F^2]$$

$$\quad + \beta_1^2\mathbb{E}[\boldsymbol{P}_\ell^{(t)}(\boldsymbol{P}_\ell^{(t)})^\top(\boldsymbol{G}_\ell^{(t)} - \nabla_\ell f(\boldsymbol{x}^{(t)}))\|_F^2]$$

$$\leq(1-\beta_1)\mathbb{E}[\|\tilde{\boldsymbol{M}}_\ell^{(t-1)} - \nabla_\ell f(\boldsymbol{x}^{(t)})\|_F^2] + \beta_1\mathbb{E}[\|(\boldsymbol{I} - \boldsymbol{P}_\ell^{(t)}(\boldsymbol{P}_\ell^{(t)})^\top)\nabla_\ell f(\boldsymbol{x}^{(t)})\|_F^2$$

$$\quad + \beta_1^2\mathbb{E}[\boldsymbol{P}_\ell^{(t)}(\boldsymbol{P}_\ell^{(t)})^\top(\boldsymbol{G}_\ell^{(t)} - \nabla_\ell f(\boldsymbol{x}^{(t)}))\|_F^2], \tag{65}$$

where the second equality uses the unbiasedness of $\boldsymbol{G}_\ell^{(t)}$ and the independence implied by $\boldsymbol{P}_\ell^{(t)} = \boldsymbol{P}_\ell^{(t-1)}$, the inequality uses Jensen's inequality. The first term is similarly bounded as (64). For the second term, we have

$$\mathbb{E}[\|(\boldsymbol{I} - \boldsymbol{P}_\ell^{(k\tau)}(\boldsymbol{P}_\ell^{(k\tau)})^\top)\nabla_\ell f(\boldsymbol{x}^{(t)})\|_F^2]$$

$$\leq\left(1+\frac{\delta_\ell}{4}\right)\mathbb{E}[\|(\boldsymbol{I} - \boldsymbol{P}_\ell^{(k\tau)}(\boldsymbol{P}_\ell^{(k\tau)})^\top)\nabla_\ell f(\boldsymbol{x}^{(k\tau)})\|_F^2]$$

$$\quad + \left(1+\frac{4}{\delta_\ell}\right)\mathbb{E}[\|(\boldsymbol{I} - \boldsymbol{P}_\ell^{(k\tau)}(\boldsymbol{P}_\ell^{(k\tau)})^\top)(\nabla_\ell f(\boldsymbol{x}^{(t)}) - \nabla_\ell f(\boldsymbol{x}^{(k\tau)}))\|_F^2]$$

$$\leq\left(1-\frac{3\delta_\ell}{4}\right)\mathbb{E}[\|\nabla_\ell f(\boldsymbol{x}^{(k\tau)})\|_F^2] + \left(1+\frac{4}{\delta_\ell}\right)\mathbb{E}[\|\nabla_\ell f(\boldsymbol{x}^{(t)}) - \nabla_\ell f(\boldsymbol{x}^{(k\tau)})\|_F^2], \tag{66}$$

where the first inequality uses Young's inequality, the second inequality uses Lemma 5 and $\|\boldsymbol{I} - \boldsymbol{P}_\ell^{(k\tau)}(\boldsymbol{P}_\ell^{(k\tau)})^\top\|_2 = 1$. By Young's inequality, we have

$$\mathbb{E}[\|\nabla_\ell f(\boldsymbol{x}^{(k\tau)})\|_F^2] \leq \left(1+\frac{\delta_\ell}{4}\right)\mathbb{E}[\|\nabla_\ell f(\boldsymbol{x}^{(t)})\|_F^2] + \left(1+\frac{4}{\delta_\ell}\right)\mathbb{E}[\|\nabla_\ell f(\boldsymbol{x}^{(t)}) - \nabla_\ell f(\boldsymbol{x}^{(k\tau)})\|_F^2]. \tag{67}$$

Applying (67) to (66) and applying Cauchy's inequality yields

$$\mathbb{E}[\|(\boldsymbol{I} - \boldsymbol{P}_\ell^{(k\tau)}(\boldsymbol{P}_\ell^{(k\tau)})^\top)\nabla_\ell f(\boldsymbol{x}^{(t)})\|_F^2]$$

$$\leq\left(1-\frac{\delta_\ell}{2}\right)\mathbb{E}[\|\nabla_\ell f(\boldsymbol{x}^{(t)})\|_F^2] + \frac{10r}{\delta_\ell}\sum_{i=1}^r\mathbb{E}[\|\nabla_\ell f(\boldsymbol{x}^{(k\tau+i)}) - \nabla_\ell f(\boldsymbol{x}^{(k\tau+i-1)})\|_F^2]. \tag{68}$$

For the third term, we have

$$\mathbb{E}[\|\boldsymbol{P}_\ell^{(k\tau)}(\boldsymbol{P}_\ell^{(k\tau)})^\top(\boldsymbol{G}_\ell^{(t)} - \nabla_\ell f(\boldsymbol{x}^{(t)}))\|_F^2] \leq \mathbb{E}[\|\boldsymbol{G}_\ell^{(t)} - \nabla_\ell f(\boldsymbol{x}^{(t)})\|_F^2] \leq \sigma_\ell^2, \tag{69}$$

where the first inequality uses $\|\boldsymbol{P}_\ell^{(k\tau)}(\boldsymbol{P}_\ell^{(k\tau)})^\top\|_2 = 1$, the second inequality uses Assumption 3.

Applying (64)(68)(69) to (65) yields (58). $\qquad\square$

**Lemma 11** (Momentum error). *Under Assumption 2-3, if $0 < \beta_1 \leq 1$ in GoLore using MSGD and MP (Alg. 4), it holds for any $K \geq 1$ that*

$$\sum_{t=0}^{K\tau-1} \mathbb{E}[\|\tilde{\boldsymbol{m}}^{(t)} - \nabla f(\boldsymbol{x}^{(t)})\|_2^2]$$

$$\leq \left( \frac{5(1-\beta_1)}{(1-\underline{\delta}/4)\underline{\delta}\beta_1^2} + \frac{5\tau(\tau-1)}{(1-\underline{\delta}/4)\underline{\delta}} + \frac{\tau-1}{(1-\overline{\delta}/4)\beta_1} \right) L^2 \sum_{t=0}^{K\tau-2} \mathbb{E}[\|\boldsymbol{x}^{(t+1)} - \boldsymbol{x}^{(t)}\|_2^2]$$

$$+ \left( \frac{1-\underline{\delta}/2}{1-\underline{\delta}/4} + \frac{2}{(1-\overline{\delta}/4)\tau\beta_1} \right) \sum_{t=0}^{K\tau-1} \mathbb{E}[\|\nabla f(\boldsymbol{x}^{(t)})\|_2^2] + \frac{K\tau\beta_1\sigma^2}{1-\overline{\delta}/4}. \tag{70}$$

*Proof.* By Lemma 10 we have

$$\sum_{t=0}^{K\tau-1} \mathbb{E}[\|\tilde{\boldsymbol{M}}_\ell^{(t)} - \nabla_\ell f(\boldsymbol{x}^{(t)})\|_F^2] - \left( 1 - \left( 1 - \frac{\delta_\ell}{4} \right) \beta_1 \right) \sum_{t=0}^{K\tau-2} \mathbb{E}[\|\tilde{\boldsymbol{M}}_\ell^{(t)} - \nabla_\ell f(\boldsymbol{x}^{(t)})\|_F^2]$$

$$\leq \left( \frac{5(1-\beta_1)}{\delta_\ell\beta_1} + \frac{5\tau(\tau-1)\beta_1}{\delta_\ell} + (\tau-1) \right) \sum_{t=0}^{K\tau-2} \mathbb{E}[\|\nabla_\ell f(\boldsymbol{x}^{(t+1)}) - \nabla_\ell f(\boldsymbol{x}^{(t)})\|_F^2]$$

$$+ \left( \frac{2}{\tau} + \left( 1 - \frac{\delta_\ell}{2} \right) \beta_1 \right) \sum_{t=0}^{K\tau-1} \mathbb{E}[\|\nabla_\ell f(\boldsymbol{x}^{(t)})\|_F^2] + K\tau\beta_1^2\sigma_\ell^2,$$

which implies

$$\sum_{t=0}^{K\tau-1} \mathbb{E}[\|\tilde{\boldsymbol{M}}_\ell^{(t)} - \nabla_\ell f(\boldsymbol{x}^{(t)})\|_F^2]$$

$$\leq \left( \frac{5(1-\beta_1)}{(1-\delta_\ell/4)\delta_\ell\beta_1^2} + \frac{5\tau(\tau-1)}{(1-\delta_\ell/4)\delta_\ell} + \frac{\tau-1}{(1-\delta_\ell/4)\beta_1} \right) \sum_{t=0}^{K\tau-2} \mathbb{E}[\|\nabla_\ell f(\boldsymbol{x}^{(t+1)}) - \nabla_\ell f(\boldsymbol{x}^{(t)})\|_F^2]$$

$$+ \left( \frac{1-\delta_\ell/2}{1-\delta_\ell/4} + \frac{2}{(1-\delta_\ell/4)\tau\beta_1} \right) \sum_{t=0}^{K\tau-1} \mathbb{E}[\|\nabla_\ell f(\boldsymbol{x}^{(t)})\|_F^2] + \frac{K\tau\beta_1\sigma_\ell^2}{1-\delta_\ell/4}. \tag{71}$$

Summing (71) for $\ell = 1, \cdots, N_L$ and applying Assumption 2-3 yields (70). $\square$

Now we are ready to prove the convergence of Alg. 4.

**Theorem 8** (Convergence of Golore). *Under Assumptions 1-3, if hyperparameters*

$$0 < \beta_1 \leq 1, \quad \tau \geq \frac{64}{3\beta_1\underline{\delta}}, \quad 0 < \eta \leq \min \left\{ \frac{1}{4L}, \sqrt{\frac{3\underline{\delta}\beta_1^2}{80L^2}}, \sqrt{\frac{3\underline{\delta}}{80\tau^2L^2}}, \sqrt{\frac{3\beta_1}{16\tau L^2}} \right\}, \tag{72}$$

*GoLore using small-batch stochastic gradients and MSGD with MP (Alg. 4) converges as*

$$\frac{1}{K\tau} \sum_{t=0}^{K\tau-1} \mathbb{E}\|\nabla f(\boldsymbol{x}^{(t)})\|_2^2 \leq \frac{16\Delta}{\underline{\delta}\eta K\tau} + \frac{32\beta_1\sigma^2}{3\underline{\delta}} \tag{73}$$

*for any $K \geq 1$, where $\Delta = f(\boldsymbol{x}^{(0)}) - \inf_{\boldsymbol{x}} f(\boldsymbol{x})$.*

*Proof.* By Lemma 4 we have

$$\sum_{t=0}^{K\tau-1} \mathbb{E}[\|\nabla f(\boldsymbol{x}^{(t)})\|_2^2] \leq \frac{2[f(\boldsymbol{x}^{(0)}) - \mathbb{E}[f(\boldsymbol{x}^{(K\tau)})]]}{\eta} + \sum_{t=0}^{K\tau-1} \mathbb{E}[\|\tilde{\boldsymbol{m}}^{(t)} - \nabla f(\boldsymbol{x}^{(t)})\|_2^2]$$

$$- \left( \frac{1}{\eta^2} - \frac{L}{\eta} \right) \sum_{t=0}^{K\tau-1} \mathbb{E}[\|\boldsymbol{x}^{(t+1)} - \boldsymbol{x}^{(t)}\|_2^2]. \tag{74}$$

Applying Lemma 11 to (74) and using $\underline{\delta} \leq \overline{\delta} < 1$ yields

$$\left(\frac{\underline{\delta}}{4} - \frac{8}{3\tau\beta_1}\right) \sum_{t=0}^{K\tau-1} \mathbb{E}[\|\nabla f(\boldsymbol{x}^{(t)})\|_2^2]$$

$$\leq \frac{2}{\eta} \mathbb{E}[f(\boldsymbol{x}^{(0)}) - f(\boldsymbol{x}^{(K\tau)})] + \frac{4K\tau\beta_1\sigma^2}{3}$$

$$- \left(\frac{1}{\eta^2} - \frac{L}{\eta} - \frac{20(1-\beta_1)L^2}{3\underline{\delta}\beta_1^2} - \frac{20\tau(\tau-1)L^2}{3\underline{\delta}} - \frac{4(\tau-1)L^2}{3\beta_1}\right) \sum_{t=0}^{K\tau-1} \mathbb{E}[\|\boldsymbol{x}^{(t+1)} - \boldsymbol{x}^{(t)}\|_2^2].$$

$$\tag{75}$$

By (72) we have

$$\frac{\underline{\delta}}{4} - \frac{8}{3\tau\beta_1} \geq \frac{\underline{\delta}}{8}, \quad \text{and} \quad \frac{1}{4\eta^2} \geq \max\left\{\frac{L}{\eta}, \frac{20(1-\beta_1)L^2}{3\underline{\delta}\beta_1^2}, \frac{20\tau(\tau-1)L^2}{3\underline{\delta}}, \frac{4(\tau-1)L^2}{3\beta_1}\right\}. \tag{76}$$

Applying (76) to (75) yields (73). $\qquad\qquad\qquad\qquad\qquad\qquad\qquad\qquad\qquad\qquad\qquad\square$

We now prove Theorem 4, which is restated as follows.

**Corollary 3** (Convergence complexity of GoLore). *Under Assumptions 1-3, if $T \geq 2 + 128/(3\underline{\delta}) + (128\sigma)^2/(9\sqrt{\underline{\delta}}L\Delta)$ and we choose*

$$\beta_1 = \left(1 + \sqrt{\frac{\underline{\delta}^{3/2}\sigma^2 T}{L\Delta}}\right)^{-1},$$

$$\tau = \left\lceil \frac{64}{3\underline{\delta}\beta_1} \right\rceil,$$

$$\eta = \left(4L + \sqrt{\frac{80L^2}{3\underline{\delta}\beta_1^2}} + \sqrt{\frac{80\tau^2 L^2}{3\underline{\delta}}} + \sqrt{\frac{16\tau L^2}{3\beta_1}}\right)^{-1},$$

*GoLore using small-batch stochastic gradients and MSGD with MP (Alg. 4) converges as*

$$\frac{1}{T}\sum_{t=0}^{T-1} \mathbb{E}[\|\nabla f(\boldsymbol{x}^{(t)})\|_2^2] = \mathcal{O}\left(\frac{L\Delta}{\underline{\delta}^{5/2}T} + \sqrt{\frac{L\Delta\sigma^2}{\underline{\delta}^{7/2}T}}\right), \tag{77}$$

*where $\Delta = f(\boldsymbol{x}^{(0)}) - \inf_{\boldsymbol{x}} f(\boldsymbol{x})$. Consequently, the computation complexity to reach an $\varepsilon$-accurate solution $\boldsymbol{x}$ such that $\|\nabla f(\boldsymbol{x})\|_2^2 \leq \varepsilon$ is $\mathcal{O}\left(\frac{L\Delta\sigma^2}{\underline{\delta}^{7/2}\varepsilon^2} + \frac{L\Delta}{\underline{\delta}^{5/2}\varepsilon} + \frac{\sigma^2}{\underline{\delta}^{1/2}L\Delta} + \frac{1}{\underline{\delta}}\right)$.*

*Proof.* $T \geq 2 + 128/(3\underline{\delta}) + (128\sigma)^2/(9\sqrt{\underline{\delta}}L\Delta)$ guarantees $T \geq \tau$. Let $T = K\tau + r$, where $K \in \mathbb{N}^*$ and $0 \leq r < \tau$. If $r = 0$, (77) is a direct result of Theorem 8. If $r > 0$, applying Theorem 8 to $\tilde{K} := K + 1$ yields

$$\frac{1}{T}\sum_{t=0}^{T-1} \mathbb{E}[\|\nabla f(\boldsymbol{x}^{(t)})\|_2^2] \leq \frac{\tilde{K}\tau}{T} \cdot \frac{1}{\tilde{K}\tau}\sum_{t=0}^{\tilde{K}\tau-1} \mathbb{E}[\|\nabla f(\boldsymbol{x}^{(t)})\|_2^2] = \mathcal{O}\left(\frac{L\Delta}{\underline{\delta}^{5/2}T} + \sqrt{\frac{L\Delta\sigma^2}{\underline{\delta}^{7/2}T}}\right).$$

$$\qquad\qquad\qquad\qquad\qquad\qquad\qquad\qquad\qquad\qquad\qquad\qquad\qquad\qquad\qquad\qquad\square$$

## C   RESULTS FOR SPARSE SUBSPACE OPTIMIZATION

In this section, we illustrate how to transfer the main results of this paper to sparse subspace optimization algorithms. We first present the detailed algorithm formulation, then present the theoretical results corresponding to GaLore/GoLore. Although it only requires little effort to transfer results in GaLore/GoLore to sparse subspace optimization, we still include proofs for completeness.

**Algorithm 5** GaSare / GoSare algorithms using stochastic / deterministic / large-batch gradients

**Input:** Initial point $\boldsymbol{x}^{(0)}$, data distribution $\mathcal{D}$, learning rate $\eta$, subspace changing frequency $\tau$, rank $\{r_\ell\}_{\ell=1}^{N_L}$, optimizer hyperparameters $\beta_1$, $\beta_2$, $\epsilon$, large batch size $\mathcal{B}$.

**Output:** $\{\boldsymbol{x}^{(t)}\}_{t=0}^{T}$.

  Initialize optimizer state $\{\boldsymbol{M}_\ell^{(-1)}\}_{\ell=1}^{N_L}$ and $\{\boldsymbol{V}_\ell^{(-1)}\}_{\ell=1}^{N_L}$ to zero;

  **for** $t = 0, 1, \cdots, T-1$ **do**

    **for** $\ell = 1, 2, \cdots, N_L$ **do**

      **if** $t \equiv 0 \pmod{\tau}$ **then**

        $\boldsymbol{G}_\ell^{(t)} \leftarrow \nabla_\ell F(\boldsymbol{x}^{(t)}; \xi^{(t)})$;  (stochastic)

        $\boldsymbol{G}_\ell^{(t)} \leftarrow \nabla_\ell f(\boldsymbol{x}^{(t)})$;  (deterministic)

        $\boldsymbol{G}_\ell^{(t)} \leftarrow \frac{1}{\mathcal{B}} \sum_{b=1}^{\mathcal{B}} \nabla_\ell F(\boldsymbol{x}^{(t)}; \xi^{(t,b)})$;  (large-batch)

        $\boldsymbol{S}_\ell^{(t)} \leftarrow \text{Top}_k(\boldsymbol{G}_\ell^{(t)})$;  (GaSare)

        Sample $\boldsymbol{S}_\ell^{(t)} \sim \mathcal{U}(\text{Sp}_{m_\ell, n_\ell}^{k_\ell})$;  (GoSare)

      **else**

        $\boldsymbol{G}_\ell^{(t)} \leftarrow \nabla_\ell F(\boldsymbol{x}^{(t)}; \xi^{(t)})$;  (stochastic)

        $\boldsymbol{G}_\ell^{(t)} \leftarrow \nabla_\ell f(\boldsymbol{x}^{(t)})$;  (deterministic)

        $\boldsymbol{G}_\ell^{(t)} \leftarrow \nabla_\ell F(\boldsymbol{x}^{(t)}; \xi^{(t)})$;  (large-batch)

        $\boldsymbol{S}_\ell^{(t)} \leftarrow \boldsymbol{S}_\ell^{(t-1)}$;

      **end if**

      $\boldsymbol{R}_\ell^{(t)} \leftarrow \boldsymbol{S}_\ell^{(t)} \odot \boldsymbol{G}_\ell^{(t)}$;

      $\boldsymbol{M}_\ell^{(t)} \leftarrow (1-\beta_1)\boldsymbol{S}_\ell^{(t)} \odot \boldsymbol{M}_\ell^{(t-1)} + \beta_1 \boldsymbol{R}_\ell^{(t)}$;

      $\boldsymbol{V}_\ell^{(t)} \leftarrow (1-\beta_2)\boldsymbol{S}_\ell^{(t)} \odot \boldsymbol{V}_\ell^{(t-1)} + \beta_2 \boldsymbol{R}_\ell^{(t)} \odot \boldsymbol{R}_\ell^{(t)}$;

      **if** using Adam **then**

        $\boldsymbol{M}_\ell^{(t)} \leftarrow \boldsymbol{M}_\ell^{(t)}/(1-\beta_1^t)$,    $\boldsymbol{V}_\ell^{(t)} \leftarrow \boldsymbol{V}_\ell^{(t)}/(1-\beta_2^t)$,    $\boldsymbol{N}_\ell^{(t)} \leftarrow \boldsymbol{M}_\ell^{(t)}/(\sqrt{\boldsymbol{V}_\ell^{(t)}} + \epsilon)$;

      **else if** using MSGD **then**

        $\boldsymbol{N}_\ell^{(t)} \leftarrow \boldsymbol{M}_\ell^{(t)}$;

      **end if**

      $\boldsymbol{X}_\ell^{(t+1)} \leftarrow \boldsymbol{X}_\ell^{(t)} - \eta \boldsymbol{S}_\ell^{(t)} \odot \boldsymbol{N}_\ell^{(t)}$;

    **end for**

  **end for**

## C.1 ALGORITHM DESIGN

While low-rank subspace optimization algorithms like GaLore/GoLore project full-parameter gradient $\boldsymbol{G} \in \mathbb{R}^{(m \times n)}$ into low-rank subspaces via projection like $\boldsymbol{P}^\top \boldsymbol{G}$, sparse subspace optimization algorithms use a sparse mask $\boldsymbol{S}$ to get $\boldsymbol{S} \odot \boldsymbol{G}$. Specifically, consider the following set

$$\text{Sp}_{m,n}^k = \{\boldsymbol{S} \in \{0,1\}^{m \times n} \mid \|\boldsymbol{S}\|_F^2 = k\},$$

*i.e.*, a set of $m \times n$ matrices contains $k$ ones and $(mn - k)$ zeros. Corresponding to the subspace selecting strategy in GaLore, we consider a Top-$k$ strategy which places the $k$ ones at indices corresponding to $\boldsymbol{G}$'s elements with the $k$ largest absolute values. We also consider a Rand-$k$ strategy which samples the sparse mask matrix $\boldsymbol{S}$ from the uniform distribution on $\text{SP}_{m,n}^k$ corresponding to GoLore. For convenience, we name the algorithm using Top-$k$ strategy as GaSare (**Gra**dient **Spar**se proj**e**ction), and the one using Rand-$k$ strategy as GoSare (**G**radient rand**o**m **Spar**se proj**e**ction). The concerned sparse subspace descent algorithms are described as in Alg. 5

## C.2 NOTATIONS AND USEFUL LEMMAS

We assume the model parameters consist of $N_L$ weight matrices. We use $\boldsymbol{X}_\ell \in \mathbb{R}^{m_\ell \times n_\ell}$ to denote the $\ell$-th weight matrix and $\boldsymbol{x} \in \mathbb{R}^d = (\text{vec}(\boldsymbol{X}_1)^\top, \cdots, \text{vec}(\boldsymbol{X}_{N_L})^\top)^\top$ to denote the vector collecting all the parameters, $d = \sum_{\ell=1}^{N_L} m_\ell n_\ell$. We assume GaSare/GoSare applies sparse mask in $\text{Sp}_{m_\ell, n_\ell}^{k_\ell}$ to the $\ell$-th weight matrix and denote

$$\delta_\ell = \frac{k_\ell}{m_\ell n_\ell}, \quad \underline{\delta} = \min_{1 \leq \ell \leq N_L} \delta_\ell, \quad \overline{\delta} = \max_{1 \leq \ell \leq N_l} \delta_\ell.$$

We define $\tilde{\boldsymbol{M}}_\ell^{(t)} = \boldsymbol{S}_\ell^{(t)} \odot \boldsymbol{M}_\ell^{(t)}$ and $\tilde{\boldsymbol{m}} = (\text{vec}(\tilde{\boldsymbol{M}}_1)^\top, \cdots, \text{vec}(\tilde{\boldsymbol{M}}_{N_L})^\top)^\top$. While using Alg. 5 with MSGD, it holds that

$$\tilde{\boldsymbol{M}}_\ell^{(t)} = \begin{cases} \beta_1 \boldsymbol{S}_\ell^{(0)} \odot \boldsymbol{G}_\ell^{(0)}, & t = 0; \\ \boldsymbol{S}_\ell^{(t)} \odot \left( (1 - \beta_1) \tilde{\boldsymbol{M}}_\ell^{(t-1)} + \beta_1 \boldsymbol{G}_\ell^{(t)} \right), & t = k\tau, \ k \in \mathbb{N}^*; \\ (1 - \beta_1) \tilde{\boldsymbol{M}}_\ell^{(t-1)} + \beta_1 \boldsymbol{S}_\ell^{(t)} \odot \boldsymbol{G}_\ell^{(t)}, & t = k\tau + r, \ k \in \mathbb{N}, \ 1 \leq r < \tau; \end{cases}$$

and that

$$\boldsymbol{X}_\ell^{(t+1)} = \boldsymbol{X}_\ell^{(t)} - \eta \tilde{\boldsymbol{M}}_\ell^{(t)}.$$

We use $\boldsymbol{E}_{m,n}$ to denote the all-one $m \times n$ matrix, *i.e.*,

$$\boldsymbol{E}_{m,n} = \begin{pmatrix} 1 & 1 & \cdots & 1 \\ 1 & 1 & \cdots & 1 \\ \vdots & \vdots & \ddots & \vdots \\ 1 & 1 & \cdots & 1 \end{pmatrix} \in \mathbb{R}^{m \times n}.$$

**Lemma 12** (Error of GaSare's projection). *Let $\boldsymbol{S}$ be the Top-k mask of $\boldsymbol{G} \in \mathbb{R}^{m \times n}$, it holds that*

$$\|\boldsymbol{S} \odot \boldsymbol{G} - \boldsymbol{G}\|_F^2 \leq \left( 1 - \frac{k}{mn} \right) \|\boldsymbol{G}\|_F^2.$$

*Proof.* Let $g_1, g_2, \cdots, g_{mn}$ be elements of $\boldsymbol{G}$ such that $|g_1| \geq |g_2| \geq \cdots \geq |g_{mn}|$. It holds that

$$\begin{aligned} \|\boldsymbol{S} \odot \boldsymbol{G} - \boldsymbol{G}\|_F^2 &= \sum_{i=1}^{k} (g_k - g_k)^2 + \sum_{i=k+1}^{mn} (0 - g_k)^2 \\ &= \sum_{i=k+1}^{mn} g_k^2 \\ &\leq \left( 1 - \frac{k}{mn} \right) \sum_{i=1}^{mn} g_k^2 \\ &= \left( 1 - \frac{k}{mn} \right) \|\boldsymbol{G}\|_F^2, \end{aligned}$$

where the inequality uses $\frac{1}{mn-k} \sum_{i=k+1}^{mn} g_i^2 \leq \frac{1}{k} \sum_{i=1}^{k} g_i^2$. $\qquad \square$

**Lemma 13** (Error of GoSare's projection). *Let $\boldsymbol{S} \sim \mathcal{U}(\text{Sp}_{m,n}^k)$, it holds for all $\boldsymbol{G} \in \mathbb{R}^{m \times n}$ that*

$$\mathbb{E}[\boldsymbol{S}] = \frac{k}{mn} \cdot \boldsymbol{E}_{m,n}, \tag{78}$$

*and*

$$\mathbb{E}[\|\boldsymbol{S} \odot \boldsymbol{G} - \boldsymbol{G}\|_F^2] = \left( 1 - \frac{k}{mn} \right) \|\boldsymbol{G}\|_F^2. \tag{79}$$

*Proof.* To prove (78), it suffices to note that for any element $S_{i,j}$ in $\boldsymbol{S}$, it holds that

$$\mathbb{E}[S_{i,j}] = \mathbb{P}[S_{i,j} = 1] = \frac{(mn-1)!/[(mn-k)!(k-1)!]}{(mn)!/[(mn-k)!k!]} = \frac{k}{mn}.$$

To prove (79), we have

$$\mathbb{E}[\|\boldsymbol{S} \odot \boldsymbol{G} - \boldsymbol{G}\|_F^2] = \sum_{1 \le i \le m, 1 \le j \le n} \mathbb{P}[S_{i,j} = 0]\boldsymbol{G}_{i,j}^2 = \left(1 - \frac{k}{mn}\right)\|\boldsymbol{G}\|_F^2.$$

$\square$

## C.3 NON-CONVERGENCE OF GASARE

In this subsection, we present the non-convergence result of GaSare, similar to that of GaLore.

**Theorem 9** (Non-convergence of GaSare). *There exists an objective function $f : \mathbb{R}^d \to \mathbb{R}$ satisfying Assumptions 1, 2, a stochastic gradient oracle $(F, \mathcal{D})$ satisfying Assumption 3, an initial point $\boldsymbol{x}^{(0)}$, a constant $\epsilon_0 > 0$ such that for GaSare with any sparsity level $k_\ell < m_\ell n_\ell$, subspace changing frequency $\tau$ and any subspace optimizer $\rho$ with arbitrary hyperparameters and any $t > 0$, it holds that*

$$\|\nabla f(\boldsymbol{x}^{(t)})\|_2^2 \ge \epsilon_0.$$

*Proof.* Consider target function $f(\boldsymbol{X}) = \frac{L}{2}\|(\boldsymbol{p}\boldsymbol{p}^\top) \odot \boldsymbol{X}\|_F^2$ where $L > 0$, $\boldsymbol{X} \in \mathbb{R}^{n \times n}$ with $n > 1$ and $\boldsymbol{p} = (1, 0, \cdots, 0)^\top \in \mathbb{R}^n$. It holds that

$$f(\boldsymbol{X}) = \frac{LX_{1,1}^2}{2} \ge 0,$$

thus $f$ satisfies Assumption 1. Since $\nabla f(\boldsymbol{X}) = L(\boldsymbol{p}\boldsymbol{p}^\top) \odot \boldsymbol{X}$, it holds that

$$\|\nabla f(\boldsymbol{X}) - \nabla f(\boldsymbol{Y})\|_F = L\|(\boldsymbol{p}\boldsymbol{p}^\top) \odot (\boldsymbol{X} - \boldsymbol{Y})\|_F \le L\|\boldsymbol{X} - \boldsymbol{Y}\|_F,$$

thus $f$ satisfies Assumption 2.

Consider the following stochastic gradient oracle:

$$F(\boldsymbol{X}; \xi) = f(\boldsymbol{X}) + \xi\tilde{\sigma} \cdot \mathrm{tr}(\boldsymbol{Q}\boldsymbol{X}), \quad \text{and} \quad \mathbb{P}_{\xi \sim \mathcal{D}}[\xi = 1] = \mathbb{P}_{\xi \sim \mathcal{D}}[\xi = -1] = 0.5,$$

where $\tilde{\sigma} = \sigma/\sqrt{n^2(n^2-1)/2}$ and

$$\boldsymbol{Q} = \begin{pmatrix} 0 & \sqrt{n} & \cdots & \sqrt{n^2-n} \\ \sqrt{1} & \sqrt{n+1} & \cdots & \sqrt{n^2-n+1} \\ \vdots & \vdots & \ddots & \vdots \\ \sqrt{n-1} & \sqrt{2n-1} & \cdots & \sqrt{n^2-1} \end{pmatrix} \in \mathbb{R}^{n \times n}.$$

Note that $\nabla F(\boldsymbol{X}; \xi) = \nabla f(\boldsymbol{X}) + \xi\tilde{\sigma}\boldsymbol{Q}$, it holds for any $\boldsymbol{X} \in \mathbb{R}^{n \times n}$ that

$$\mathbb{E}_{\xi \sim \mathcal{D}}[\nabla F(\boldsymbol{X}; \xi)] = \nabla f(\boldsymbol{X})$$

$$\mathbb{E}_{\xi \sim \mathcal{D}}[\|\nabla F(\boldsymbol{X}; \xi) - \nabla f(\boldsymbol{X})\|_F^2] = \tilde{\sigma}^2\|\boldsymbol{Q}\|_F^2 = \frac{\sigma^2}{n^2(n^2-1)/2} \cdot \sum_{i=1}^{n^2-1} i = \sigma^2,$$

thus oracle $(F, \mathcal{D})$ satisfies Assumption 3.

Consider the initial point $\boldsymbol{X}^{(0)}$ with $X_{1,1}^{(0)} = \lambda$, where $0 < \lambda < \tilde{\sigma}/L$ is a scalar. We show that GaSare with the above objective function $f$, stochastic gradient oracle $(F, \mathcal{D})$, initial point $\boldsymbol{X}^{(0)}$, arbitrary sparsity level $0 < k < n^2$, arbitrary subspace changing frequency $\tau$ and arbitrary subspace optimizer $\rho$, can only output points $\boldsymbol{X}^{(t)}$ with $\|\nabla f(\boldsymbol{X}^{(t)})\|_F^2 \ge \epsilon_0$ for $\epsilon_0 = L^2\lambda^2 > 0$.

When $\tau \mid t$, GaSare recomputes the sparse mask matrix at iteration $t$. If $X_{1,1}^{(t)} = \lambda$, the stochastic gradient is given by

$$\boldsymbol{G}^{(t)} = L(\boldsymbol{p}\boldsymbol{p}^\top) \odot \boldsymbol{X} + \xi^{(t)}\tilde{\sigma}\boldsymbol{Q}.$$

since $L\lambda < \tilde{\sigma}$, the Top-$k$ mask $\boldsymbol{S} \in \mathbb{R}^{n \times n}$ satisfies

$$\text{vec}(\boldsymbol{S}) = (\underbrace{0, 0, \cdots, 0}_{(n^2-k)\times}, \underbrace{1, 1, \cdots, 1}_{k\times})^\top \in \mathbb{R}^{n^2},$$

Using this mask matrix, the subspace updates in the following $\tau$ iterations is as

$$\boldsymbol{X}^{(t+\Delta_t)} = \boldsymbol{X}^{(t)} + \boldsymbol{S}^{(t)} \odot \left( \sum_{s=0}^{\Delta_t-1} \rho^{(t+s)}(\boldsymbol{S}^{(t)} \odot \boldsymbol{G}^{(t)}) \right) \quad \Rightarrow \quad X_{1,1}^{(t+\Delta_t)} = X_{1,1}^{(t)} = \lambda,$$

for $\Delta_t = 1, 2, \cdots, \tau$. Since $X_{1,1}^{(0)} = \lambda$, it holds for all $t > 0$ that $\boldsymbol{X}_{1,1}^{(t)} = \lambda$ and thus

$$\|\nabla f(\boldsymbol{X}^{(t)})\|_F^2 = L^2\lambda^2 = \epsilon_0.$$

$\square$

### C.4 CONVERGENCE OF DETERMINISTIC GASARE

In this subsection, we prove the convergence properties of GaSare with deterministic gradients. The results and proofs are similar to those of deterministic GaLore in Appendix B.3.

**Lemma 14** (Momentum contraction). *In deterministic GaSare using MSGD (Alg. 5), if $0 < \beta_1 \leq 1$, term $\tilde{\boldsymbol{M}}_\ell^{(t)}$ has the following contraction properties:*

- *When $t = 0$, it holds that*

$$\|\tilde{\boldsymbol{M}}_\ell^{(0)} - \nabla_\ell f(\boldsymbol{X}^{(0)})\|_F^2 \leq (\tau-1)(1-\delta_\ell\beta_1) \sum_{r=0}^{\tau-2} \|\nabla_\ell f(\boldsymbol{x}^{(r+1)}) - \nabla_\ell f(\boldsymbol{x}^{(r)})\|_F^2$$

$$+ \frac{2(1-\delta_\ell\beta_1)}{\tau} \sum_{r=0}^{\tau-1} \|\nabla_\ell f(\boldsymbol{x}^{(r)})\|_F^2; \tag{80}$$

- *When $t = k\tau$, $k \in \mathbb{N}^*$, it holds that*

$$\|\tilde{\boldsymbol{M}}_\ell^{(t)} - \nabla_\ell f(\boldsymbol{x}^{(t)})\|_F^2 - \left(1 - \left(1 - \frac{\delta_\ell}{4}\right)\beta_1\right) \|\tilde{\boldsymbol{M}}_\ell^{(t-1)} - \nabla_\ell f(\boldsymbol{x}^{(t-1)})\|_F^2$$

$$\leq \frac{2(1-\delta_\ell)}{\tau} \sum_{r=0}^{\tau-1} \|\nabla_l f(\boldsymbol{x}^{(k\tau+r)})\|_F^2 + \frac{5(1-\beta_1)}{\delta_\ell\beta_1} \|\nabla_\ell f(\boldsymbol{x}^{(t)}) - \nabla_\ell f(\boldsymbol{x}^{(t-1)})\|_F^2$$

$$+ (\tau-1)(1-\delta_\ell) \sum_{r=0}^{\tau-2} \|\nabla_\ell f(\boldsymbol{x}^{(k\tau+r+1)}) - \nabla_\ell f(\boldsymbol{x}^{(k\tau+r)})\|_F^2; \tag{81}$$

- *When $t = k\tau + r$, $k \in \mathbb{N}$, $1 \leq r < \tau$, it holds that*

$$\mathbb{E}[\|\tilde{\boldsymbol{M}}_\ell^{(t)} - \nabla_\ell f(\boldsymbol{x}^{(t)})\|_F^2] - \left(1 - \left(1 - \frac{\delta_\ell}{4}\right)\beta_1\right) \mathbb{E}[\|\tilde{\boldsymbol{M}}_\ell^{(t-1)} - \nabla_\ell f(\boldsymbol{x}^{(t-1)})\|_F^2]$$

$$\leq \left(1 - \frac{\delta_\ell}{2}\right)\beta_1 \mathbb{E}[\|\nabla_\ell f(\boldsymbol{x}^{(t)})\|_F^2] + \frac{5(1-\beta_1)}{\delta_\ell\beta_1} \mathbb{E}[\|\nabla_\ell f(\boldsymbol{x}^{(t)}) - \nabla_\ell f(\boldsymbol{x}^{(t-1)})\|_F^2]$$

$$+ \frac{10r\beta_1}{\delta_\ell} \sum_{i=1}^{r} \mathbb{E}[\|\nabla_\ell f(\boldsymbol{x}^{(k\tau+i)}) - \nabla_\ell f(\boldsymbol{x}^{(k\tau+i-1)})\|_F^2]. \tag{82}$$

*Proof.* For convenience we use $\boldsymbol{E}$ to denote $\boldsymbol{E}_{m_\ell, n_\ell}$. When $t = 0$, we have

$$\|\tilde{\boldsymbol{M}}_\ell^{(0)} - \nabla_\ell f(\boldsymbol{x}^{(0)})\|_F^2 = \|\beta_1(\boldsymbol{S}_\ell^{(0)} - \boldsymbol{E}) \odot \nabla_\ell f(\boldsymbol{x}^{(0)}) - (1 - \beta_1)\nabla_\ell f(\boldsymbol{x}^{(0)})\|_F^2$$

$$\leq \beta_1(1 - \delta_\ell)\|\nabla_\ell f(\boldsymbol{x}^{(0)})\|_F^2 + (1 - \beta_1)\|\nabla_\ell f(\boldsymbol{x}^{(0)})\|_F^2$$

$$= (1 - \delta_\ell\beta_1)\|\nabla_\ell f(\boldsymbol{x}^{(0)})\|_F^2, \tag{83}$$

where the inequality uses Lemma 12 and Jensen's inequality. Applying Lemma 2 to (83) yields (80).

When $t = k\tau$, $k \in \mathbb{N}^*$, we have

$$\|\tilde{\boldsymbol{M}}_\ell^{(t)} - \nabla_\ell f(\boldsymbol{x}^{(t)})\|_F^2$$
$$= \|\boldsymbol{S}_\ell^{(t)} \odot [(1-\beta_1)\tilde{\boldsymbol{M}}_\ell^{(t-1)} + \beta_1 \boldsymbol{G}_\ell^{(t)} - \nabla_\ell f(\boldsymbol{x}^{(t)})] - (\boldsymbol{E} - \boldsymbol{S}_\ell^{(t)}) \odot \nabla_\ell f(\boldsymbol{x}^{(t)})\|_F^2$$
$$= \|\boldsymbol{S}_\ell^{(t)} \odot [(1-\beta_1)(\tilde{\boldsymbol{M}}_\ell^{(t-1)} - \nabla_\ell f(\boldsymbol{x}^{(t)}))]\|_F^2 + \|(\boldsymbol{E} - \boldsymbol{S}_\ell^{(t)}) \odot \nabla_\ell f(\boldsymbol{x}^{(t)})\|_F^2$$
$$\leq \|(1-\beta_1)(\tilde{\boldsymbol{M}}_\ell^{(t-1)} - \nabla_\ell f(\boldsymbol{x}^{(t)}))\|_F^2 + (1-\delta_\ell)\|\nabla_\ell f(\boldsymbol{x}^{(t)})\|_F^2, \qquad (84)$$

where the inequality uses Lemma 12. By Young's inequality, we have

$$\|\tilde{\boldsymbol{M}}_\ell^{(t-1)} - \nabla_\ell f(\boldsymbol{x}^{(t)})\|_F^2$$
$$= \|(\tilde{\boldsymbol{M}}_\ell^{(t-1)} - \nabla_\ell f(\boldsymbol{x}^{(t-1)})) - (\nabla_\ell f(\boldsymbol{x}^{(t)}) - \nabla_\ell f(\boldsymbol{x}^{(t-1)}))\|_F^2$$
$$\leq \left(1 + \frac{\delta_\ell \beta_1}{4}\right)\|\tilde{\boldsymbol{M}}_\ell^{(t-1)} - \nabla_\ell f(\boldsymbol{x}^{(t-1)})\|_F^2 + \left(1 + \frac{4}{\delta_\ell \beta_1}\right)\|\nabla_\ell f(\boldsymbol{x}^{(t)}) - \nabla_\ell f(\boldsymbol{x}^{(t-1)})\|_F^2.$$
$$(85)$$

Applying Lemma 2 and (85) to (84) yields (81).

When $t = k\tau + r$, $k \in \mathbb{N}$, $1 \leq r < \tau$, we have

$$\|\tilde{\boldsymbol{M}}_\ell^{(t)} - \nabla_\ell f(\boldsymbol{x}^{(t)})\|_F^2$$
$$= \|(1-\beta_1)(\tilde{\boldsymbol{M}}_\ell^{(t-1)} - \nabla_\ell f(\boldsymbol{x}^{(t)})) + \beta_1(\boldsymbol{S}_\ell^{(t)} - \boldsymbol{E}) \odot \nabla_\ell f(\boldsymbol{x}^{(t)})\|_F^2$$
$$\leq (1-\beta_1)\|\tilde{\boldsymbol{M}}_\ell^{(t-1)} - \nabla_\ell f(\boldsymbol{x}^{(t)})\|_F^2 + \beta_1\|(\boldsymbol{E} - \boldsymbol{S}_\ell^{(k\tau)}) \odot \nabla_\ell f(\boldsymbol{x}^{(t)})\|_F^2, \qquad (86)$$

where the inequality uses Jensen's inequality and $\boldsymbol{S}_\ell^{(t)} = \boldsymbol{S}_\ell^{(t-1)} = \cdots = \boldsymbol{S}_\ell^{(k\tau)}$. The first term can be similarly upper bounded as (85). For the second term, we have

$$(\boldsymbol{E} - \boldsymbol{S}_\ell^{(k\tau)}) \odot \nabla_\ell f(\boldsymbol{x}^{(t)})\|_F^2$$
$$\leq \left(1 + \frac{\delta_\ell}{4}\right)\|(\boldsymbol{E} - \boldsymbol{S}_\ell^{(k\tau)}) \odot \nabla_\ell f(\boldsymbol{x}^{(k\tau)})\|_F^2$$
$$+ \left(1 + \frac{4}{\delta_\ell}\right)\|(\boldsymbol{E} - \boldsymbol{S}_\ell^{(k\tau)}) \odot (\nabla_\ell f(\boldsymbol{x}^{(t)}) - \nabla_\ell f(\boldsymbol{x}^{(k\tau)}))\|_F^2$$
$$\leq \left(1 + \frac{\delta_\ell}{4}\right)(1-\delta_\ell)\|\nabla_\ell f(\boldsymbol{x}^{(k\tau)})\|_F^2 + \frac{5}{\delta_\ell}\|\nabla_\ell f(\boldsymbol{x}^{(t)}) - \nabla_\ell f(\boldsymbol{x}^{(k\tau)})\|_F^2, \qquad (87)$$

where the first inequality uses Young's inequality and the second inequality uses Lemma 12. By Young's inequality, we have

$$\|\nabla_\ell f(\boldsymbol{x}^{(k\tau)})\|_F^2 \leq \left(1 + \frac{\delta_\ell}{4}\right)\|\nabla_\ell f(\boldsymbol{x}^{(t)})\|_F^2 + \left(1 + \frac{4}{\delta_\ell}\right)\|\nabla_\ell f(\boldsymbol{x}^{(t)}) - \nabla_\ell f(\boldsymbol{x}^{(k\tau)})\|_F^2. \quad (88)$$

Note that $t = k\tau + r$, we further have

$$\|\nabla_\ell f(\boldsymbol{x}^{(t)}) - \nabla_\ell f(\boldsymbol{x}^{(k\tau)})\|_F^2 = \left\|\sum_{i=1}^r \nabla_\ell f(\boldsymbol{x}^{(k\tau+i)}) - \nabla_\ell f(\boldsymbol{x}^{(k\tau+i-1)})\right\|_F^2$$
$$\leq r\sum_{i=1}^r \|\nabla_\ell f(\boldsymbol{x}^{(k\tau+i)}) - \nabla_\ell f(\boldsymbol{x}^{(k\tau+i-1)})\|_F^2, \qquad (89)$$

where the inequality uses Cauchy's inequality. Applying (88)(89) to (87) yields

$$(\boldsymbol{E} - \boldsymbol{S}_\ell^{(k\tau)}) \odot \nabla_\ell f(\boldsymbol{x}^{(t)})\|_F^2$$
$$\leq \left(1 - \frac{\delta_\ell}{2}\right)\|\nabla_\ell f(\boldsymbol{x}^{(t)})\|_F^2 + \frac{10r}{\delta_\ell}\sum_{i=1}^r \|\nabla_\ell f(\boldsymbol{x}^{(k\tau+i)}) - \nabla_\ell f(\boldsymbol{x}^{(k\tau+i-1)})\|_F^2. \qquad (90)$$

Applying (85)(90) to (86) yields (82). $\qquad\square$

Based on Lemma 14, we can prove the convergence properties of deterministic GaSare similarly as the proofs of Lemma 7, Theorem 6 and Corollary 1. Below we directly present the final convergence results.

**Theorem 10** (Convergence of deterministic GaSare). *Under Assumptions 1-2, if hyperparameters*

$$0 < \beta_1 \leq 1, \quad \tau \geq \frac{64}{3\beta_1\underline{\delta}}, \quad 0 < \eta \leq \min\left\{\frac{1}{4L}, \sqrt{\frac{3\underline{\delta}\beta_1^2}{80L^2}}, \sqrt{\frac{3\underline{\delta}}{80\tau^2L^2}}, \sqrt{\frac{3\beta_1}{16\tau L^2}}\right\},$$

*GaSare using deterministic gradients and MSGD (Alg. 5) converges as*

$$\frac{1}{K\tau}\sum_{t=0}^{K\tau-1}\|\nabla f(\boldsymbol{x}^{(t)})\|_2^2 \leq \frac{16\Delta}{\underline{\delta}\eta K\tau}$$

*for any $K \geq 1$, where $\Delta = f(\boldsymbol{x}^{(0)}) - \inf_{\boldsymbol{x}} f(\boldsymbol{x})$. If $T \geq 64/(3\underline{\delta})$ and we further choose*

$$\beta_1 = 1$$

$$\tau = \left\lceil\frac{64}{3\underline{\delta}\beta_1}\right\rceil$$

$$\eta = \left(4L + \sqrt{\frac{80L^2}{3\underline{\delta}\beta_1^2}} + \sqrt{\frac{80\tau^2L^2}{3\underline{\delta}}} + \sqrt{\frac{16\tau L^2}{3\beta_1}}\right)^{-1},$$

*GaSare using deterministic gradients and MSGD (Alg. 5) converges as*

$$\frac{1}{T}\sum_{t=0}^{T-1}\|\nabla f(\boldsymbol{x}^{(t)})\|_2^2 = \mathcal{O}\left(\frac{L\Delta}{\underline{\delta}^{5/2}T}\right).$$

*Consequently, the computation complexity to reach an $\varepsilon$-accurate solution $\boldsymbol{x}$ such that $\|\nabla f(\boldsymbol{x})\|_2^2 \leq \varepsilon$ is $\mathcal{O}\left(\frac{L\Delta}{\underline{\delta}^{5/2}\varepsilon} + \frac{1}{\underline{\delta}}\right)$.*

### C.5 CONVERGENCE OF LARGE-BATCH GASARE

In this subsection, we present the convergence properties of GaSare with large-batch stochastic gradients. The results and proofs are similar to those of large-batch GaLore in Appendix B.4.

**Lemma 15** (Momentum contraction). *Under Assumption 3, in large-batch GaSare using MSGD (Alg. 5), if $0 < \beta_1 \leq 1$, term $\tilde{\boldsymbol{M}}_\ell^{(t)}$ has the following contraction properties:*

- *When $t = 0$, it holds that*

$$\mathbb{E}[\|\tilde{\boldsymbol{M}}_\ell^{(0)} - \nabla_\ell f(\boldsymbol{X}^{(0)})\|_F^2] \leq 2(\tau-1)(1-\delta_\ell\beta_1)\sum_{r=0}^{\tau-2}\mathbb{E}[\|\nabla_\ell f(\boldsymbol{x}^{(r+1)}) - \nabla_\ell f(\boldsymbol{x}^{(r)})\|_F^2]$$

$$+ \frac{4(1-\delta_\ell\beta_1)}{\tau}\sum_{r=0}^{\tau-1}\mathbb{E}[\|\nabla_\ell f(\boldsymbol{x}^{(r)})\|_F^2] + \frac{4\beta_1\sigma_\ell^2}{\mathcal{B}}; \quad (91)$$

- *When $t = k\tau$, $k \in \mathbb{N}^*$, it holds that*

$$\mathbb{E}[\|\tilde{\boldsymbol{M}}_\ell^{(t)} - \nabla_\ell f(\boldsymbol{x}^{(t)})\|_F^2] - \left(1 - \left(1 - \frac{\delta_\ell}{4}\right)\beta_1\right)\mathbb{E}[\|\tilde{\boldsymbol{M}}_\ell^{(t-1)} - \nabla_\ell f(\boldsymbol{x}^{(t-1)})\|_F^2]$$

$$\leq \frac{4(1-\delta_\ell)}{\tau}\sum_{r=0}^{\tau-1}\mathbb{E}[\|\nabla_l f(\boldsymbol{x}^{(k\tau+r)})\|_F^2] + \frac{5(1-\beta_1)}{\delta_\ell\beta_1}\mathbb{E}[\|\nabla_\ell f(\boldsymbol{x}^{(t)}) - \nabla_\ell f(\boldsymbol{x}^{(t-1)})\|_F^2]$$

$$+ 2(\tau-1)(1-\delta_\ell)\sum_{r=0}^{\tau-2}\mathbb{E}[\|\nabla_\ell f(\boldsymbol{x}^{(k\tau+r+1)}) - \nabla_\ell f(\boldsymbol{x}^{(k\tau+r)})\|_F^2] + \frac{5\sigma_\ell^2}{\mathcal{B}}; \quad (92)$$

- *When $t = k\tau + r$, $k \in \mathbb{N}$, $1 \le r < \tau$, it holds that*

$$\mathbb{E}[\|\tilde{\boldsymbol{M}}_\ell^{(t)} - \nabla_\ell f(\boldsymbol{x}^{(t)})\|_F^2] - \left(1 - \left(1 - \frac{\delta_\ell}{4}\right)\beta_1\right)\mathbb{E}[\|\tilde{\boldsymbol{M}}_\ell^{(t-1)} - \nabla_\ell f(\boldsymbol{x}^{(t-1)})\|_F^2]$$

$$\le \left(1 - \frac{\delta_\ell}{2}\right)\beta_1 \mathbb{E}[\|\nabla_\ell f(\boldsymbol{x}^{(t)})\|_F^2] + \frac{5(1-\beta_1)}{\delta_\ell \beta_1}\mathbb{E}[\|\nabla_\ell f(\boldsymbol{x}^{(t)}) - \nabla_\ell f(\boldsymbol{x}^{(t-1)})\|_F^2]$$

$$+ \frac{15r\beta_1}{\delta_\ell}\sum_{i=1}^{r}\mathbb{E}[\|\nabla_\ell f(\boldsymbol{x}^{(k\tau+i)}) - \nabla_\ell f(\boldsymbol{x}^{(k\tau+i-1)})\|_F^2] + \left(\frac{11\beta_1}{\delta_\ell \mathcal{B}} + \beta_1^2\right)\sigma_\ell^2. \quad (93)$$

*Proof.* For convenience we use $\boldsymbol{E}$ to denote $\boldsymbol{E}_{m_\ell, n_\ell}$. When $t = 0$, we have

$$\mathbb{E}[\|\tilde{\boldsymbol{M}}_\ell^{(0)} - \nabla_\ell f(\boldsymbol{x}^{(0)})\|_F^2]$$

$$= \mathbb{E}[\|\beta_1 \boldsymbol{S}_\ell^{(0)} \odot \boldsymbol{G}_\ell^{(0)} - \nabla_\ell f(\boldsymbol{x}^{(0)})\|_F^2]$$

$$= \mathbb{E}[\|\beta_1(\boldsymbol{S}_\ell^{(0)} - \boldsymbol{E}) \odot \boldsymbol{G}_\ell^{(0)} + \beta_1(\boldsymbol{G}_\ell^{(0)} - \nabla_\ell f(\boldsymbol{x}^{(0)})) - (1-\beta_1)\nabla_\ell f(\boldsymbol{x}^{(0)})\|_F^2]$$

$$\le \beta_1 \mathbb{E}[\|(\boldsymbol{S}_\ell^{(0)} - \boldsymbol{E}) \odot \boldsymbol{G}_\ell^{(0)} + \boldsymbol{G}_\ell^{(0)} - \nabla_\ell f(\boldsymbol{x}^{(0)})\|_F^2] + (1-\beta_1)\|\nabla_\ell f(\boldsymbol{x}^{(0)})\|_F^2, \quad (94)$$

where the inequality uses Jensen's inequality. For the first term we have

$$\mathbb{E}[\|(\boldsymbol{S}_\ell^{(0)} - \boldsymbol{E}) \odot \boldsymbol{G}_\ell^{(0)} + \boldsymbol{G}_\ell^{(0)} - \nabla_\ell f(\boldsymbol{x}^{(0)})\|_F^2]$$

$$\le 2\mathbb{E}[\|(\boldsymbol{E} - \boldsymbol{S}_\ell^{(0)}) \odot \boldsymbol{G}_\ell^{(0)}\|_F^2] + 2\mathbb{E}[\|\boldsymbol{G}_\ell^{(0)} - \nabla_\ell f(\boldsymbol{x}^{(0)})\|_F^2]$$

$$\le 2(1 - \delta_\ell)\mathbb{E}[\|\boldsymbol{G}_\ell\|_F^2] + 2\mathbb{E}[\|\boldsymbol{G}_\ell^{(0)} - \nabla_\ell f(\boldsymbol{x}^{(0)})\|_F^2]$$

$$\le 2(1 - \delta_\ell)\|\nabla_\ell f(\boldsymbol{x}^{(0)})\|_F^2 + \frac{(4 - 2\delta_\ell)\sigma_\ell^2}{\mathcal{B}}, \quad (95)$$

where the first inequality uses Cauchy's inequality, the second inequality uses Lemma 12, the third inequality uses $\mathbb{E}[\|\boldsymbol{G}_\ell^{(0)} - \nabla_\ell f(\boldsymbol{x}^{(0)})\|_F^2] \le \sigma_\ell^2/\mathcal{B}$ (Assumption 3). Applying (95) and Lemma 2 to (94) yields (91).

When $t = k\tau$, $k \in \mathbb{N}^*$, we have

$$\mathbb{E}[\|\tilde{\boldsymbol{M}}_\ell^{(t)} - \nabla_\ell f(\boldsymbol{x}^{(t)})\|_F^2]$$

$$= \mathbb{E}[\|\boldsymbol{S}_\ell^{(t)} \odot [(1-\beta_1)\tilde{\boldsymbol{M}}_\ell^{(t-1)} + \beta_1 \boldsymbol{G}_\ell^{(t)} - \nabla_\ell f(\boldsymbol{x}^{(t)})] - (\boldsymbol{E} - \boldsymbol{S}_\ell^{(t)}) \odot \nabla_\ell f(\boldsymbol{x}^{(t)})\|_F^2]$$

$$= \mathbb{E}[\|\boldsymbol{S}_\ell^{(t)} \odot [(1-\beta_1)\tilde{\boldsymbol{M}}_\ell^{(t-1)} + \beta_1 \boldsymbol{G}_\ell^{(t)} - \nabla_\ell f(\boldsymbol{x}^{(t)})]\|_F^2] + \mathbb{E}[\|(\boldsymbol{E} - \boldsymbol{S}_\ell^{(t)}) \odot \nabla_\ell f(\boldsymbol{x}^{(t)})\|_F^2]. \quad (96)$$

We further have

$$\mathbb{E}[\|\boldsymbol{S}_\ell^{(t)} \odot [(1-\beta_1)\tilde{\boldsymbol{M}}_\ell^{(t-1)} + \beta_1 \boldsymbol{G}_\ell^{(t)} - \nabla_\ell f(\boldsymbol{x}^{(t)})]\|_F^2]$$

$$\le \mathbb{E}[\|(1-\beta_1)\tilde{\boldsymbol{M}}_\ell^{(t-1)} + \beta_1 \boldsymbol{G}_\ell^{(t)} - \nabla_\ell f(\boldsymbol{x}^{(t)})\|_F^2]$$

$$= \mathbb{E}[\|(1-\beta_1)(\tilde{\boldsymbol{M}}_\ell^{(t-1)} - \nabla_\ell f(\boldsymbol{x}^{(t)})) + \beta_1(\boldsymbol{G}_\ell^{(t)} - \nabla_\ell f(\boldsymbol{x}^{(t)}))\|_F^2]$$

$$\le \mathbb{E}[\|(1-\beta_1)(\tilde{\boldsymbol{M}}_\ell^{(t-1)} - \nabla_\ell f(\boldsymbol{x}^{(t)}))\|_F^2] + \beta_1^2 \mathbb{E}[\|\boldsymbol{G}_\ell^{(t)} - \nabla_\ell f(\boldsymbol{x}^{(t)})\|_F^2], \quad (97)$$

where the last inequality uses the unbiasedness of $\boldsymbol{G}_\ell^{(t)}$ (Assumption 3). By Young's inequality, we have

$$\mathbb{E}[\|\tilde{\boldsymbol{M}}_\ell^{(t-1)} - \nabla_\ell f(\boldsymbol{x}^{(t)})\|_F^2]$$

$$= \mathbb{E}[\|(\tilde{\boldsymbol{M}}_\ell^{(t-1)} - \nabla_\ell f(\boldsymbol{x}^{(t-1)})) - (\nabla_\ell f(\boldsymbol{x}^{(t)}) - \nabla_\ell f(\boldsymbol{x}^{(t-1)})\|_F^2]$$

$$\le \left(1 + \frac{\delta_\ell \beta_1}{4}\right)\mathbb{E}[\|\tilde{\boldsymbol{M}}_\ell^{(t-1)} - \nabla_\ell f(\boldsymbol{x}^{(t-1)})\|_F^2] + \left(1 + \frac{4}{\delta_\ell \beta_1}\right)\mathbb{E}[\|\nabla_\ell f(\boldsymbol{x}^{(t)}) - \nabla_\ell f(\boldsymbol{x}^{(t-1)})\|_F^2]. \quad (98)$$

Applying (98) to (97) yields

$$\mathbb{E}[\|\boldsymbol{S}_\ell^{(t)} \odot [(1-\beta_1)\tilde{\boldsymbol{M}}_\ell^{(t-1)} + \beta_1 \boldsymbol{G}_\ell^{(t)} - \nabla_\ell f(\boldsymbol{x}^{(t)})]\|_F^2]$$

$$\leq \left(1 - \left(1 - \frac{\delta_\ell}{4}\right)\beta_1\right)\mathbb{E}[\|\tilde{\boldsymbol{M}}_\ell^{(t-1)} - \nabla_\ell f(\boldsymbol{x}^{(t-1)})\|_F^2] + \frac{\beta_1^2 \sigma^2}{\mathcal{B}}$$

$$+ \frac{5(1-\beta_1)}{\delta_\ell \beta_1}\mathbb{E}[\|\nabla_\ell f(\boldsymbol{x}^{(t)}) - \nabla_\ell f(\boldsymbol{x}^{(t-1)})\|_F^2]. \tag{99}$$

For the second term in (96), we have

$$\mathbb{E}[\|(\boldsymbol{E} - \boldsymbol{S}_\ell^{(t)}) \odot \nabla_\ell f(\boldsymbol{x}^{(t)})\|_F^2]$$

$$\leq 2\mathbb{E}[\|(\boldsymbol{E} - \boldsymbol{S}_\ell^{(t)}) \odot \boldsymbol{G}_\ell^{(t)}\|_F^2] + 2\mathbb{E}[\|(\boldsymbol{E} - \boldsymbol{S}_\ell^{(t)}) \odot (\boldsymbol{G}_\ell^{(t)} - \nabla_\ell f(\boldsymbol{x}^{(t)}))\|_F^2]$$

$$\leq 2(1-\delta_\ell)\mathbb{E}[\|\boldsymbol{G}_\ell^{(t)}\|_F^2] + 2\mathbb{E}[\|\boldsymbol{G}_\ell^{(t)} - \nabla_\ell f(\boldsymbol{x}^{(t)})\|_F^2]$$

$$\leq 2(1-\delta_\ell)\mathbb{E}[\|\nabla_\ell f(\boldsymbol{x}^{(t)})\|_F^2] + \frac{4\sigma_\ell^2}{\mathcal{B}}, \tag{100}$$

where the first inequality uses Cauchy's inequality, the second inequality uses Lemma 12, the third inequality uses Assumption 3. Applying (99)(100) to (96) and using Lemma 2 yields (92).

When $t = k\tau + r$, $k \in \mathbb{N}$, $1 \leq r < \tau$, we have

$$\mathbb{E}[\|\tilde{\boldsymbol{M}}_\ell^{(t)} - \nabla_\ell f(\boldsymbol{x}^{(t)})\|_F^2]$$

$$= \mathbb{E}[\|(1-\beta_1)(\tilde{\boldsymbol{M}}_\ell^{(t-1)} - \nabla_\ell f(\boldsymbol{x}^{(t)})) + \beta_1(\boldsymbol{S}_\ell^{(t)} \odot \boldsymbol{G}_\ell^{(t)} - \nabla_\ell f(\boldsymbol{x}^{(t)}))\|_F^2]$$

$$= \mathbb{E}[\|(1-\beta_1)(\tilde{\boldsymbol{M}}_\ell^{(t-1)} - \nabla_\ell f(\boldsymbol{x}^{(t)})) + \beta_1(\boldsymbol{S}_\ell^{(t)} - \boldsymbol{E}) \odot \nabla_\ell f(\boldsymbol{x}^{(t)})\|_F^2]$$

$$+ \beta_1^2 \mathbb{E}[\|\boldsymbol{S}_\ell^{(t)} \odot (\boldsymbol{G}_\ell^{(t)} - \nabla_\ell f(\boldsymbol{x}^{(t)}))\|_F^2]$$

$$\leq (1-\beta_1)\mathbb{E}[\|\tilde{\boldsymbol{M}}_\ell^{(t-1)} - \nabla_\ell f(\boldsymbol{x}^{(t)})\|_F^2] + \beta_1 \mathbb{E}[\|(\boldsymbol{E} - \boldsymbol{S}_\ell^{(t)}) \odot \nabla_\ell f(\boldsymbol{x}^{(t)})\|_F^2$$

$$+ \beta_1^2 \mathbb{E}[\|\boldsymbol{S}_\ell^{(t)} \odot (\boldsymbol{G}_\ell^{(t)} - \nabla_\ell f(\boldsymbol{x}^{(t)}))\|_F^2], \tag{101}$$

where the second equality uses the unbiasedness of $\boldsymbol{G}_\ell^{(t)}$ and the independence implied by $\boldsymbol{S}_\ell^{(t)} = \boldsymbol{S}_\ell^{(t-1)}$, the inequality uses Jensen's inequality. The first term is similarly bounded as (98). For the second term, we have

$$\mathbb{E}[\|(\boldsymbol{E} - \boldsymbol{S}_\ell^{(k\tau)}) \odot \nabla_\ell f(\boldsymbol{x}^{(t)})\|_F^2]$$

$$\leq \left(1 + \frac{\delta_\ell}{4}\right)\mathbb{E}[\|(\boldsymbol{E} - \boldsymbol{S}_\ell^{(k\tau)}) \odot \boldsymbol{G}_\ell^{(k\tau)}\|_F^2]$$

$$+ \left(1 + \frac{4}{\delta_\ell}\right)\mathbb{E}[\|(\boldsymbol{E} - \boldsymbol{S}_\ell^{(k\tau)}) \odot (\nabla_\ell f(\boldsymbol{x}^{(t)}) - \boldsymbol{G}_\ell^{(k\tau)})\|_F^2]$$

$$\leq \left(1 - \frac{3\delta_\ell}{4}\right)\mathbb{E}[\|\boldsymbol{G}_\ell^{(k\tau)}\|_F^2] + 2\left(1 + \frac{4}{\delta_\ell}\right)\mathbb{E}[\|\boldsymbol{G}_\ell^{(k\tau)} - \nabla_\ell f(\boldsymbol{x}^{(k\tau)})\|_F^2]$$

$$+ 2\left(1 + \frac{4}{\delta_\ell}\right)\mathbb{E}[\|\nabla_\ell f(\boldsymbol{x}^{(t)}) - \nabla_\ell f(\boldsymbol{x}^{(k\tau)})\|_F^2], \tag{102}$$

where the first inequality uses Young's inequality, the second inequality uses Lemma 12 and Cauchy's inequality. We further have

$$\left(1 - \frac{3\delta_\ell}{4}\right)\mathbb{E}[\|\boldsymbol{G}_\ell^{(k\tau)}\|_F^2] + 2\left(1 + \frac{4}{\delta_\ell}\right)\mathbb{E}[\|\boldsymbol{G}_\ell^{(k\tau)} - \nabla_\ell f(\boldsymbol{x}^{(k\tau)})\|_F^2]$$

$$\leq \left(1 - \frac{3\delta_\ell}{4}\right)\mathbb{E}[\|\nabla_\ell f(\boldsymbol{x}^{(k\tau)})\|_F^2] + \frac{11}{\delta_\ell}\mathbb{E}[\|\boldsymbol{G}_\ell^{(k\tau)} - \nabla_\ell f(\boldsymbol{x}^{(k\tau)})\|_F^2]$$

$$\leq \left(1 - \frac{3\delta_\ell}{4}\right)\mathbb{E}[\|\nabla_\ell f(\boldsymbol{x}^{(k\tau)})\|_F^2] + \frac{11\sigma_\ell^2}{\delta_\ell \mathcal{B}}$$

$$\leq \left(1 - \frac{\delta_\ell}{2}\right)\mathbb{E}[\|\nabla_\ell f(\boldsymbol{x}^{(t)})\|_F^2] + \left(1 + \frac{4}{\delta_\ell}\right)\mathbb{E}[\|\nabla_\ell f(\boldsymbol{x}^{(t)}) - \nabla_\ell f(\boldsymbol{x}^{(k\tau)})\|_F^2] + \frac{11\sigma_\ell^2}{\delta_\ell \mathcal{B}}, \tag{103}$$

where the first inequality uses unbiasedness of $\boldsymbol{G}_\ell^{(k\tau)}$, the second inequality uses Assumption 3, the third inequality uses Young's inequality.

Applying (103) to (102) and applying Cauchy's inequality yields

$$
\begin{aligned}
&\mathbb{E}[\|(\boldsymbol{E} - \boldsymbol{S}_\ell^{(k\tau)}) \odot \nabla_\ell f(\boldsymbol{x}^{(t)})\|_F^2] \\
&\leq \left(1 - \frac{\delta_\ell}{2}\right) \mathbb{E}[\|\nabla_\ell f(\boldsymbol{x}^{(t)})\|_F^2] + \frac{11\sigma_\ell^2}{\delta_\ell \mathcal{B}} + \frac{15r}{\delta_\ell} \sum_{i=1}^{r} \mathbb{E}[\|\nabla_\ell f(\boldsymbol{x}^{(k\tau+i)}) - \nabla_\ell f(\boldsymbol{x}^{(k\tau+i-1)})\|_F^2].
\end{aligned}
$$
(104)

For the third term, we have

$$
\mathbb{E}[\|\boldsymbol{S}_\ell^{(k\tau)} \odot (\boldsymbol{G}_\ell^{(t)} - \nabla_\ell f(\boldsymbol{x}^{(t)}))\|_F^2] \leq \mathbb{E}[\|\boldsymbol{G}_\ell^{(t)} - \nabla_\ell f(\boldsymbol{x}^{(t)})\|_F^2] \leq \sigma_\ell^2,
$$
(105)

where the second inequality uses Assumption 3.

Applying (98)(104)(105) to (101) yields (93). □

Based on Lemma 15, we can prove the convergence properties of large-batch GaSare similarly as the proofs of Lemma 9, Theorem 7 and Corollary 2. Below we directly present the final convergence results.

**Theorem 11** (Convergence of large-batch GaSare). *Under Assumptions 1-3, if hyperparameters*

$$
0 < \beta_1 \leq 1, \quad \tau \geq \frac{128}{3\beta_1\underline{\delta}}, \quad 0 < \eta \leq \min\left\{\frac{1}{4L}, \sqrt{\frac{3\underline{\delta}\beta_1^2}{80L^2}}, \sqrt{\frac{\underline{\delta}}{40\tau^2 L^2}}, \sqrt{\frac{3\beta_1}{32\tau L^2}}\right\},
$$

*GaSare using large-batch stochastic gradients and MSGD (Alg. 5) converges as*

$$
\frac{1}{K\tau} \sum_{t=0}^{K\tau-1} \mathbb{E}\|\nabla f(\boldsymbol{x}^{(t)})\|_2^2 \leq \frac{16\Delta}{\underline{\delta}\eta K\tau} + \left(\frac{160}{3\beta_1\underline{\delta}\tau\mathcal{B}} + \frac{352}{3\underline{\delta}^2\mathcal{B}} + \frac{32\beta_1}{3\underline{\delta}}\right)\sigma^2
$$

*for any $K \geq 1$, where $\Delta = f(\boldsymbol{x}^{(0)}) - \inf_{\boldsymbol{x}} f(\boldsymbol{x})$. If $T \geq 2 + 256/(3\underline{\delta}) + (256\sigma)^2/(9\sqrt{\underline{\delta}}L\Delta)$ and we further choose*

$$
\beta_1 = \left(1 + \sqrt{\frac{\underline{\delta}^{3/2}\sigma^2 T}{L\Delta}}\right)^{-1},
$$

$$
\tau = \left\lceil\frac{128}{3\underline{\delta}\beta_1}\right\rceil,
$$

$$
\eta = \left(4L + \sqrt{\frac{80L^2}{3\underline{\delta}\beta_1^2}} + \sqrt{\frac{40\tau^2 L^2}{\underline{\delta}}} + \sqrt{\frac{32\tau L^2}{3\beta_1}}\right)^{-1},
$$

$$
\mathcal{B} = \left\lceil\frac{1}{\underline{\delta}\beta_1}\right\rceil,
$$

*GaSare using large-batch stochastic gradients and MSGD (Alg. 5) converges as*

$$
\frac{1}{T} \sum_{t=0}^{T-1} \mathbb{E}[\|\nabla f(\boldsymbol{x}^{(t)})\|_2^2] = \mathcal{O}\left(\frac{L\Delta}{\underline{\delta}^{5/2}T} + \sqrt{\frac{L\Delta\sigma^2}{\underline{\delta}^{7/2}T}}\right).
$$

*Consequently, the computation complexity to reach an $\varepsilon$-accurate solution $\boldsymbol{x}$ such that $\|\nabla f(\boldsymbol{x})\|_2^2 \leq \varepsilon$ is $\mathcal{O}\left(\frac{L\Delta\sigma^2}{\underline{\delta}^{7/2}\varepsilon^2} + \frac{L\Delta}{\underline{\delta}^{5/2}\varepsilon} + \frac{\sigma^2}{\underline{\delta}^{1/2}L\Delta} + \frac{1}{\underline{\delta}}\right)$.*

## C.6 CONVERGENCE OF GOSARE

In this subsection, we present the convergence properties of GoSare with small-batch stochastic gradients. The results and proofs are similar to those of GoLore in Appendix B.5.

**Lemma 16** (Momentum contraction). *Under Assumption 3, in GoSare using MSGD (Alg. 5), if $0 < \beta_1 \leq 1$, term $\tilde{\boldsymbol{M}}_\ell^{(t)}$ has the following contraction properties:*

- *When $t = 0$, it holds that*

$$
\mathbb{E}[\|\tilde{\boldsymbol{M}}_\ell^{(0)} - \nabla_\ell f(\boldsymbol{X}^{(0)})\|_F^2] \leq (\tau-1)(1-\delta_\ell\beta_1) \sum_{r=0}^{\tau-2} \mathbb{E}[\|\nabla_\ell f(\boldsymbol{x}^{(r+1)}) - \nabla_\ell f(\boldsymbol{x}^{(r)})\|_F^2]
$$

$$
+ \frac{2(1-\delta_\ell\beta_1)}{\tau} \sum_{r=0}^{\tau-1} \mathbb{E}[\|\nabla_\ell f(\boldsymbol{x}^{(r)})\|_F^2] + \delta_\ell\beta_1^2\sigma_\ell^2; \quad (106)
$$

- *When $t = k\tau$, $k \in \mathbb{N}^*$, it holds that*

$$
\mathbb{E}[\|\tilde{\boldsymbol{M}}_\ell^{(t)} - \nabla_\ell f(\boldsymbol{x}^{(t)})\|_F^2] - \delta_\ell\left(1 - \left(1 - \frac{\delta_\ell}{4}\right)\beta_1\right) \mathbb{E}[\|\tilde{\boldsymbol{M}}_\ell^{(t-1)} - \nabla_\ell f(\boldsymbol{x}^{(t-1)})\|_F^2]
$$

$$
\leq \frac{2(1-\delta_\ell)}{\tau} \sum_{r=0}^{\tau-1} \mathbb{E}[\|\nabla_l f(\boldsymbol{x}^{(k\tau+r)})\|_F^2] + \frac{5(1-\beta_1)}{\beta_1} \mathbb{E}[\|\nabla_\ell f(\boldsymbol{x}^{(t)}) - \nabla_\ell f(\boldsymbol{x}^{(t-1)})\|_F^2]
$$

$$
+ (\tau-1)(1-\delta_\ell) \sum_{r=0}^{\tau-2} \mathbb{E}[\|\nabla_\ell f(\boldsymbol{x}^{(k\tau+r+1)}) - \nabla_\ell f(\boldsymbol{x}^{(k\tau+r)})\|_F^2] + \delta_\ell\beta_1^2\sigma_\ell^2; \quad (107)
$$

- *When $t = k\tau + r$, $k \in \mathbb{N}$, $1 \leq r < \tau$, it holds that*

$$
\mathbb{E}[\|\tilde{\boldsymbol{M}}_\ell^{(t)} - \nabla_\ell f(\boldsymbol{x}^{(t)})\|_F^2] - \left(1 - \left(1 - \frac{\delta_\ell}{4}\right)\beta_1\right) \mathbb{E}[\|\tilde{\boldsymbol{M}}_\ell^{(t-1)} - \nabla_\ell f(\boldsymbol{x}^{(t-1)})\|_F^2]
$$

$$
\leq \left(1 - \frac{\delta_\ell}{2}\right)\beta_1\mathbb{E}[\|\nabla_\ell f(\boldsymbol{x}^{(t)})\|_F^2] + \frac{5(1-\beta_1)}{\delta_\ell\beta_1}\mathbb{E}[\|\nabla_\ell f(\boldsymbol{x}^{(t)}) - \nabla_\ell f(\boldsymbol{x}^{(t-1)})\|_F^2]
$$

$$
+ \frac{10r\beta_1}{\delta_\ell} \sum_{i=1}^{r} \mathbb{E}[\|\nabla_\ell f(\boldsymbol{x}^{(k\tau+i)}) - \nabla_\ell f(\boldsymbol{x}^{(k\tau+i-1)})\|_F^2] + \beta_1^2\sigma_\ell^2. \quad (108)
$$

*Proof.* For convenience we use $\boldsymbol{E}$ to denote $\boldsymbol{E}_{m_\ell,n_\ell}$. When $t = 0$, we have

$$
\mathbb{E}[\|\tilde{\boldsymbol{M}}_\ell^{(0)} - \nabla_\ell f(\boldsymbol{x}^{(0)})\|_F^2]
$$

$$
= \mathbb{E}[\|\beta_1\boldsymbol{S}_\ell^{(0)} \odot \boldsymbol{G}_\ell^{(0)} - \nabla_\ell f(\boldsymbol{x}^{(0)})\|_F^2]
$$

$$
= \mathbb{E}[\|(\beta_1\boldsymbol{S}_\ell^{(0)} - \boldsymbol{E}) \odot \nabla_\ell f(\boldsymbol{x}^{(0)})\|_F^2] + \beta_1^2\mathbb{E}[\|\boldsymbol{S}_\ell^{(0)} \odot (\boldsymbol{G}_\ell^{(0)} - \nabla_\ell f(\boldsymbol{x}^{(0)}))\|_F^2], \quad (109)
$$

where the second equality uses unbiasedness of $\boldsymbol{G}_\ell^{(0)}$. By Lemma 5 we have

$$
\mathbb{E}[\|(\beta_1\boldsymbol{S}_\ell^{(0)} - \boldsymbol{E}) \odot \nabla_\ell f(\boldsymbol{x}^{(0)})\|_F^2
$$

$$
= \sum_{1 \leq i \leq m_\ell, 1 \leq j \leq n_\ell} \mathbb{E}[(\beta_1[S_\ell^{(0)}]_{i,j} - 1)^2][\nabla_\ell f(\boldsymbol{x}^{(0)})]_{i,j}^2
$$

$$
= \sum_{1 \leq i \leq m_\ell, 1 \leq j \leq n_\ell} (1 - 2\beta_1\delta_\ell + \beta_1^2\delta_\ell)[\nabla_\ell f(\boldsymbol{x}^{(0)})]_{i,j}^2
$$

$$
\leq (1 - \delta_\ell\beta_1)\|\nabla_\ell f(\boldsymbol{x}^{(0)})\|_F^2. \quad (110)
$$

Similarly, by Lemma 5 we have

$$
\mathbb{E}[\|\boldsymbol{S}_\ell^{(0)} \odot (\boldsymbol{G}_\ell^{(0)} - \nabla_\ell f(\boldsymbol{x}^{(0)}))\|_F^2]
$$

$$
= \sum_{1 \leq i \leq m_\ell, 1 \leq j \leq n_\ell} \mathbb{E}[[S_\ell^{(0)}]_{i,j}^2][\boldsymbol{G}_\ell^{(0)} - \nabla_\ell f(\boldsymbol{x}^{(0)})]_{i,j}^2
$$

$$
= \delta_\ell\mathbb{E}[\|\boldsymbol{G}_\ell^{(0)} - \nabla_\ell f(\boldsymbol{x}^{(0)})\|_F^2]
$$

$$
\leq \delta_\ell\sigma_\ell^2, \quad (111)
$$

where the inequality uses Assumption 3. Applying (110)(111) and Lemma 2 to (109) yields (106).

When $t = k\tau$, $k \in \mathbb{N}^*$, we have

$$\mathbb{E}[\|\tilde{\boldsymbol{M}}_\ell^{(t)} - \nabla_\ell f(\boldsymbol{x}^{(t)})\|_F^2]$$
$$= \mathbb{E}[\|\boldsymbol{S}_\ell^{(t)} \odot [(1 - \beta_1)\tilde{\boldsymbol{M}}_\ell^{(t-1)} + \beta_1 \boldsymbol{G}_\ell^{(t)} - \nabla_\ell f(\boldsymbol{x}^{(t)})] - (\boldsymbol{E} - \boldsymbol{S}_\ell^{(t)}) \odot \nabla_\ell f(\boldsymbol{x}^{(t)})\|_F^2]$$
$$= \delta_\ell \mathbb{E}[\|(1 - \beta_1)\tilde{\boldsymbol{M}}_\ell^{(t-1)} + \beta_1 \boldsymbol{G}_\ell^{(t)} - \nabla_\ell f(\boldsymbol{x}^{(t)})\|_F^2] + (1 - \delta_\ell)\mathbb{E}[\|\nabla_\ell f(\boldsymbol{x}^{(t)})\|_F^2], \qquad (112)$$

where the second equality uses Lemma 13. For the first term, we have

$$\mathbb{E}[\|(1 - \beta_1)\tilde{\boldsymbol{M}}_\ell^{(t-1)} + \beta_1 \boldsymbol{G}_\ell^{(t)} - \nabla_\ell f(\boldsymbol{x}^{(t)})\|_F^2]$$
$$= \mathbb{E}[\|(1 - \beta_1)(\tilde{\boldsymbol{M}}_\ell^{(t-1)} - \nabla_\ell f(\boldsymbol{x}^{(t)})) + \beta_1(\boldsymbol{G}_\ell^{(t)} - \nabla_\ell f(\boldsymbol{x}^{(t)}))\|_F^2]$$
$$\leq \mathbb{E}[\|(1 - \beta_1)(\tilde{\boldsymbol{M}}_\ell^{(t-1)} - \nabla_\ell f(\boldsymbol{x}^{(t)}))\|_F^2] + \beta_1^2 \mathbb{E}[\|\boldsymbol{G}_\ell^{(t)} - \nabla_\ell f(\boldsymbol{x}^{(t)})\|_F^2]$$
$$\leq (1 - \beta_1)\mathbb{E}[\|\tilde{\boldsymbol{M}}_\ell^{(t-1)} - \nabla_\ell f(\boldsymbol{x}^{(t)})\|_F^2] + \beta_1^2 \sigma_\ell^2, \qquad (113)$$

where both inequalities use Assumption 3. By Young's inequality, we have

$$\mathbb{E}[\|\tilde{\boldsymbol{M}}_\ell^{(t-1)} - \nabla_\ell f(\boldsymbol{x}^{(t)})\|_F^2]$$
$$= \mathbb{E}[\|(\tilde{\boldsymbol{M}}_\ell^{(t-1)} - \nabla_\ell f(\boldsymbol{x}^{(t-1)})) - (\nabla_\ell f(\boldsymbol{x}^{(t)}) - \nabla_\ell f(\boldsymbol{x}^{(t-1)})\|_F^2]$$
$$\leq \left(1 + \frac{\delta_\ell \beta_1}{4}\right) \mathbb{E}[\|\tilde{\boldsymbol{M}}_\ell^{(t-1)} - \nabla_\ell f(\boldsymbol{x}^{(t-1)})\|_F^2] + \left(1 + \frac{4}{\delta_\ell \beta_1}\right) \mathbb{E}[\|\nabla_\ell f(\boldsymbol{x}^{(t)}) - \nabla_\ell f(\boldsymbol{x}^{(t-1)})\|_F^2].$$
$$(114)$$

Applying (113)(114) and Lemma 2 to (112) yields (107).

When $t = k\tau + r$, $k \in \mathbb{N}$, $1 \leq r < \tau$, we have

$$\mathbb{E}[\|\tilde{\boldsymbol{M}}_\ell^{(t)} - \nabla_\ell f(\boldsymbol{x}^{(t)})\|_F^2]$$
$$= \mathbb{E}[\|(1 - \beta_1)(\tilde{\boldsymbol{M}}_\ell^{(t-1)} - \nabla_\ell f(\boldsymbol{x}^{(t)})) + \beta_1(\boldsymbol{S}_\ell^{(t)} \odot \boldsymbol{G}_\ell^{(t)} - \nabla_\ell f(\boldsymbol{x}^{(t)}))\|_F^2]$$
$$= \mathbb{E}[\|(1 - \beta_1)(\tilde{\boldsymbol{M}}_\ell^{(t-1)} - \nabla_\ell f(\boldsymbol{x}^{(t)})) + \beta_1(\boldsymbol{S}_\ell^{(t)} - \boldsymbol{E}) \odot \nabla_\ell f(\boldsymbol{x}^{(t)})\|_F^2]$$
$$\quad + \beta_1^2 \mathbb{E}[\|\boldsymbol{S}_\ell^{(t)} \odot (\boldsymbol{G}_\ell^{(t)} - \nabla_\ell f(\boldsymbol{x}^{(t)}))\|_F^2]$$
$$\leq (1 - \beta_1)\mathbb{E}[\|\tilde{\boldsymbol{M}}_\ell^{(t-1)} - \nabla_\ell f(\boldsymbol{x}^{(t)})\|_F^2] + \beta_1 \mathbb{E}[\|(\boldsymbol{E} - \boldsymbol{S}_\ell^{(t)}) \odot \nabla_\ell f(\boldsymbol{x}^{(t)})\|_F^2$$
$$\quad + \beta_1^2 \mathbb{E}[\|\boldsymbol{S}_\ell^{(t)} \odot (\boldsymbol{G}_\ell^{(t)} - \nabla_\ell f(\boldsymbol{x}^{(t)}))\|_F^2], \qquad (115)$$

where the second equality uses the unbiasedness of $\boldsymbol{G}_\ell^{(t)}$ and the independence implied by $\boldsymbol{S}_\ell^{(t)} = \boldsymbol{S}_\ell^{(t-1)}$, the inequality uses Jensen's inequality. The first term is similarly bounded as (114). For the second term, we have

$$\mathbb{E}[\|(\boldsymbol{E} - \boldsymbol{S}_\ell^{(k\tau)}) \odot \nabla_\ell f(\boldsymbol{x}^{(t)})\|_F^2]$$
$$\leq \left(1 + \frac{\delta_\ell}{4}\right) \mathbb{E}[\|(\boldsymbol{E} - \boldsymbol{S}_\ell^{(k\tau)}) \odot \nabla_\ell f(\boldsymbol{x}^{(k\tau)})\|_F^2]$$
$$\quad + \left(1 + \frac{4}{\delta_\ell}\right) \mathbb{E}[\|(\boldsymbol{E} - \boldsymbol{S}_\ell^{(k\tau)}) \odot (\nabla_\ell f(\boldsymbol{x}^{(t)}) - \nabla_\ell f(\boldsymbol{x}^{(k\tau)}))\|_F^2]$$
$$\leq \left(1 - \frac{3\delta_\ell}{4}\right) \mathbb{E}[\|\nabla_\ell f(\boldsymbol{x}^{(k\tau)})\|_F^2] + \left(1 + \frac{4}{\delta_\ell}\right) \mathbb{E}[\|\nabla_\ell f(\boldsymbol{x}^{(t)}) - \nabla_\ell f(\boldsymbol{x}^{(k\tau)})\|_F^2], \qquad (116)$$

where the first inequality uses Young's inequality, the second inequality uses Lemma 13. By Young's inequality, we have

$$\mathbb{E}[\|\nabla_\ell f(\boldsymbol{x}^{(k\tau)})\|_F^2] \leq \left(1 + \frac{\delta_\ell}{4}\right) \mathbb{E}[\|\nabla_\ell f(\boldsymbol{x}^{(t)})\|_F^2] + \left(1 + \frac{4}{\delta_\ell}\right) \mathbb{E}[\|\nabla_\ell f(\boldsymbol{x}^{(t)}) - \nabla_\ell f(\boldsymbol{x}^{(k\tau)})\|_F^2].$$
$$(117)$$

Applying (117) to (116) and applying Cauchy's inequality yields

$$\mathbb{E}[\|(\boldsymbol{E} - \boldsymbol{S}_\ell^{(k\tau)}) \odot \nabla_\ell f(\boldsymbol{x}^{(t)})\|_F^2]$$

$$\leq \left(1 - \frac{\delta_\ell}{2}\right) \mathbb{E}[\|\nabla_\ell f(\boldsymbol{x}^{(t)})\|_F^2] + \frac{10r}{\delta_\ell} \sum_{i=1}^{r} \mathbb{E}[\|\nabla_\ell f(\boldsymbol{x}^{(k\tau+i)}) - \nabla_\ell f(\boldsymbol{x}^{(k\tau+i-1)})\|_F^2]. \tag{118}$$

For the third term, we have

$$\mathbb{E}[\|\boldsymbol{S}_\ell^{(k\tau)} \odot (\boldsymbol{G}_\ell^{(t)} - \nabla_\ell f(\boldsymbol{x}^{(t)}))\|_F^2] \leq \mathbb{E}[\|\boldsymbol{G}_\ell^{(t)} - \nabla_\ell f(\boldsymbol{x}^{(t)})\|_F^2] \leq \sigma_\ell^2, \tag{119}$$

where the second inequality uses Assumption 3.

Applying (114)(118)(119) to (115) yields (108). $\qquad\square$

Based on Lemma 16, we can prove the convergence properties of GoSare similarly as the proofs of Lemma 11, Theorem 8 and Corollary 3. Below we directly present the final convergence results.

**Theorem 12** (Convergence of GoSare). *Under Assumptions 1-3, if hyperparameters*

$$0 < \beta_1 \leq 1, \quad \tau \geq \frac{64}{3\beta_1\underline{\delta}}, \quad 0 < \eta \leq \min\left\{\frac{1}{4L}, \sqrt{\frac{3\underline{\delta}\beta_1^2}{80L^2}}, \sqrt{\frac{3\underline{\delta}}{80\tau^2L^2}}, \sqrt{\frac{3\beta_1}{16\tau L^2}}\right\},$$

*GoSare using small-batch stochastic gradients and MSGD (Alg. 5) converges as*

$$\frac{1}{K\tau} \sum_{t=0}^{K\tau-1} \mathbb{E}\|\nabla f(\boldsymbol{x}^{(t)})\|_2^2 \leq \frac{16\Delta}{\underline{\delta}\eta K\tau} + \frac{32\beta_1\sigma^2}{3\underline{\delta}}$$

*for any $K \geq 1$, where $\Delta = f(\boldsymbol{x}^{(0)}) - \inf_{\boldsymbol{x}} f(\boldsymbol{x})$. If $T \geq 2 + 128/(3\underline{\delta}) + (128\sigma)^2/(9\sqrt{\underline{\delta}}L\Delta)$ and we further choose*

$$\beta_1 = \left(1 + \sqrt{\frac{\underline{\delta}^{3/2}\sigma^2 T}{L\Delta}}\right)^{-1},$$

$$\tau = \left\lceil \frac{64}{3\underline{\delta}\beta_1} \right\rceil,$$

$$\eta = \left(4L + \sqrt{\frac{80L^2}{3\underline{\delta}\beta_1^2}} + \sqrt{\frac{80\tau^2L^2}{3\underline{\delta}}} + \sqrt{\frac{16\tau L^2}{3\beta_1}}\right)^{-1},$$

*GoSare using small-batch stochastic gradients and MSGD (Alg. 5) converges as*

$$\frac{1}{T} \sum_{t=0}^{T-1} \mathbb{E}[\|\nabla f(\boldsymbol{x}^{(t)})\|_2^2] = \mathcal{O}\left(\frac{L\Delta}{\underline{\delta}^{5/2}T} + \sqrt{\frac{L\Delta\sigma^2}{\underline{\delta}^{7/2}T}}\right).$$

*Consequently, the computation complexity to reach an $\varepsilon$-accurate solution $\boldsymbol{x}$ such that $\|\nabla f(\boldsymbol{x})\|_2^2 \leq \varepsilon$ is $\mathcal{O}\left(\frac{L\Delta\sigma^2}{\underline{\delta}^{7/2}\varepsilon^2} + \frac{L\Delta}{\underline{\delta}^{5/2}\varepsilon} + \frac{\sigma^2}{\underline{\delta}^{1/2}L\Delta} + \frac{1}{\underline{\delta}}\right)$.*

## D  THE RELORA-LIKE IMPLEMENTATION

An equivalent, ReLoRA-like implementation of Alg. 1 is as illustrated in Alg. 6, where we only present the case with small-batch stochastic gradients for convenience. In fact, applying ReLoRA with a fixed $\boldsymbol{A}$ or $\boldsymbol{B}$ is not our contribution, as it has already been used in several previous works(Hao et al., 2024; Loeschcke et al., 2024). While leading to the same results, this ReLoRA-like implementation (Alg. 6) can potentially save computation as it computes the subspace gradient directly without computing the full-parameter one. Consider the case where $m \leq n$ and we use MSGD and a batch size of $b$. The computation complexity of GaLore's original implementation is $2bmn$ for forward propagation, $4bmn$ for backward propagation, $4rmn$ for projection, $3rn$ for momentum update and $2mn$ for weight update. The computational complexity of our ReLoRA-like implementation is $2bmn + 2brm + 2brn$ for forward propagation, $2bmn + 2brm + 2brn$ for backward propagation, $3rn$ for momentum updates and $2rn$ for weight updates. As illustrated in Table 1, our implementation can potentially reduce computation with little memory overhead.

**Algorithm 6** ReLoRA-like implementation of GaLore / GoLore algorithm using stochastic gradients with / without momentum projection

**Input:** Initial point $\boldsymbol{x}^{(0)}$, data distribution $\mathcal{D}$, learning rate $\eta$, subspace changing frequency $\tau$, rank $\{r_\ell\}_{\ell=1}^{N_L}$, optimizer hyperparameters $\beta_1$, $\beta_2$, $\epsilon$, large batch size $\mathcal{B}$.

**Output:** $\{\boldsymbol{x}^{(t)}\}_{t=0}^T$.

Initialize LoRA adaptation $\boldsymbol{X}_\ell = \boldsymbol{W}_\ell + \boldsymbol{B}_\ell \boldsymbol{A}_\ell$ for $\ell = 1, 2, \cdots, N_L$, where $\boldsymbol{W}_\ell^{(0)} = \boldsymbol{X}_\ell^{(0)}$, $\boldsymbol{A}_\ell^{(0)} = 0$ and $\boldsymbol{B}_\ell^{(0)} = 0$;

Initialize optimizer state $\{\boldsymbol{M}_\ell^{(-1)}\}_{\ell=1}^{N_L}$ and $\{\boldsymbol{V}_\ell^{(-1)}\}_{\ell=1}^{N_L}$ to zero;

**for** $t = 0, 1, \cdots, T - 1$ **do**

  **for** $\ell = 1, 2, \cdots, N_L$ **do**

    **if** $t \equiv 0 \pmod \tau$ **then**

      $\boldsymbol{G}_\ell^{(t)} \leftarrow \nabla_\ell F(\boldsymbol{x}^{(t)}; \xi^{(t)})$;

      $\boldsymbol{U}, \boldsymbol{\Sigma}, \boldsymbol{V} \leftarrow \mathrm{SVD}(\boldsymbol{G}_\ell^{(t)}), \boldsymbol{P}_\ell^{(t)} \leftarrow \boldsymbol{U}[:, : r_\ell], \boldsymbol{Q}_\ell^{(t)} \leftarrow \boldsymbol{V}[:, : r_\ell]$;  (GaLore)

      Sample $\boldsymbol{P}_\ell^{(t)} \sim \mathcal{U}(\mathrm{St}_{m_\ell, r_\ell})$,  $\boldsymbol{Q}_\ell^{(t)} \sim \mathcal{U}(\mathrm{St}_{n_\ell, r_\ell})$;  (GoLore)

      $\boldsymbol{R}_\ell^{(t)} \leftarrow \begin{cases} (\boldsymbol{P}_\ell^{(t)})^\top \boldsymbol{G}_\ell^{(t)}, & \text{if } m_\ell \le n_\ell; \\ \boldsymbol{G}_\ell^{(t)} \boldsymbol{Q}_\ell^{(t)}, & \text{if } m_\ell > n_\ell; \end{cases}$

    **else**

      $\boldsymbol{R}_\ell^{(t)} \leftarrow \begin{cases} \nabla_{\boldsymbol{A}_\ell} F(\boldsymbol{x}^{(t)}; \xi^{(t)}), & \text{if } m_\ell \le n_\ell; \\ \nabla_{\boldsymbol{B}_\ell} F(\boldsymbol{x}^{(t)}; \xi^{(t)}), & \text{if } m_\ell > n_\ell; \end{cases}$

    **end if**

    $\boldsymbol{M}_\ell^{(t)} \leftarrow \begin{cases} (1 - \beta_1)(\boldsymbol{P}_\ell^{(t)})^\top \boldsymbol{B}_\ell^{(t)} \boldsymbol{M}_\ell^{(t-1)} + \beta_1 \boldsymbol{R}_\ell^{(t)}, & \text{if } m_\ell \le n_\ell; \\ (1 - \beta_1) \boldsymbol{M}_\ell^{(t-1)} \boldsymbol{A}_\ell^{(t)} \boldsymbol{Q}_\ell^{(t)} + \beta_1 \boldsymbol{R}_\ell^{(t)}, & \text{if } m_\ell > n_\ell; \end{cases}$  (with MP)

    $\boldsymbol{M}_\ell^{(t)} \leftarrow (1 - \beta_1) \boldsymbol{M}_\ell^{(t-1)} + \beta_1 \boldsymbol{R}_\ell^{(t)}$;  (without MP)

    $\boldsymbol{V}_\ell^{(t)} \leftarrow (1 - \beta_2) \boldsymbol{V}_\ell^{(t-1)} + \beta_2 \boldsymbol{R}_\ell^{(t)} \odot \boldsymbol{R}_\ell^{(t)}$;

    **if** using Adam **then**

      $\boldsymbol{M}_\ell^{(t)} \leftarrow \boldsymbol{M}_\ell^{(t)} / (1 - \beta_1^t)$,  $\boldsymbol{V}_\ell^{(t)} \leftarrow \boldsymbol{V}_\ell^{(t)} / (1 - \beta_2^t)$,  $\boldsymbol{N}_\ell^{(t)} \leftarrow \boldsymbol{M}_\ell^{(t)} / (\sqrt{\boldsymbol{V}_\ell^{(t)}} + \epsilon)$;

    **else if** using MSGD **then**

      $\boldsymbol{N}_\ell^{(t)} \leftarrow \boldsymbol{M}_\ell^{(t)}$;

    **end if**

    **if** $t \equiv 0 \pmod \tau$ **then**

      $\boldsymbol{W}_\ell^{(t+1)} \leftarrow \boldsymbol{W}_\ell^{(t)} + \boldsymbol{B}_\ell^{(t)} \boldsymbol{A}_\ell^{(t)}$;

      $\boldsymbol{A}_\ell^{(t+1)} \leftarrow \begin{cases} -\eta \boldsymbol{N}_\ell^{(t)}, & \text{if } m_\ell \le n_\ell; \\ (\boldsymbol{Q}_\ell^{(t)})^\top, & \text{if } m_\ell > n_\ell; \end{cases}$

      $\boldsymbol{B}_\ell^{(t+1)} \leftarrow \begin{cases} \boldsymbol{P}_\ell^{(t)}, & \text{if } m_\ell \le n_\ell; \\ -\eta \boldsymbol{N}_\ell^{(t)}, & \text{if } m_\ell > n_\ell; \end{cases}$

    **else**

      $\boldsymbol{W}_\ell^{(t+1)} \leftarrow \boldsymbol{W}_\ell^{(t)}$;

      $\boldsymbol{A}_\ell^{(t+1)} \leftarrow \begin{cases} \boldsymbol{A}_\ell^{(t)} - \eta \boldsymbol{N}_\ell^{(t)}, & \text{if } m_\ell \le n_\ell; \\ \boldsymbol{A}_\ell^{(t)}, & \text{if } m_\ell > n_\ell; \end{cases}$

      $\boldsymbol{B}_\ell^{(t+1)} \leftarrow \begin{cases} \boldsymbol{B}_\ell^{(t)}, & \text{if } m_\ell \le n_\ell; \\ \boldsymbol{B}_\ell^{(t)} - \eta \boldsymbol{N}_\ell^{(t)}, & \text{if } m_\ell > n_\ell; \end{cases}$

    **end if**

  **end for**

**end for**

# E    EXPERIMENTAL SPECIFICATIONS

In this section, we elaborate on the missing details concerned with the experiments we present in Sec. 7.

**GaLore's non-convergence.** We compared Galore, large-batch GaLore, GoLore and full-parameter training on the constructed quadratic problem defined in (1). We used a batch size of 128 for large-batch GaLore and a batch size of 1 for the others.

**Pre-training tasks on C4 dataset.** We pre-trained LLaMA-60M on C4 dataset for 10,000 iterations on 4 NVIDIA A100 40G GPUs. We use batch size 128, learning rate 1.0e-3, rank 128, scaling factor $\alpha = 1$, subspace changing frequency $\tau = 200$, and a max sequence length of 256. Results under 8-bit training are shown in Fig. 6. Fig. 7 presents the results of different algorithms after trained on more tokens.

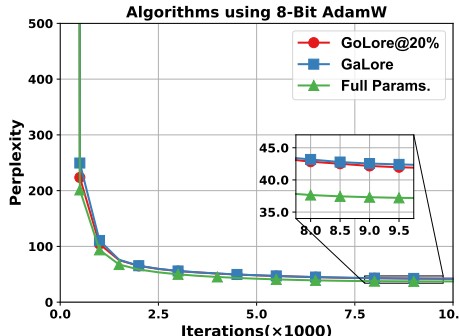
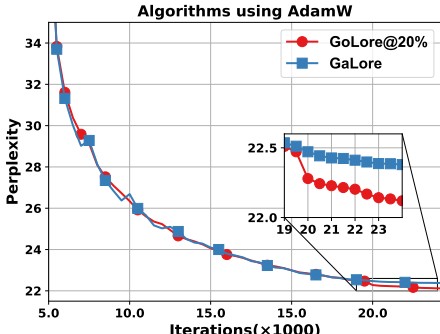

Figure 6: Pre-training curves of algorithms using 8-bit AdamW.

Figure 7: Pre-training curves of algorithms using AdamW.

**Fine-tuning tasks on WinoGrande dataset.** We fine-tune pre-trained LLaMA2-7B model on the WinoGrande dataset for 30 epochs on 4 NVIDIA A100 80G GPUs. We use batch size 1, rank 1024, subspaces changing frequency $\tau = 500$ and a max sequence length of 2048. The learning rate and scaling factor are set as 1.0e-4 and $\alpha = 4$ for GaLore/GoLore, thus corresponds to a learning rate of 4.0e-4 in full-parameter fine-tuning. The test accuracy is presented in Table 3, where GoLore performs similarly to GaLore due to overfitting.

**Fine-tuning tasks on BoolQ dataset.** We fine-tune pre-trained LLaMA2-7B model on the BoolQ (Clark et al., 2019) dataset on 4 NVIDIA A100 80G GPUs. We use batch size 1, rank 1024, subspaces changing frequency $\tau = 500$ and a max sequence length of 2048. We use MSGD as the subspace optimizer, where the learning rate and scaling factor are set as 1.0e-4 and $\alpha = 4$ for GaLore/GoLore, corresponding to a learning rate of 4.0e-4 in full-parameter fine-tuning. Table 3 presents the test accuracy of different algorithms, where GoLore outperforms GaLore. We further fine-tune pre-trained OPT-13B (Zhang et al., 2022) for 1 epoch using the same experimental setup, whose results are shown in Table 4.

Table 3: Evaluating GaLore/GoLore for fine-tuning on WinoGrande and BoolQ using pre-trained LLaMA2-7B.

| Algorithm | BoolQ (1 epoch) | BoolQ (3 epochs) | WinoGrande (80 epochs) |
|---|---|---|---|
| Full Params. | 86.48 | 87.43 | 69.85 |
| GaLore | 84.89 | 86.79 | **68.51** |
| GoLore@20% | **85.81** | **86.88** | **68.51** |

Table 4: Results for fine-tuning pre-trained OPT-13B models on BoolQ. *OOM* stands for "out of memory".

| Algorithm | Memory | Accuracy |
|---|---|---|
| Full Params. | OOM | - |
| GaLore | 77.68 GB | 79.79 |
| GoLore@30% | 77.68 GB | **81.96** |

**Fine-tuning tasks on GLUE benchmark.** We fine-tune pre-trained RoBERTa-Base model on the GLUE benchmark for 30 epochs on a single GeForce RTX 4090. Training details including batch size, learning rate, rank, scaling factor $\alpha$ and max sequence length are illustrated in Table 5.

Table 5: Hyperparameters used in fine-tuning pre-trained RoBERTa-Base model on the GLUE benchmark.

| Hyperparameter | CoLA | STS-B | MRPC | RTE | SST2 | MNLI | QNLI | QQP |
|---|---|---|---|---|---|---|---|---|
| batch size | 32 | 16 | 16 | 16 | 16 | 16 | 16 | 16 |
| Learning Rate | 2.5e-5 | 2.0e-5 | 3.5e-5 | 7.0e-6 | 1.0e-5 | 1.0e-5 | 1.0e-5 | 1.0e-5 |
| Rank | 4 | 4 | 4 | 4 | 4 | 4 | 4 | 4 |
| GaLore's $\alpha$ | 4 | 4 | 4 | 4 | 4 | 4 | 4 | 4 |
| FLORA'S $\alpha$ | 4 | 4 | 4 | 4 | 4 | 4 | 4 | 4 |
| GoLore's $\alpha$ | 4 | 4 | 4 | 4 | 4 | 4 | 4 | 4 |
| Frequency $\tau$ | 500 | 500 | 500 | 500 | 500 | 500 | 500 | 500 |
| Max Seq. Len. | 512 | 512 | 512 | 512 | 512 | 512 | 512 | 512 |

# F  CONNECTIONS WITH OTHER ALGORITHMS

**Connection with zero-th order methods.** Zero-th order methods (Malladi et al., 2023; Zhang et al., 2023; Chen et al., 2024) are another line of works on memory-efficient training. While these algorithms randomly select a direction to estimate the directional derivatives by finite difference, GoLore computes subspace gradients via back propagation. The directions used in zero-th order methods change every iteration, while GoLore applies a more lazily strategy changing its subspace every $\tau$ iterations.

**Connection with gradient sketching methods.** Gradient sketching methods like Hanzely et al. (2018) and Wang et al. (2024) uses gradient sketches in algorithm iterates. These methods recover gradient estimates from projected gradients and retains full-size gradients and optimizer states. In comparison, GoLore directly updates with projected gradients and retains compressed gradients and optimizer states, which is more memory-efficient.

# G  CONVERGENCE OF GALORE UNDER ISOTROPIC NOISE ASSUMPTIONS

Based on the anisotropic gradient noise we use to construct the counter-example in the proof of GaLore's non-convergence under standard assumptions, an interesting open question is whether GaLore is guaranteed to converge if the noise are further assumed isotropic. In this section, we consider the following additional assumption:

**Assumption 4** (Isotropic noise)**.** *The distribution of stochastic noise for each gradient matrix is invariant under orthogonal transformations, i.e., it holds for any layer $\ell = 1, \cdots, N_L$, parameter $\boldsymbol{x} \in \mathbb{R}^d$ and orthogonal matrix $\boldsymbol{O}_1 \in \mathbb{R}^{m_\ell \times m_\ell}$, $\boldsymbol{O}_2 \in \mathbb{R}^{n_\ell \times n_\ell}$ that*

$$\nabla_\ell F(\boldsymbol{x}; \xi) - \nabla_\ell f(\boldsymbol{x}) \stackrel{\text{dist}}{=} \boldsymbol{O}_1[\nabla_\ell F(\boldsymbol{x}; \xi) - \nabla_\ell f(\boldsymbol{x})]\boldsymbol{O}_2,$$

*where $A \stackrel{\text{dist}}{=} B$ represents $A$ and $B$ shares the same distribution.*

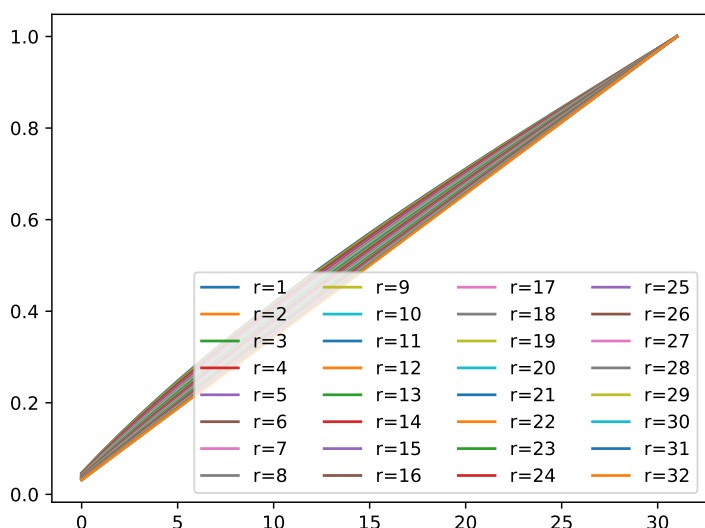

Figure 8: Observations with a small noise scale $\sigma = 0.1$.

**Remark.** The property in Assumption 4 can be satisfied by multivariate Gaussian distribution, *e.g.*,
$$\text{vec}(\nabla_\ell F(\boldsymbol{x}; \xi) - \nabla_\ell f(\boldsymbol{x})) \sim \mathcal{N}(0, \frac{\sigma_\ell^2}{m_\ell n_\ell} \cdot \boldsymbol{I}_{m_\ell \times n_\ell}).$$

Besides Assumption 4, we consider an additional assumption, which is crucial in analyzing the projection error.

**Assumption 5** (Leading property)**.** *Let $\mathcal{D}_\ell(\boldsymbol{x})$ denotes the distribution of gradient noise $\nabla_\ell F(\boldsymbol{x}; \xi) - \nabla_\ell f(\boldsymbol{x})$. We assume $\mathcal{D}_\ell(\boldsymbol{x})$ satisfies the following "leading property": if $\boldsymbol{A} \sim \mathcal{D}_\ell(\boldsymbol{x})$, $\boldsymbol{B} \in \mathbb{R}^{m_\ell \times n_\ell}$ satisfies $B_{11} \geq B_{22} \geq \cdots \geq B_{\min\{m_\ell, n_\ell\}, \min\{m_\ell, n_\ell\}} \geq 0$ and $B_{ij} = 0$ for $i \neq j$, the SVD decomposition $\boldsymbol{U}\boldsymbol{\Sigma}\boldsymbol{V}^\top$ satisfies*

$$\begin{cases} \frac{1}{r} \sum_{i=1}^{k} \sum_{j=1}^{r} \mathbb{E}[U_{ij}^2] \geq \frac{k}{n}, & \forall 1 \leq k, r \leq m, \quad \text{if } m_\ell \leq n_\ell; \\ \frac{1}{r} \sum_{i=1}^{k} \sum_{j=1}^{r} \mathbb{E}[V_{ij}^2] \geq \frac{k}{n}, & \forall 1 \leq k, r \leq n, \quad \text{if } m_\ell > n_\ell. \end{cases}$$

Though not fully established in theory, we can empirically validate that multivariate Gaussian distribution may satisfy Assumption 5.

Specifically, we consider the following experiment setup. Let $\text{vec}(\boldsymbol{A}) \sim \mathcal{N}(0, \sigma^2 \cdot \boldsymbol{I}_{32 \times 32})$ for some noise scale $\sigma > 0$ and select a fixed matrix $\boldsymbol{B}$ with $B_{11} \geq B_{22} \geq \cdots \geq B_{32,32} \geq 0$. In order to validate the properties in expectation, we sample matrix $\boldsymbol{A}$ for 200,000 times and uses the empirical expectations $\hat{\mathbb{E}}[U_{ij}]$'s to estimate the true expectations $\mathbb{E}[U_{ij}]$'s. Figures 8, 9, 10 represent results under different noise scales $\sigma = 10, 1, 0.1$, respectively, where "$r = r_0$" in each figure plots the line connecting points $(k, \frac{1}{r_0} \sum_{i=1}^{k} \sum_{j=1}^{r_0} \hat{\mathbb{E}}[U_{ij}^2])$ for $k = 1, 2, \cdots, 32$. As presented, all lines "$r = r_0$" with $r_0 < 32$ are above the line "$r = 32$", which is guaranteed to pass through the points $(k, \frac{k}{32})$, $k = 1, 2, \cdots, 32$, in theory. Consequently, we have good reason to believe that multivariate Gaussian distribution can empirically satisfy Assumption 5.

With Assumptions 4 and 5, we can establish new error bounds for GaLore's SVD projection.

**Lemma 17** (Error of GaLore's projection under isotropic noise)**.** *Let $\boldsymbol{G} = \nabla_\ell f(\boldsymbol{x})$ and $\boldsymbol{E} = \nabla_\ell F(\boldsymbol{x}; \xi) - \nabla_\ell f(\boldsymbol{x})$, projection matrix $\boldsymbol{P} = \boldsymbol{U}[:, : r_\ell]$, $\boldsymbol{Q} = \boldsymbol{V}[:, : r_\ell]$ where $\boldsymbol{U}\boldsymbol{\Sigma}\boldsymbol{V}^\top = \boldsymbol{G} + \boldsymbol{E}$ is the SVD of stochastic gradient $\nabla_\ell F(\boldsymbol{x}; \xi)$, it holds under Assumptions 4 and 5 for $m_\ell \leq n_\ell$ that*

$$\mathbb{E}[\|\boldsymbol{P}\boldsymbol{P}^\top\boldsymbol{G} - \boldsymbol{G}\|_F^2] \leq \left(1 - \frac{r_\ell}{m_\ell}\right) \|\boldsymbol{G}\|_F^2,$$

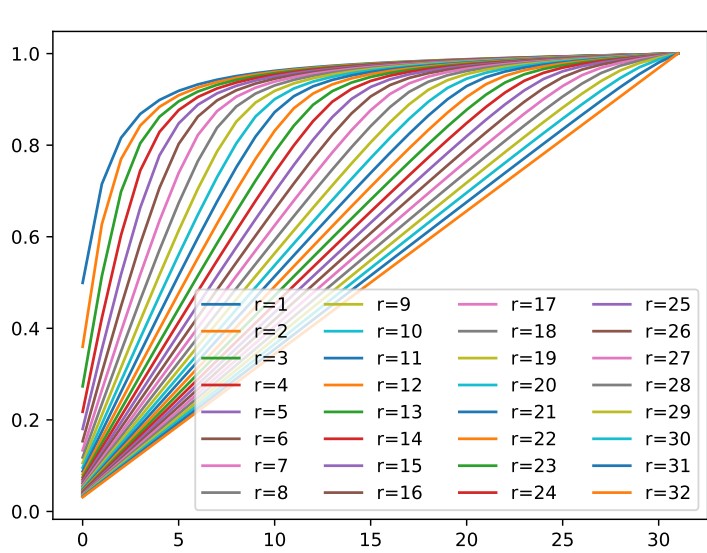

Figure 9: Observations with a medium noise scale $\sigma = 1$.

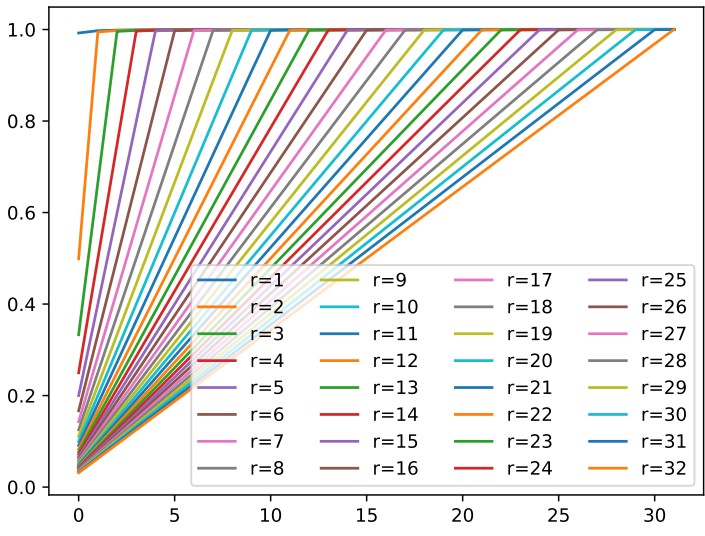

Figure 10: Observations with a large noise scale $\sigma = 10$.

and for $m_\ell > n_\ell$ that

$$\mathbb{E}[\|\boldsymbol{G}\boldsymbol{Q}\boldsymbol{Q}^\top - \boldsymbol{G}\|_F^2] \le \left(1 - \frac{r_\ell}{n_\ell}\right)\|\boldsymbol{G}\|_F^2.$$

*Proof.* We only consider the case where $m_\ell < n_\ell$, as the proof for the other case is similar. We first conduct SVD of $\boldsymbol{G}$ and get $\boldsymbol{G} = \boldsymbol{U}_0\boldsymbol{\Sigma}_0\boldsymbol{V}_0^\top$. It holds that

$$\begin{aligned}
\|\boldsymbol{P}\boldsymbol{P}^\top\boldsymbol{G}\|_F^2 &= \text{tr}(\boldsymbol{G}^\top\boldsymbol{P}\boldsymbol{P}^\top\boldsymbol{G}) \\
&= \text{tr}(\boldsymbol{V}_0\boldsymbol{\Sigma}_0^\top\boldsymbol{U}_0^\top\boldsymbol{P}\boldsymbol{P}^\top\boldsymbol{U}_0\boldsymbol{\Sigma}_0\boldsymbol{V}_0^\top) \\
&= \text{tr}(\boldsymbol{\Sigma}_0\boldsymbol{\Sigma}_0^\top\boldsymbol{U}_0^\top\boldsymbol{P}\boldsymbol{P}^\top\boldsymbol{U}_0).
\end{aligned} \tag{120}$$

Denote $\tilde{\boldsymbol{U}} = \boldsymbol{U}_0^\top\boldsymbol{U}$ and $\tilde{\boldsymbol{V}} = \boldsymbol{V}_0^\top\boldsymbol{V}$, it holds that $\tilde{\boldsymbol{U}}\boldsymbol{\Sigma}_0\tilde{\boldsymbol{V}}^\top = (\boldsymbol{U}_0^\top\boldsymbol{U})\boldsymbol{\Sigma}_0(\boldsymbol{V}_0^\top\boldsymbol{V})^\top$ is SVD of $\boldsymbol{U}_0^\top(\boldsymbol{G} + \boldsymbol{E})\boldsymbol{V}_0 = \boldsymbol{U}_0^\top\boldsymbol{E}\boldsymbol{V}_0 + \boldsymbol{\Sigma}_0 \overset{\text{dist}}{=} \boldsymbol{E} + \boldsymbol{\Sigma}_0$. By Assumption 5 we have

$$\frac{1}{r_\ell}\sum_{i=1}^{k}\sum_{j=1}^{r_\ell}\mathbb{E}[\tilde{U}_{ij}^2] \ge \frac{k}{m_\ell}, \quad k = 1, 2, \cdots, m_\ell. \tag{121}$$

Let $\sigma_1 \ge \sigma_2 \ge \cdots \sigma_{m_\ell} \ge 0$ represent the singular values of $\boldsymbol{G}$, taking expectations of (120) yields

$$\begin{aligned}
\mathbb{E}[\|\boldsymbol{P}\boldsymbol{P}^\top\boldsymbol{G}\|_F^2] &= \text{tr}(\boldsymbol{\Sigma}_0\boldsymbol{\Sigma}_0^\top\mathbb{E}[\boldsymbol{U}_0^\top\boldsymbol{P}\boldsymbol{P}^\top\boldsymbol{U}_0]) \\
&= \sum_{i=1}^{m_\ell}\sigma_i^2\sum_{j=1}^{r_\ell}\mathbb{E}[\tilde{U}_{ij}^2] \\
&\ge \sum_{i=1}^{m_\ell}\sigma_i^2 \cdot \frac{r_\ell}{m_\ell} = \frac{r_\ell}{m_\ell} \cdot \|\boldsymbol{G}\|_F^2,
\end{aligned} \tag{122}$$

where the inequality applies $\sigma_1^2 \ge \sigma_2^2 \ge \cdots \sigma_{m_\ell}^2$ and (121). Based on (122), we have

$$\mathbb{E}[\|\boldsymbol{P}\boldsymbol{P}^\top\boldsymbol{G} - \boldsymbol{G}\|_F^2] = \|\boldsymbol{G}\|_F^2 - \mathbb{E}[\|\boldsymbol{P}\boldsymbol{P}^\top\boldsymbol{G}\|_F^2] \le \left(1 - \frac{r_\ell}{m_\ell}\right)\|\boldsymbol{G}\|_F^2,$$

which completes the proof. □

**Lemma 18** (Momentum contraction)**.** *Under Assumption 3-5, in GaLore using MSGD with MP, if $0 < \beta_1 \le 1$, term $\tilde{\boldsymbol{M}}_\ell^{(t)}$ has the following contraction properties:*

- *When $t = 0$, it holds that*

$$\begin{aligned}
\mathbb{E}[\|\tilde{\boldsymbol{M}}_\ell^{(0)} - \nabla_\ell f(\boldsymbol{X}^{(0)})\|_F^2] \le &(\tau-1)(2-\delta_\ell)\sum_{r=0}^{\tau-2}\mathbb{E}[\|\nabla_\ell f(\boldsymbol{x}^{(r+1)}) - \nabla_\ell f(\boldsymbol{x}^{(r)})\|_F^2] \\
&+ \frac{2(2-\delta_\ell)}{\tau}\sum_{r=0}^{\tau-1}\mathbb{E}[\|\nabla_\ell f(\boldsymbol{x}^{(r)})\|_F^2] + \beta_1^2\sigma_\ell^2;
\end{aligned} \tag{123}$$

- *When $t = k\tau$, $k \in \mathbb{N}^*$, it holds that*

$$\begin{aligned}
&\mathbb{E}[\|\tilde{\boldsymbol{M}}_\ell^{(t)} - \nabla_\ell f(\boldsymbol{x}^{(t)})\|_F^2] - \left(1 - \left(1 - \frac{\delta_\ell}{4}\right)\beta_1\right)\mathbb{E}[\|\tilde{\boldsymbol{M}}_\ell^{(t-1)} - \nabla_\ell f(\boldsymbol{x}^{(t-1)})\|_F^2] \\
\le &\frac{2(1-\delta_\ell)}{\tau}\sum_{r=0}^{\tau-1}\mathbb{E}[\|\nabla_l f(\boldsymbol{x}^{(k\tau+r)})\|_F^2] + \frac{5(1-\beta_1)}{\delta_\ell\beta_1}\mathbb{E}[\|\nabla_\ell f(\boldsymbol{x}^{(t)}) - \nabla_\ell f(\boldsymbol{x}^{(t-1)})\|_F^2] \\
&+ (\tau-1)(1-\delta_\ell)\sum_{r=0}^{\tau-2}\mathbb{E}[\|\nabla_\ell f(\boldsymbol{x}^{(k\tau+r+1)}) - \nabla_\ell f(\boldsymbol{x}^{(k\tau+r)})\|_F^2] + \beta_1^2\sigma_\ell^2;
\end{aligned} \tag{124}$$

- *When $t = k\tau + r$, $k \in \mathbb{N}$, $1 \le r < \tau$, it holds that*

$$\mathbb{E}[\|\tilde{\boldsymbol{M}}_\ell^{(t)} - \nabla_\ell f(\boldsymbol{x}^{(t)})\|_F^2] - \left(1 - \left(1 - \frac{\delta_\ell}{4}\right)\beta_1\right)\mathbb{E}[\|\tilde{\boldsymbol{M}}_\ell^{(t-1)} - \nabla_\ell f(\boldsymbol{x}^{(t-1)})\|_F^2]$$

$$\le \left(1 - \frac{\delta_\ell}{2}\right)\beta_1\mathbb{E}[\|\nabla_\ell f(\boldsymbol{x}^{(t)})\|_F^2] + \frac{5(1-\beta_1)}{\delta_\ell\beta_1}\mathbb{E}[\|\nabla_\ell f(\boldsymbol{x}^{(t)}) - \nabla_\ell f(\boldsymbol{x}^{(t-1)})\|_F^2]$$

$$+ \frac{10r\beta_1}{\delta_\ell}\sum_{i=1}^{r}\mathbb{E}[\|\nabla_\ell f(\boldsymbol{x}^{(k\tau+i)}) - \nabla_\ell f(\boldsymbol{x}^{(k\tau+i-1)})\|_F^2] + \beta_1^2\sigma_\ell^2. \quad (125)$$

*Proof.* Without loss of generality assume $m_\ell \le n_\ell$ (the other case can be proved similarly). When $t = 0$, (123) is direct result of Lemma 8 by letting $\mathcal{B} = 1$. When $t = 0$, we have

$$\mathbb{E}[\|\tilde{\boldsymbol{M}}_\ell^{(0)} - \nabla_\ell f(\boldsymbol{x}^{(0)})\|_F^2]$$

$$=\mathbb{E}[\|\beta_1\boldsymbol{P}_\ell^{(0)}(\boldsymbol{P}_\ell^{(0)})^\top\boldsymbol{G}_\ell^{(0)} - \nabla_\ell f(\boldsymbol{x}^{(0)})\|_F^2]$$

$$=\mathbb{E}[\|(\beta_1\boldsymbol{P}_\ell^{(0)}(\boldsymbol{P}_\ell^{(0)})^\top - \boldsymbol{I})\nabla_\ell f(\boldsymbol{x}^{(0)})\|_F^2] + \beta_1^2\mathbb{E}[\|\boldsymbol{P}_\ell^{(0)}(\boldsymbol{P}_\ell^{(0)})^\top(\boldsymbol{G}_\ell^{(0)} - \nabla_\ell f(\boldsymbol{x}^{(0)}))\|_F^2], \quad (126)$$

For the first term, we have

$$\mathbb{E}[\|(\beta_1\boldsymbol{P}_\ell^{(0)}(\boldsymbol{P}_\ell^{(0)})^\top - \boldsymbol{I})\nabla_\ell f(\boldsymbol{x}^{(0)})\|_F^2]$$

$$=(1-\beta_1)^2\mathbb{E}[\|\boldsymbol{P}_\ell^{(0)}(\boldsymbol{P}_\ell^{(0)})^\top\nabla_\ell f(\boldsymbol{x}^{(0)})\|_F^2] + \mathbb{E}[\|(\boldsymbol{I} - \boldsymbol{P}_\ell^{(0)}(\boldsymbol{P}_\ell^{(0)})^\top)\nabla_\ell f(\boldsymbol{x}^{(0)})\|_F^2]$$

$$\le \left((1-\beta_1)^2 + (1-\delta_\ell)\right)\|\nabla_\ell f(\boldsymbol{x}^{(0)})\|_F^2 \le (2-\delta_\ell)\|\nabla_\ell f(\boldsymbol{x}^{(0)})\|_F^2, \quad (127)$$

where the first inequality uses Lemma 17. For the second term, we have

$$\mathbb{E}[\|\boldsymbol{P}_\ell^{(0)}(\boldsymbol{P}_\ell^{(0)})^\top(\boldsymbol{G}_\ell^{(0)} - \nabla_\ell f(\boldsymbol{x}^{(0)}))\|_F^2] \le \mathbb{E}[\|\boldsymbol{G}_\ell^{(0)} - \nabla_\ell f(\boldsymbol{x}^{(0)})\|_F^2] \le \sigma_\ell^2. \quad (128)$$

Applying (127)(128) to (126) and using Lemma 2 yields (123).

When $t = k\tau$, $k \in \mathbb{N}^*$, according to the proof of Lemma 8, we have

$$\mathbb{E}[\|\tilde{\boldsymbol{M}}_\ell^{(t-1)} - \nabla_\ell f(\boldsymbol{x}^{(t)})\|_F^2]$$

$$\le \left(1 + \frac{\delta_\ell\beta_1}{4}\right)\mathbb{E}[\|\tilde{\boldsymbol{M}}_\ell^{(t-1)} - \nabla_\ell f(\boldsymbol{x}^{(t-1)})\|_F^2] + \left(1 + \frac{4}{\delta_\ell\beta_1}\right)\mathbb{E}[\|\nabla_\ell f(\boldsymbol{x}^{(t)}) - \nabla_\ell f(\boldsymbol{x}^{(t-1)})\|_F^2], \quad (129)$$

and

$$\mathbb{E}[\|\tilde{\boldsymbol{M}}_\ell^{(t)} - \nabla_\ell f(\boldsymbol{x}^{(t)})\|_F^2]$$

$$=\mathbb{E}[\|\boldsymbol{P}_\ell^{(t)}(\boldsymbol{P}_\ell^{(t)})^\top[(1-\beta_1)\tilde{\boldsymbol{M}}_\ell^{(t-1)} + \beta_1\boldsymbol{G}_\ell^{(t)} - \nabla_\ell f(\boldsymbol{x}^{(t)})]\|_F^2]$$

$$+ \mathbb{E}[\|(\boldsymbol{I} - \boldsymbol{P}_\ell^{(t)}(\boldsymbol{P}_\ell^{(t)})^\top)\nabla_\ell f(\boldsymbol{x}^{(t)})\|_F^2]$$

$$\le\mathbb{E}[\|(1-\beta_1)(\tilde{\boldsymbol{M}}_\ell^{(t-1)} - \nabla_\ell f(\boldsymbol{x}^{(t)}))\|_F^2] + \beta_1^2\sigma_\ell^2 + (1-\delta_\ell)\mathbb{E}[\|\nabla_\ell f(\boldsymbol{x}^{(t)})\|_F^2], \quad (130)$$

where the last inequality applies Lemma 17. Applying (129) to (130) and using Lemma 2 yields (124).

When $t = k\tau + r$, $k \in \mathbb{N}$, $1 \le r < \tau$, we have the following results according to the proof of Lemma 8:

$$\mathbb{E}[\|\tilde{\boldsymbol{M}}_\ell^{(t)} - \nabla_\ell f(\boldsymbol{x}^{(t)})\|_F^2]$$

$$\le(1-\beta_1)\mathbb{E}[\|\tilde{\boldsymbol{M}}_\ell^{(t-1)} - \nabla_\ell f(\boldsymbol{x}^{(t)})\|_F^2] + \beta_1\mathbb{E}[\|(\boldsymbol{I} - \boldsymbol{P}_\ell^{(t)}(\boldsymbol{P}_\ell^{(t)})^\top)\nabla_\ell f(\boldsymbol{x}^{(t)})\|_F^2$$

$$+ \beta_1^2\mathbb{E}[\boldsymbol{P}_\ell^{(t)}(\boldsymbol{P}_\ell^{(t)})^\top(\boldsymbol{G}_\ell^{(t)} - \nabla_\ell f(\boldsymbol{x}^{(t)}))\|_F^2]$$

$$\le(1-\beta_1)\mathbb{E}[\|\tilde{\boldsymbol{M}}_\ell^{(t-1)} - \nabla_\ell f(\boldsymbol{x}^{(t)})\|_F^2] + \beta_1\mathbb{E}[\|(\boldsymbol{I} - \boldsymbol{P}_\ell^{(t)}(\boldsymbol{P}_\ell^{(t)})^\top)\nabla_\ell f(\boldsymbol{x}^{(t)})\|_F^2 + \beta_1^2\sigma_\ell^2, \quad (131)$$

For the second term, we have

$$\mathbb{E}[\|(\boldsymbol{I} - \boldsymbol{P}_\ell^{(k\tau)}(\boldsymbol{P}_\ell^{(k\tau)})^\top)\nabla_\ell f(\boldsymbol{x}^{(t)})\|_F^2]$$

$$\leq \left(1 + \frac{\delta_\ell}{4}\right)\mathbb{E}[\|(\boldsymbol{I} - \boldsymbol{P}_\ell^{(k\tau)}(\boldsymbol{P}_\ell^{(k\tau)})^\top)\nabla_\ell f(\boldsymbol{x}^{(k\tau)})\|_F^2]$$

$$+ \left(1 + \frac{4}{\delta_\ell}\right)\mathbb{E}[\|(\boldsymbol{I} - \boldsymbol{P}_\ell^{(k\tau)}(\boldsymbol{P}_\ell^{(k\tau)})^\top)(\nabla_\ell f(\boldsymbol{x}^{(t)}) - \nabla_\ell f(\boldsymbol{x}^{(k\tau)}))\|_F^2]$$

$$\leq \left(1 - \frac{3\delta_\ell}{4}\right)\mathbb{E}[\|\nabla_\ell f(\boldsymbol{x}^{(k\tau)})\|_F^2] + \left(1 + \frac{4}{\delta_\ell}\right)\mathbb{E}[\|\nabla_\ell f(\boldsymbol{x}^{(t)}) - \nabla_\ell f(\boldsymbol{x}^{(k\tau)})\|_F^2]$$

$$\leq \left(1 - \frac{\delta_\ell}{2}\right)\mathbb{E}[\|\nabla_\ell f(\boldsymbol{x}^{(t)})\|_F^2] + 2\left(1 + \frac{4}{\delta_\ell}\right)\mathbb{E}[\|\nabla_\ell f(\boldsymbol{x}^{(t)}) - \nabla_\ell f(\boldsymbol{x}^{(k\tau)})\|_F^2]$$

$$\leq \left(1 - \frac{\delta_\ell}{2}\right)\mathbb{E}[\|\nabla_\ell f(\boldsymbol{x}^{(t)})\|_F^2] + \frac{10r}{\delta_\ell}\sum_{i=1}^r \mathbb{E}[\|\nabla_\ell f(\boldsymbol{x}^{(k\tau+i)}) - \nabla_\ell f(\boldsymbol{x}^{(k\tau+i-1)})\|_F^2], \quad (132)$$

where the first inequality applies Young's inequality, the second inequality applies Lemma 17, the third inequality applies Young's inequality, the last inequality applies Cauchy's inequality. Applying (129)(132) to (131) yields (125). □

**Lemma 19** (Momentum error). *Under Assumption 2-5, if $0 < \beta_1 \leq 1$ in GaLore using MSGD and MP, it holds for any $K \geq 1$ that*

$$\sum_{t=0}^{K\tau-1}\mathbb{E}[\|\tilde{\boldsymbol{m}}^{(t)} - \nabla f(\boldsymbol{x}^{(t)})\|_2^2]$$

$$\leq \left(\frac{5(1-\beta_1)}{(1-\underline{\delta}/4)\underline{\delta}\beta_1^2} + \frac{5\tau(\tau-1)}{(1-\underline{\delta}/4)\underline{\delta}} + \frac{2(\tau-1)}{(1-\overline{\delta}/4)\beta_1}\right)L^2\sum_{t=0}^{K\tau-2}\mathbb{E}[\|\boldsymbol{x}^{(t+1)} - \boldsymbol{x}^{(t)}\|_2^2]$$

$$+ \left(\frac{1-\underline{\delta}/2}{1-\underline{\delta}/4} + \frac{4}{(1-\overline{\delta}/4)\tau\beta_1}\right)\sum_{t=0}^{K\tau-1}\mathbb{E}[\|\nabla f(\boldsymbol{x}^{(t)})\|_2^2] + \frac{K\tau\beta_1\sigma^2}{1-\overline{\delta}/4}. \quad (133)$$

*Proof.* By Lemma 18 we have

$$\sum_{t=0}^{K\tau-1}\mathbb{E}[\|\tilde{\boldsymbol{M}}_\ell^{(t)} - \nabla_\ell f(\boldsymbol{x}^{(t)})\|_F^2] - \left(1 - \left(1 - \frac{\delta_\ell}{4}\right)\beta_1\right)\sum_{t=0}^{K\tau-2}\mathbb{E}[\|\tilde{\boldsymbol{M}}_\ell^{(t)} - \nabla_\ell f(\boldsymbol{x}^{(t)})\|_F^2]$$

$$\leq \left(\frac{5(1-\beta_1)}{\delta_\ell\beta_1} + \frac{5\tau(\tau-1)\beta_1}{\delta_\ell} + 2(\tau-1)\right)\sum_{t=0}^{K\tau-2}\mathbb{E}[\|\nabla_\ell f(\boldsymbol{x}^{(t+1)}) - \nabla_\ell f(\boldsymbol{x}^{(t)})\|_F^2]$$

$$+ \left(\frac{4}{\tau} + \left(1 - \frac{\delta_\ell}{2}\right)\beta_1\right)\sum_{t=0}^{K\tau-1}\mathbb{E}[\|\nabla_\ell f(\boldsymbol{x}^{(t)})\|_F^2] + K\tau\beta_1^2\sigma_\ell^2,$$

which implies

$$\sum_{t=0}^{K\tau-1}\mathbb{E}[\|\tilde{\boldsymbol{M}}_\ell^{(t)} - \nabla_\ell f(\boldsymbol{x}^{(t)})\|_F^2]$$

$$\leq \left(\frac{5(1-\beta_1)}{(1-\delta_\ell/4)\delta_\ell\beta_1^2} + \frac{5\tau(\tau-1)}{(1-\delta_\ell/4)\delta_\ell} + \frac{2(\tau-1)}{(1-\delta_\ell/4)\beta_1}\right)\sum_{t=0}^{K\tau-2}\mathbb{E}[\|\nabla_\ell f(\boldsymbol{x}^{(t+1)}) - \nabla_\ell f(\boldsymbol{x}^{(t)})\|_F^2]$$

$$+ \left(\frac{1-\delta_\ell/2}{1-\delta_\ell/4} + \frac{4}{(1-\delta_\ell/4)\tau\beta_1}\right)\sum_{t=0}^{K\tau-1}\mathbb{E}[\|\nabla_\ell f(\boldsymbol{x}^{(t)})\|_F^2] + \frac{K\tau\beta_1\sigma_\ell^2}{1-\delta_\ell/4}. \quad (134)$$

Summing (134) for $\ell = 1, \cdots, N_L$ and applying Assumption 2-3 yields (133). □

Now we are ready to prove the convergence of GaLore with small-batch stochastic gradients under isotropic noise assumptions.

**Theorem 13** (Convergence of Galore under isotropic noise assumptions). *Under Assumptions 1-5, if hyperparameters*

$$0 < \beta_1 \le 1, \quad \tau \ge \frac{128}{3\beta_1\underline{\delta}}, \quad 0 < \eta \le \min\left\{\frac{1}{4L}, \sqrt{\frac{3\underline{\delta}\beta_1^2}{80L^2}}, \sqrt{\frac{3\underline{\delta}}{80\tau^2L^2}}, \sqrt{\frac{3\beta_1}{32\tau L^2}}\right\}, \quad (135)$$

*GaLore using small-batch stochastic gradients and MSGD with MP converges as*

$$\frac{1}{K\tau}\sum_{t=0}^{K\tau-1}\mathbb{E}\|\nabla f(\boldsymbol{x}^{(t)})\|_2^2 \le \frac{16\Delta}{\underline{\delta}\eta K\tau} + \frac{32\beta_1\sigma^2}{3\underline{\delta}} \quad (136)$$

*for any $K \ge 1$, where $\Delta = f(\boldsymbol{x}^{(0)}) - \inf_{\boldsymbol{x}} f(\boldsymbol{x})$.*

*Proof.* By Lemma 4 we have

$$\sum_{t=0}^{K\tau-1}\mathbb{E}[\|\nabla f(\boldsymbol{x}^{(t)})\|_2^2] \le \frac{2[f(\boldsymbol{x}^{(0)}) - \mathbb{E}[f(\boldsymbol{x}^{(K\tau)})]]}{\eta} + \sum_{t=0}^{K\tau-1}\mathbb{E}[\|\tilde{\boldsymbol{m}}^{(t)} - \nabla f(\boldsymbol{x}^{(t)})\|_2^2]$$

$$- \left(\frac{1}{\eta^2} - \frac{L}{\eta}\right)\sum_{t=0}^{K\tau-1}\mathbb{E}[\|\boldsymbol{x}^{(t+1)} - \boldsymbol{x}^{(t)}\|_2^2]. \quad (137)$$

Applying Lemma 19 to (137) and using $\underline{\delta} \le \bar{\delta} < 1$ yields

$$\left(\frac{\underline{\delta}}{4} - \frac{16}{3\tau\beta_1}\right)\sum_{t=0}^{K\tau-1}\mathbb{E}[\|\nabla f(\boldsymbol{x}^{(t)})\|_2^2]$$

$$\le \frac{2}{\eta}\mathbb{E}[f(\boldsymbol{x}^{(0)}) - f(\boldsymbol{x}^{(K\tau)})] + \frac{4K\tau\beta_1\sigma^2}{3}$$

$$- \left(\frac{1}{\eta^2} - \frac{L}{\eta} - \frac{20(1-\beta_1)L^2}{3\underline{\delta}\beta_1^2} - \frac{20\tau(\tau-1)L^2}{3\underline{\delta}} - \frac{8(\tau-1)L^2}{3\beta_1}\right)\sum_{t=0}^{K\tau-1}\mathbb{E}[\|\boldsymbol{x}^{(t+1)} - \boldsymbol{x}^{(t)}\|_2^2].$$

$$(138)$$

By (135) we have

$$\frac{\underline{\delta}}{4} - \frac{16}{3\tau\beta_1} \ge \frac{\underline{\delta}}{8}, \quad \text{and} \quad \frac{1}{4\eta^2} \ge \max\left\{\frac{L}{\eta}, \frac{20(1-\beta_1)L^2}{3\underline{\delta}\beta_1^2}, \frac{20\tau(\tau-1)L^2}{3\underline{\delta}}, \frac{8(\tau-1)L^2}{3\beta_1}\right\}. \quad (139)$$

Applying (139) to (138) yields (136). $\quad\square$

**Corollary 4** (Convergence complexity of GaLore under isotropic noise assumptions). *Under Assumptions 1-5, if $T \ge 2 + 256/(3\underline{\delta}) + (256\sigma)^2/(9\sqrt{\underline{\delta}}L\Delta)$ and we choose*

$$\beta_1 = \left(1 + \sqrt{\frac{\underline{\delta}^{3/2}\sigma^2 T}{L\Delta}}\right)^{-1},$$

$$\tau = \left\lceil\frac{128}{3\underline{\delta}\beta_1}\right\rceil,$$

$$\eta = \left(4L + \sqrt{\frac{80L^2}{3\underline{\delta}\beta_1^2}} + \sqrt{\frac{80\tau^2L^2}{3\underline{\delta}}} + \sqrt{\frac{32\tau L^2}{3\beta_1}}\right)^{-1},$$

*GaLore using small-batch stochastic gradients and MSGD with MP converges as*

$$\frac{1}{T}\sum_{t=0}^{T-1}\mathbb{E}[\|\nabla f(\boldsymbol{x}^{(t)})\|_2^2] = \mathcal{O}\left(\frac{L\Delta}{\underline{\delta}^{5/2}T} + \sqrt{\frac{L\Delta\sigma^2}{\underline{\delta}^{7/2}T}}\right), \quad (140)$$

*where $\Delta = f(\boldsymbol{x}^{(0)}) - \inf_{\boldsymbol{x}} f(\boldsymbol{x})$. Consequently, the computation complexity to reach an $\varepsilon$-accurate solution $\boldsymbol{x}$ such that $\|\nabla f(\boldsymbol{x})\|_2^2 \le \varepsilon$ is $\mathcal{O}\left(\frac{L\Delta\sigma^2}{\underline{\delta}^{7/2}\varepsilon^2} + \frac{L\Delta}{\underline{\delta}^{5/2}\varepsilon} + \frac{\sigma^2}{\underline{\delta}^{1/2}L\Delta} + \frac{1}{\underline{\delta}}\right)$.*

*Proof.* $T \geq 2 + 256/(3\underline{\delta}) + (256\sigma)^2/(9\sqrt{\underline{\delta}}L\Delta)$ guarantees $T \geq \tau$. Let $T = K\tau + r$, where $K \in \mathbb{N}^*$ and $0 \leq r < \tau$. If $r = 0$, (140) is a direct result of Theorem 13. If $r > 0$, applying Theorem 13 to $\tilde{K} := K + 1$ yields

$$\frac{1}{T}\sum_{t=0}^{T-1}\mathbb{E}[\|\nabla f(\boldsymbol{x}^{(t)})\|_2^2] \leq \frac{\tilde{K}\tau}{T} \cdot \frac{1}{\tilde{K}\tau}\sum_{t=0}^{\tilde{K}\tau-1}\mathbb{E}[\|\nabla f(\boldsymbol{x}^{(t)})\|_2^2] = \mathcal{O}\left(\frac{L\Delta}{\underline{\delta}^{5/2}T} + \sqrt{\frac{L\Delta\sigma^2}{\underline{\delta}^{7/2}T}}\right).$$

$\square$

