# OpenReview forum: "Subspace Optimiztion for Large Language Models with Convergence Guarantees"
_ICLR.cc/2025/Conference — Submitted to ICLR 2025_

### Official Review · Reviewer_9uB7 · 2024-10-17

**Soundness:** 3
**Presentation:** 3
**Contribution:** 2
**Rating:** 6
**Confidence:** 4

**Summary:**

This paper studies the non-convergence of GaLore in stochastic optimization, and addresses this issue using simple approaches. Some numerical experiments are provided to support the proposed GoLore.

**Strengths:**

(+) The convergence of GaLore is characterized under common assumptions for smooth stochastic nonconvex optimization. The results show that GaLore's projection (via SVD) cannot cope with gradient noise in some cases.

(+) The authors then address the non-convergence of GaLore by substituting its projection with random projection. The proposed GoLore is proved to converge.

**Weaknesses:**

(-) The non-convergence of GaLore is studied and addressed within the smooth optimization, which is not the major application of GaLore. While non-smooth might be challenging, there are still some analyzable problems, e.g., those in GaLore paper.

(-) While the choice of random projection in GoLore is not necessary the best choice in practice. The paper does not discuss on this point. There can be several natural choices, and one example is to use random r singular vectors instead of top-r singular vectors. The authors' methodology of mixing GoLore and GaLore is a supporting evidence on the possible inefficiency of GoLore.

(-) The numerical benefit of GoLore is not very obvious compared to GaLore in both pretraining and finetuning tasks. Note that the improvement of GoLore also needs to apply GaLore at initial training. The numerical comparison with FLoRA is missing. This is very important given the similarity between GoLore and FLoRA.

(-) Typo in the title: optimiztion --> optimization

**Questions:**

Q1. I would like to know the authors' point of view on the following question. GaLore is a practice-oriented approach for memory efficiency of LLM training. The major application of GaLore is clearly a non-smooth optimization problem. That being said, both GoLore and GaLore do not have theoretical guarantees. How would the benefits of random projection proposed in this paper extend to non-smooth settings? As indicated in Figure 3, GoLore and GaLore share similar performance.

---

> ### Author Response · Authors · 2024-11-24
> **Response to Reviewer 9uB7 (Part 1/2)**
>
> We thank the reviewer for the detailed comments. All questions have been clarified as best as we can, and we are glad to address further comments or questions.
>
> **Weakness 1. The non-convergence of GaLore is studied and addressed with the smooth optimization, which is not the major application of GaLore.**
>
> We appreciate the reviewer’s feedback and would like to clarify that **our non-convergence result in Theorem 1 extends to the non-smooth setting**. Specifically, our manuscript makes three assumptions: (A1) Lower boundedness, (A2) L-smoothness, and (A3) Stochastic gradients. Theorem 1 states:
>
> *"There exists **a counter-example satisfying Assumptions A1, A2, and A3** for which GaLore cannot converge to a stationary solution."*
>
> Notably, any counter-example satisfying Assumptions A1, A2, and A3 inherently satisfies Assumptions A1 and A3. Thus, we can state the following:
>
> *"There exists **a counter-example satisfying Assumptions A1 and A3** for which GaLore cannot converge to a stationary solution."*
>
> This revised theorem demonstrates that GaLore fails to converge even in non-smooth settings, addressing the reviewer’s concern directly.
>
> **Weakness 2. The random projection strategy may not be the best choice and the authors did not discuss other choices like using random $r$ singular vectors. Mixing GaLore with GoLore is evidence for possible inefficiency of GoLore.**
>
> (1) Seeking the best random projection is beyond the scope of our contribution
>
> The reviewer may have misunderstood our contributions. Our primary contribution is to demonstrate that the SVD projection in GaLore leads to non-convergence and to propose a random projection as a solution to this issue. We do not aim to identify the optimal random projection. Besides, some properties of our random projection strategy, such as Lemma 5, are essential in the convergence proofs, whch may not be satisfied by other random strategies. While it is possible that a better random projection exists than the one we proposed, **this does not undermine our contribution of introducing the first random projection strategy that ensures GaLore’s convergence with theoretical guarantees**.
>
> (2) Mixing GaLore with GoLore is a carefully designed solution
>
> The hybrid of GaLore and GoLore reflects a carefully crafted algorithmic design based on a deep understanding of their respective properties. According to our theoretical analysis, the SVD projection in GaLore is more effective at capturing the true gradient component in the early stages of pretraining, where the true gradient dominates the gradient noise.
>
> However, in the later stages of training, where gradient noise becomes more prominent, the random projection approach in GoLore demonstrates its advantage. **In essence, the SVD projection accelerates convergence during the initial training stages, while random projections enable more accurate convergence in later stages**. This insight naturally leads to the hybrid GaLore + GoLore approach, leveraging the strengths of both methods for optimal performance across different training phases.
>
> Based on our experiences, SVD projection in the early stage outperforms any shape of random projection due to its efficiency in capturing the main gradient component. Consequently, we believe that combining SVD and random projections is consistently the superior choice for achieving both efficiency and accuracy throughout the training process.
>
> **Weakness 3. The numerical benefit is not very obvious. The comparison with Flora is missing.**
>
> (1) Numerical results
>
> Since the main contribution of this paper lies in establishing convergence guarantees for memory-efficient algorithms, we believe our experimental results have validated our theoretical findings. Please refer to our response to Point 3 in the global response for further details.
>
> In particular, we have conducted new experiments to fine-tune OPT-13B on the BoolQ dataset. As shown in Table 4 of Appendix E, GoLore achieves an accuracy of 81.96%, notably higher than GaLore's accuracy of 79.79%.
>
> Furthermore, we would like to emphasize that GoLore retains the algorithmic structure of GaLore, differing only in replacing the SVD projection with a random projection during the later phase of pre-training. **This simple modification introduces no additional computational overhead while providing strong convergence guarantees and consistently improved performance**. Given these advantages, we believe random projection is a valuable technique with both theoretical benefits and practical utility.
>
> (2) Comparison with Flora
>
> We have conducted additional experiments on Flora and added the results in Table 2, where Flora does not perform better than GaLore.
>
> **Weakness 4. Typo in the title: 'optimiztion' -> 'optimization'**
>
> We thank the reviewer for pointing out this serious problem and have corrected the typos in our revised manuscripts.

---

> ### Author Response · Authors · 2024-11-24
> **Response to Reviewer 9uB7 (Part 2/2)**
>
> **Question 1. I would like to know the authors' point of view on the following question. GaLore is a practice-oriented approach for memory efficiency of LLM training. The major application of GaLore is clearly a non-smooth optimization problem. That being said, both GoLore and GaLore do not have theoretical guarantees. How would the benefits of random projection proposed in this paper extend to non-smooth settings? As indicated in Figure 3, GoLore and GaLore share similar performance.**
>
> (1) Theoretical benefits of random projection in non-smooth settings
>
> As established in our response to Weakness 1, Theorem 1 naturally extends to non-smooth settings, demonstrating that GaLore fails to converge to stationary solutions in such scenarios due to SVD projection. This SVD-induced error can be mitigated through random projection in non-smooth settings, as the error is independent of the cost function's smoothness properties. Consequently, we can safely claim that GoLore incurs less error than GaLore due to the benefits brought by random projection. Theoretically, we believe the most convincing approach to justify this statement is to provide a rigous proof of GoLore's convergence under non-smooth settings, which is an interesting open question that we leave as our future work.
>
> We would like to emphasize that establishing convergence under the $L$-smoothness assumption is not a limitation unique to our work. Indeed, convergence guarantees for various optimization methods—including SGD, momentum SGD, adaptive SGD, and distributed SGD—are traditionally established under the standard $L$-smoothness assumption, yet these methods are widely applied to deep learning tasks with non-smooth activations. **To our knowledge, it is uncommon in the literature to establish convergence analyses for momentum SGD and Adam have not been developed without the $L$-smoothness assumption**. Our work follows this established convention.
>
> (2) Empirical benefits of random projection in real-world applications
>
> However, we can still validate the benefit of random projection through empirical results. The similar performance in Figure 3 is due to the insufficient number of tokens that the algorithms trained on, which we previously set following the setup in GaLore's paper. With insufficient training tokens, the algorithms have not reached the neighborhood of stationary solutions, where the gradients are less noise-dominated and random projection is less beneficial. In our revised manuscripts, we reproduced the experiment with more total training tokens, and GoLore shows clearer advantage in the results presented in Figure 7, Appendix E. Moreover, we have conducted new experiments to fine-tune OPT-13B on the BoolQ dataset. As shown in Table 4 of Appendix E, GoLore achieves an accuracy of 81.96%, notably higher than GaLore's accuracy of 79.79%.
>
> We thank the reviewer again for his careful comments. We hope these responses can clarify the reviewer's questions and are more than happy to address any further comments or questions.

---

> > ### Comment · Reviewer_9uB7 · 2024-11-26
> >
> > Thanks for the responses. The score is adjusted accordingly to reflect the authors' clarification on numerical results. However, the responses towards weakness 1 and 2 do not fully convince me.
> >
> > The response to W1 suggests the existence of a counterexample in a broader context beyond the targeted (non-smooth) setting. However, it remains unclear whether non-convergence still holds within the targeted non-smooth setting. It is possible that GaLore does not converge in the broader context, but still achieving convergence in the non-smooth setting. While I do understand establishing convergence on non-smooth problems can be challenging since the entire trajectory need to be considered, finding a non-converging counter example seems to be less challenging.
> >
> > The response to W2 lacks clarity. In point (2), the authors appear to advocate for “carefully designing algorithms,” yet in point (1), they state that “seeking the best random projection is beyond the scope of our contribution.” Since random projection is a straightforward alternative to the approach the authors proposed, this inconsistency raises questions about whether the algorithm has been carefully designed.

---

> > > ### Author Response · Authors · 2024-11-29
> > >
> > > We sincerely thank the reviewer for acknowledging our clarification regarding the numerical results. We are more than happy to address the further questions raised.
> > >
> > > **1. The Response to W1 suggests the existence of a counter-example in a broader context beyond the targeted (non-smooth) setting.**
> > >
> > > **(1) Clarification of the "broader context":**
> > >
> > > We would like to make a further clarification based on the previous response to W1. In fact, we believe the "broader context" we presented is a standard non-smooth scenario in convergence analysis. Typically, we say an algorithm can converge under non-smooth settings by proving its convergence under assumption A1 (lower boundedness) and assumption A3 (unbiased stochastic gradient), while relaxing the assumption of A2 ($L$-smoothness). Conversely, an algorithm is not guaranteed to converge under the non-smooth setting if a counter-example can be constructed satisfying A1 and A3 but violating convergence.
> > >
> > > **(2) Constructing a non-smooth counter-example for GaLore:**
> > >
> > > We confirm that a non-smooth counter-example exists where GaLore fails to converge. Specifically, let $f(X)$ denote the counter-example from the proof of Theorem 1. By adding an $\ell_1$ regularization term on the diagonal elements of $X$, i.e., $r(X)=\alpha\cdot\sum_{i=1}^n|X_{ii}|$, where $\alpha>0$, we obtain the non-smooth function $\Phi(X):=f(X)+r(X)$. The subgradient of $\Phi(X)$ is given by
> > > $$G^{(t)}=\begin{pmatrix} L\lambda+\alpha\cdot\mathrm{sign}(X_{11}) & 0 & \cdots & 0\\\\ 0 & \xi^{(t)}\tilde{\sigma}+\alpha\cdot\mathrm{sign}(X_{22}) & \cdots & 0 \\\\ \vdots & \vdots & \ddots & \vdots\\\\ 0 & 0 & \cdots & \sqrt{n-1}\xi^{(t)}\tilde{\sigma}+\alpha\cdot\mathrm{sign}(X_{nn})\end{pmatrix}$$
> > > When $\alpha$ is sufficiently small (i.e., $\alpha<(\tilde{\sigma}-L\lambda)/2$), the subspace captured by SVD projection remains orthogonal to the true gradient. The remaining proof showing GaLore's non-convergence on $\Phi(X)$ follows directly from the proof of Theorem 1.
> > >
> > > **2. Not seeking the best random projection: has the algorithm been carefully designed?**
> > >
> > > **(1) Careful theoretical design of the random strategy:**
> > >
> > > While our approach does not seek the *best* random projection, the random strategy is carefully chosen based on theoretical principles. The design ensures that the algorithm converges under the stated assumptions. Specifically, we sample the projection matrix from the uniform distribution on the Stiefel manifold, which guarantees the following key properties:
> > >
> > > **Lemma 5 (Error of GoLore's projection).** Let $P\sim\mathcal{U}(\mathrm{St}\_{m,r})$, $Q\sim\mathcal{U}(\mathrm{St}\_{n,r})$, it holds for all $G\in\mathbb{R}^{m\times n}$ that
> > > $$\mathbb{E}[PP^\top]=\frac{r}{m}\cdot I,\quad\mathbb{E}[QQ^\top]=\frac{r}{n}\cdot I,$$ and
> > > $$\mathbb{E}[\\|PP^\top G-G\\|_F^2]=\left(1-\frac{r}{m}\right)\\|G\\|_F^2,\quad\mathbb{E}[\\|GQQ^\top-G\\|_F^2]=\left(1-\frac{r}{n}\right)\\|G\\|_F^2.$$
> > >
> > > These properties are crucial for maintaining theoretical guarantees. Alternative random strategies may fail to satisfy such bounds, potentially compromising convergence or performance.
> > >
> > > **(2) Practical considerations and sufficiency of the chosen strategy:**
> > >
> > > While our random projection strategy may not be optimal, we believe it does not require additional careful design for the following reasons:
> > >
> > > - Theoretical guarantees: The proposed strategy satisfies Lemma 5, which ensures an error bound similar to that of GaLore's SVD-based projection (as shown in Lemma 1).
> > > - Intuitive effectiveness in late training stages: Random projections are particularly advantageous in later training stages, where gradient noise dominates. We conjecture that, under high noise levels, most strategies cannot effectively extract additional useful information from the noisy gradient to select the subspace. Consequently, different strategies are likely to perform similarly in such scenarios.
> > >
> > > We deeply appreciate your thoughtful feedback and are more than happy to address any further questions or concerns.

---

### Official Review · Reviewer_YUoQ · 2024-11-03

**Soundness:** 2
**Presentation:** 1
**Contribution:** 2
**Rating:** 3
**Confidence:** 3

**Summary:**

This paper analyzes the convergence properties of GaLore, an existing method for memory-efficient training of large language models (LLMs), where parameters are updated within the dominant subspace of the gradient from a previous iteration rather than the ambient space. For non-convex functions under standard assumptions, the authors establish that GaLore, with a noiseless gradient oracle, converges to a first-order stationary point at a rate of $1/T$. In the stochastic gradient setting, they demonstrate the existence of a loss function and noisy gradient oracle where GaLore fails to converge with probability 1. However, they show that with a sufficiently large and increasing batch size, GaLore provably converges to a first-order stationary point in a mean-square sense at a rate of $1/\sqrt{T}$. Finally, they propose GoLore, a variant of GaLore in which parameters are updated in a random subspace, and prove that it converges to a first-order stationary point in a mean-square sense at a rate of $1/\sqrt{T}$ with a noisy gradient oracle. Brief experiments indicate slight improvements when using GoLore for the final 20% of training iterations on GLUE tasks with LLMs.

**Strengths:**

The paper provides theoretical results on the convergence of GaLore in both deterministic and specific stochastic settings, circumventing previous assumptions in the literature that required gradient projections to be unbiased and/or contractive. Interestingly, the projection matrices do not necessarily need to be updated at each iteration and can instead be reused across iterations, which may be applicable to other contexts that work with lossy gradient estimates. The counterexample demonstrating when GaLore fails to converge is clear, and the paper is well-motivated.

**Weaknesses:**

Overall, the discussion of GaLore’s shortcomings is unconvincing both theoretically and experimentally. While the authors present a clear example of when GaLore fails to converge—where gradient noise can be adversarially chosen to occur in a space orthogonal to the gradient of the function—their reasoning for this result is problematic. They claim that near first-order stationary points, the function gradient is dominated by the gradient noise, making it difficult for GaLore to find an informative direction. However, this intuition seems flawed, as in general, for any form of SGD, the gradient noise will dominate the function gradient near a first-order stationary point. The authors show that with a sufficient condition on batch sizes, GaLore will converge in the stochastic setting, effectively bypassing their own negative result. However, there is a lack of discussion on other conditions that could ensure the convergence of GaLore.

The experimental results show minor improvements when using GaLore followed by GoLore, with an average performance increase of 0.3% on GLUE relative to running GaLore alone. This setup does not align with the theoretical analysis or previous claims regarding GoLore’s superiority, as GaLore is still used for the majority of the training process.

Secondly, the language and presentation of the paper could be improved. The term “stationary solutions” is frequently used and appears to refer to first-order stationary points. Figure 1 is shown on page 2 but isn’t introduced until the end of page 5, and the y-axis label seems intended to represent log(f) in Equation (1) but is labeled simply as “Loss.” At the top of page 4, GaLore is described as a “subspace learning algorithm,” which is inaccurate. In the statement of Theorem 2, the use of the term MSGD and the expected value is confusing, as the source of stochasticity is unclear when the gradients are deterministic. Lastly, there are numerous typos that need correction, especially the spelling of “Optimization” in the title.

**Questions:**

**Questions**

Are there other mild conditions sufficient to prove the convergence of GaLore in the stochastic setting? Specifically, does a sufficiently small gradient norm suffice, or must the gradient noise tend to zero when computing the subspace? Additionally, what if the gradient noise is isotropic? The presented counterexample relies on the gradient noise occurring in a space orthogonal to the function gradient.

Regarding the GoLore analysis, is the GoLore projection contractive according to the definition in Section 2 of (Fatkhullin et al., 2024)? It seems a random projection should satisfy the contraction property. Also, how does the empirical performance of running only GaLore compare to running only GoLore? The theoretical claims suggest GoLore could replace GaLore entirely, so does GoLore outperform GaLore when used for all training iterations, or does it encounter issues when used independently?


**Comments**

While the paper’s motivation is intriguing, the narrative feels somewhat misguided. The claim that GaLore is unsatisfactory lacks strong support from both theoretical and experimental perspectives. The most compelling results are the positive convergence findings for GaLore in the deterministic and large-batch settings. The paper would be significantly improved by focusing on analyzing conditions under which GaLore converges with mild assumptions, rather than asserting that GaLore has fundamental flaws. Investigating the computational interplay between subspace update frequency, gradient noise, and step size would add valuable insights.

---

> ### Author Response · Authors · 2024-11-24
> **Response to Reviewer YUoQ (Part 1/4)**
>
> We thank the reviewer for the detailed comments. **First of all, we would like to clarify that we fully acknowledge the value of the GaLore method. In our view, GaLore is an excellent algorithm with novel insights and strong empirical performance. This inspired us to investigate and establish convergence guarantees for the great algorithm. However, in the proving process, we identified several aspects of its algorithmic structure that may hinder convergence. For this reason, we have clarified several conditions under which GaLore can converge and proposed a simple random projection technique to address its convergence issues. We believe our work strengthens the theoretical foundation and practical applicability of GaLore.**
>
> All questions raised by the reviewers have been clarified. We are glad to address any further comments or questions.
>
> **Weakness 1. The reasoning for why GaLore fails to converge seems flawed as in general, for any form of SGD, the gradient noise will dominate the function gradient near a first-order stationary point.**
>
> We respectfully note that the reviewer's interpretation of GaLore's failure to converge to a stationary solution requires clarification. While it is true that the gradient noise dominates the true gradient near a first-order stationary point in any SGD variant, this phenomenon is not the primary cause of non-convergence in GaLore. Rather, the non-convergence stems from the SVD projection inherent to GaLore's methodology. Specifically, **it is using SVD projection in the noise-dominant scenario that prevents GaLore from achieving convergence**.
>
> In particular, as the algorithm approaches the stationary solution and gradient noise begins to dominate the true gradient, the SVD-derived subspace may exclusively capture the noise component (particularly when selecting a sufficiently small $r$) and **completely discard the true gradient information**. Consequently, when no meaningful gradient information can be utilized, GaLore naturally fails to converge. This is the primary intuition underlying our counter-example presented in Equation (1).
>
> In contrast, **our proposed random projection consistently captures the true gradient information in expectation, regardless of whether the gradient noise dominates the true gradient or how small $r$ is chosen**. To see it, let $G$ denote the true gradient and $\epsilon$ represent the zero-mean random gradient noise. Additionally, we assume the random projection $P$ is independent of the gradient noise. By applying Lemma 5 in our manuscript, the gradient estimate satisfies
>
> $$
> \mathbb{E}[PP^T (G + \epsilon)] = \mathbb{E}[PP^T] G + \mathbb{E}[PP^T] \mathbb{E}[\epsilon] = \mathbb{E}[PP^T] G = \frac{r}{m} G.
> $$
>
> This derivation demonstrates that random projection consistently preserves gradient information, even in regimes where $\epsilon$ dominates $G$. In contrast, SVD projection fails to achieve this preservation property.
>
> **Weakness 2. A lack of discussion on other conditions that could ensure the convergence of GaLore.**
>
> Given that GaLore's non-convergence arises from employing the greedy SVD strategy in noise-dominated scenarios, the most natural solutions are either to avoid using the greedy SVD strategy or to reduce the noise scale, both of which are covered in our discussion. Additionally, the strong counter-example constructed in Theorem 1 rules out alternative approaches, such as introducing a "bounded gradient" assumption, modifying the optimizer $\rho$, or adjusting hyperparameters such as the learning rate $\eta$, momentum parameter $\beta_1$, or subspace update frequency $\tau$. While we agree that exploring other conditions under which GaLore can converge is an interesting open question, it is beyond the scope of this paper and could be left to future work.

---

> ### Author Response · Authors · 2024-11-24
> **Response to Reviewer YUoQ (Part 2/4)**
>
> **Weakness 3. The improvement in Table 2 is minor. Using GaLore for the majority process is not consistent with the theory.**
>
> (1) Results in Table 2
>
> We respectfully disagree that the improvement in Table 2 is minor. Though our methods present a 0.26 higher score than GaLore, it's worth noting that the performance gap between GaLore and full-parameter training is only 0.38. Regarding full-parameter training as the performance limitation, we have a 3 times smaller performance gap, which should not be regarded as minor. Please refer to our response to Point 3 in the global response for further details.
>
> In particular, we have conducted new experiments to fine-tune OPT-13B on the BoolQ dataset. As shown in Table 4 of Appendix E, GoLore achieves an accuracy of **81.96%**, notably higher than GaLore's accuracy of **79.79%**.
>
> Furthermore, we would like to emphasize that GoLore retains the algorithmic structure of GaLore, differing only in replacing the SVD projection with a random projection during the later phase of pre-training. **This simple modification introduces no additional computational overhead while providing strong convergence guarantees and consistently improved performance**. Given these advantages, we believe random projection is a valuable technique with both theoretical benefits and practical utility.
>
> (2) The hybrid of GaLore and GoLore is derived from our theory
>
> The hybrid of GaLore and GoLore does not contradict our theoretical findings regarding GaLore's non-convergence and GoLore's convergence. According to our theories, the SVD projection in GaLore is more efficient than the random projection in capturing the true gradient component in the early stages of pre-training, where the true gradient dominates the gradient noise. The advantage of using random projection becomes evident in the later stages of training, where the gradient noise dominates the true gradient. In short, **the SVD projection enables faster convergence in the initial training stages, while random projections facilitate more accurate convergence in the later stages**. This naturally leads to the hybrid GaLore + GoLore approach.
>
>
>
> **Weakness 4. Suggestions on improving the language and presentation, including:**
>
> **a) The term “stationary solutions” is frequently used and appears to refer to first-order stationary points.**
>
> We thank the reviewer for the suggestion. In order to avoid ambiguity, we included the following clarification: "By *stationary solutions*, we refer to first-order stationary points $x\in\mathbb{R}^d$ such that $\nabla f(x)=0$ for objective function $f:\mathbb{R}^d\rightarrow\mathbb{R}$."
>
> **b) Figure 1 is shown on page 2 but isn’t introduced until the end of page 5, and the y-axis label seems intended to represent log(f) in Equation (1) but is labeled simply as “Loss.”**
>
> The y-axis label is exactly the function value $f$ and is negative only because the global minimum of the qradratic function $f$ is negative. We have changed the y-axis to "$f(x)-f_{\min}$" to avoid ambiguity and added detailed descriptions in the caption of Figure 1.
>
> **c) At the top of page 4, GaLore is described as a “subspace learning algorithm,” which is inaccurate.**
>
> We refer to GaLore as a "subspace learning algorithm" because it trains LLMs by projecting gradients into subspaces, maintaining optimizer states within subspaces, and accumulating updates in subspaces. If the reviewer has specific concerns or suggestions about why it may not be accurately described as a "subspace learning algorithm," we would be happy to revise our terminology based on the feedback.
>
> **d) In the statement of Theorem 2, the use of the term MSGD and the expected value is confusing, as the source of stochasticity is unclear when the gradients are deterministic.**
>
> We thank the reviewer for the insightful suggestion. There is no stochasticity in this setting and we have removed the expectation, including those in the results or proofs related to deterministic GaLore/GaSare. The term MSGD used is consistent with Algorithm 1 and it might be wordy to use a new term like "MGD" instead, hence we remarked below Theorem 2 to avoid any possible ambiguity: "In fact, MSGD here reduces to momentum gradient descent by using deterministic gradients".
>
> **e) There are numerous typos that need correction, especially the spelling of “Optimization” in the title.**
>
> We sincerely apologize for all the typos which may bring the reviewer a poor reading experience, and had made our best to fix these typos in our revised manuscripts.

---

> ### Author Response · Authors · 2024-11-24
> **Response to Reviewer YUoQ (Part 3/4)**
>
> **Question 1. Whether other mild conditions are sufficient to prove the convergence of GaLore in the stochastic setting, e.g., a  sufficiently small gradient norm, assuming the gradient noise is isotropic.**
>
> We greatly appreciate the reviewer’s insightful comments on isotropic noise. While conditions such as a bounded gradient norm or tuned hyperparameters may seem promising, they are insufficient to resolve the convergence issue, as our counter-example remains valid even under these settings.
>
> Assuming isotropic gradient noise is an intriguing perspective, and indeed, it provides a defense against our counter-example. This opens an interesting direction for future research—whether GaLore converges under isotropic noise remains an open question. However, it is widely recognized within the community that gradient noise in SGD is typically anisotropic, as evidenced by studies such as [arXiv:1803.00195], [arXiv:2006.08680], [arXiv:2105.09557], [arXiv:2207.02628], and [arXiv:2310.00692]. This raises a concern about the practical relevance of studying GaLore’s convergence under the isotropic noise assumption.
>
> On the other hand, our proposed solution—using random projections—addresses GaLore's convergence issue in both isotropic and anisotropic noise settings. This approach provides a more robust and broadly applicable solution to the problem.
>
> **Question 2. Is the GoLore projection contractive according to the definition in Section 2 of (Fatkhullin et al., 2024)? Also, how does the empirical performance of running only GaLore compare to running only GoLore?**
>
> (1) Contractive property
>
> The contractive property only holds for $\tau\mid t$ and does not hold for $\tau\nmid t$. This is because the reused projection matrices are no longer random given the filtration at iteration  $t$ $(\tau\nmid t)$.
>
> (2) GoLore@100%
>
> In most of our experiments, we find that GoLore@100% is not comparable to GaLore. However, **this does not contradict our theoretical findings** regarding GaLore's non-convergence and GoLore's convergence. According to our theories, the SVD projection in GaLore is more efficient than the random projection in capturing the true gradient component in the early stages of pre-training, where the true gradient dominates the gradient noise. The advantage of using random projection becomes evident in the later stages of training, where the gradient noise dominates the true gradient. In short, **the SVD projection enables faster convergence in the initial training stages, while random projections facilitate more accurate convergence in the later stages**. This naturally leads to the hybrid GaLore + GoLore approach. This further explains why GoLore@100% fails to surpass GaLore in performance. The slower convergence of GoLore@100%, particularly during the initial training stages, adversely affects its fine-tuning performance.

---

> ### Author Response · Authors · 2024-11-24
> **Response to Reviewer YUoQ (Part 4/4)**
>
> **Comments. While the paper’s motivation is intriguing, the narrative feels somewhat misguided. The claim that GaLore is unsatisfactory lacks strong support from both theoretical and experimental perspectives. The most compelling results are the positive convergence findings for GaLore in the deterministic and large-batch settings. The paper would be significantly improved by focusing on analyzing conditions under which GaLore converges with mild assumptions, rather than asserting that GaLore has fundamental flaws. Investigating the computational interplay between subspace update frequency, gradient noise, and step size would add valuable insights.**
>
> First of all, we would like to clarify that we fully acknowledge the value of the GaLore method. In our view, GaLore is an excellent algorithm with novel insights and strong empirical performance. This inspired us to investigate and establish convergence guarantees for the great algorithm. However, in the proving process, we identified several aspects of its algorithmic structure that may hinder convergence. For this reason, we have clarified several conditions under which GaLore can converge and proposed a simple random projection technique to address its convergence issues. We believe our work strengthens the theoretical foundation and practical applicability of GaLore.
>
> Second, we respectfully disagree with the claim that our theoretical results are misguided. Both our theoretical analysis and experimental evidence clearly demonstrate that GaLore lacks convergence guarantees **under standard assumptions**. By “standard assumptions,” we refer to the conditions typically required for vanilla SGD to ensure convergence. Our results show that there exist scenarios where vanilla SGD converges, but GaLore does not. This highlights a theoretical limitation of the GaLore algorithm.
>
> Third, we have already provided conditions under which GaLore can converge. However, the strong counter-example constructed in Theorem 1 rules out alternative approaches, such as introducing a "bounded gradient" assumption, modifying the optimizer $\rho$, or adjusting hyperparameters such as the learning rate $\eta$, momentum parameter $\beta_1$, or subspace update frequency $\tau$.
>
> Finally, we believe that identifying simple yet effective modifications to GaLore is an important contribution to addressing its non-convergence issues. GoLore, as a direct variant of GaLore, remains within the same algorithmic family. The proposed random projection technique, which introduces no additional computational overhead, provides robust convergence guarantees and consistently improves performance. We view these adjustments as crucial for enhancing the practical and theoretical robustness of the GaLore framework.
>
> We thank the reviewer again for his careful comments. We hope these responses can clarify the reviewer's questions and are more than happy to address any further comments or questions.

---

> > ### Comment · Reviewer_YUoQ · 2024-11-25
> >
> > Thank you for responding to my comments. I have summarized the remaining points of my review below.
> >
> > **Intuition and Discussion on why GaLore can fail to converge**
> >
> > It seems like the real intuition for why GaLore can fail to converge is that taking the top-k SVD projection is not always an unbiased estimator for the true gradient. Even by setting the subspace update frequency $\tau = 1$ for GaLore, this problem will persist, so although the periodic subspace update adds an additional complication to analyzing GaLore, it is not the cause of GaLore’s potential non-convergence. This simple explanation is not clearly presented in the paper. In the first bullet in the “Contributions” section of the Introduction, it reads “The key insight is that the SVD-derived subspace primarily captures the noise component rather than the true gradient in scenarios where gradient noise predominates.” Based on the paper and your rebuttal, this intuition is incorrect as in any SGD method the gradient estimate will primarily capture the noise component when the gradient norm is small. The same misleading interpretation is written in Section 1.1 in the paragraph which begins “Contrary to expectations…” Further, in the beginning of Section 3, additional discussion of this intuition is ambiguous. The paper would be much improved by concretely stating the fact that the SVD projection is not always an unbiased estimator, a fact which you exploit when constructing your counter-example, and removing the other discussion of the intuition which does not capture this subtle point which lies at the heart of your analysis.
> >
> > Further, many lines in the paper seem to suggest much stronger results than what is ultimately proven, which adds to the confusion on what the exact shortcomings of GaLore are and how GoLore can fix them. In the first bullet of the Contributions section of the Introduction, the line “We find that GaLore cannot converge to a stationary solutions under regular assumptions” seems to suggest GaLore **never** converges under regular assumptions, but your point is that GaLore **may not** converge in general which you prove by providing your counter-example. Further, the line in Section 1.1 that reads “Contrary to expectations, our investigation reveals that GaLore does **NOT** converge…” is misleading for the same reason.
> >
> > **Other conditions for which GaLore can converge**
> >
> > Thank you for the clarification on what other assumptions may guarantee the convergence of GaLore. I think the discussion on the implications of isotropic noise is important. I am not convinced that it is okay to disregard this discussion simply because “typical” SGD gradient noise is anisotropic, as the gradient oracle used in your counter-example is certainly not representative of “typical” SGD gradient noise anyway. Overall, I find the theoretical results the most compelling aspect of this paper, and since isotropic noise is both a common assumption in machine learning and seems to play a large role for such algorithms which use compressed gradients, I think it warrants a serious discussion. It would also help clarify the intuition on when GaLore may not converge, as performing GaLore with $\tau=1$ under isotropic noise would seem to lead to unbiased gradient estimates, which is a central aspect of the GaLore non-convergence intuition.
> >
> > **Experimental improvements using GoLore**
> >
> > Thank you for providing additional experimental results. I think the paper would be improved by presenting those results in the body and removing other sections to the appendix, like section 6. Overall, GoLore clearly improves upon GaLore when close to first-order stationary points (FOSPs), but the differences are often quite small. This is somewhat expected, as the analysis shows that GoLore only helps when the iterates are already close to an FOSP and thus close to locally optimal. For the GLUE results, which are the main results in the paper’s body, you mention that GoLore achieves a .12% performance gap relative to full fine-tuning, while GaLore incurs a .38% gap which is more than 3 times worse. However, since the full fine-tuning performance is 86.15%, this means GaLore recovers 99.56% of full fine-tuning performance and GoLore recovers 99.86%. Although GoLore does perform better, I do not think these are the best results for demonstrating the usefulness of GoLore, especially relative to the other stronger results in the appendix.

---

> > > ### Comment · Reviewer_YUoQ · 2024-11-25
> > >
> > > **Overall narrative**
> > >
> > > Although the theoretical results are interesting, I think the paper needs substantial revision. Much of the language is imprecise and confusing when discussing the negative result of when GaLore may not converge as well as the intuition of why GaLore may not converge. These are central points to the motivation of the paper and are not presented clearly in the cases I mentioned above. The theory would be strengthened with a discussion of the impact of isotropic noise, as it plays a central role in understanding the cases where GaLore may fail to converge. Lastly, the experimental results are positive but not very strong. I think a larger focus on the theory would be beneficial, especially since learning algorithms which use compressed gradients are of interest not just for LLM training but also in other contexts like federated learning.
> > >
> > > **Other: Meaning of “subspace learning algorithm”**
> > >
> > > Typically this term would be used for an algorithm like PCA whose goal is to learn a subspace. Neither GaLore nor GoLore fit this description.

---

> > > > ### Author Response · Authors · 2024-12-01
> > > > **Further Response to Reviewer YUoQ (Part 1/3)**
> > > >
> > > > We sincerely thank the reviewer for the thoughtful comments and valuable suggestions.
> > > >
> > > > **(1) Intuition and Discussion on why GaLore can fail to converge**
> > > >
> > > > (a) Real intuition behind GaLore's non-convergence
> > > >
> > > > Thank you for bringing up this important discussion. During our rebuttal with the reviewer, the intuition behind GaLore's non-convergence became clearer. We do not fully agree that the bias in GaLore's SVD projection is the primary source of non-convergence. Instead, we believe the core issue is that **GaLore can completely miss the gradient information** during the training process. This happens when the following three conditions hold simultaneously:
> > > >
> > > > - (C1) Anisotropic gradient noise
> > > > - (C2) Gradient noise dominates the true gradient
> > > > - (C3) **Greedy** and **biased** SVD projection
> > > >
> > > >
> > > > When any of the conditions above are violated, GaLore can exhibit convergence guarantees. For instance:
> > > >
> > > > - (D1) When the gradient noise is isotropic, we have demonstrated in our revised manuscript (see Appendix G) that GaLore has convergence guarantees.
> > > > - (D2) When the true gradient dominates the gradient noise, as in noise-free or large-batch scenarios, we have shown in Section 4 that GaLore has convergence guarantees.
> > > > - (D3) When the SVD projection is replaced with a random projection, we have established in Section 5 that GaLore has convergence guarantees.
> > > >
> > > > Please note that in (C3), being both biased and greedy simultaneously in the SVD projection leads to GaLore's non-convergence. However, being biased alone can still result in GaLore's convergence. For example, randomly applying GaLore's SVD projection with a probability of $1/\sqrt{T}$ and GoLore's random projection with a probability of $1 - 1/\sqrt{T}$ is a biased strategy that may still enable convergence. This is because this construction can still capture true gradient information in expectation.
> > > >
> > > > (b) Statement revision
> > > >
> > > > In response to reviewer's concerns on our intuition, we will revise the statement as follows, making them more precise.
> > > >
> > > > >**Original:** “The key insight is that the SVD-derived subspace primarily captures the noise component rather than the true gradient in scenarios where gradient noise predominates.”
> > > > >
> > > > >**Revised:** “The key insight is that the SVD projection is biased and greedy; it may completely lose the true gradient information during the training process when the gradient noise is anisotropic and the noise magnitude dominates the true gradient.”
> > > >
> > > >
> > > > >**Original:** “We find that GaLore cannot converge to stationary solutions under regular assumptions”
> > > > >
> > > > >**Revised:** “We find that GaLore is not theoretically guaranteed to converge to stationary solutions under Assumptions 1-3.”
> > > >
> > > > >**Original:** “Contrary to expectations, our investigation reveals that GaLore does NOT converge ...”
> > > > >
> > > > >**Revised:** “Contrary to expectations, our investigation reveals that GaLore is NOT theoretically guaranteed to converge ...”
> > > >
> > > >
> > > > >**Original:** "In the extreme case, this noise-dominated subspace can become orthogonal to the true gradient subspace, leading to non-convergence."
> > > > >
> > > > >**Revised:** "When gradient noise is anisotropic, the noise-dominated subspace captured by SVD may become orthogonal to the true gradient subspace due to its greedy nature, leading to non-convergence."

---

> > > > ### Author Response · Authors · 2024-12-01
> > > > **Further Response to Reviewer YUoQ (Part 2/3)**
> > > >
> > > > **(2) Additional theoretical results on GaLore's convergence under isotropic noise assumptions.**
> > > >
> > > > We thank the reviewer for the valuable insights regarding isotropy. Following further investigation, we have successfully extended our theoretical analysis to establish GaLore's convergence guarantees under **isotropic noise** assumptions.
> > > >
> > > > >**Theorem 13 (Convergence of GaLore under isotropic noise assumptions).** Under **isotropic noise** and other assumptions, if hyperparameters
> > > > >$$0<\beta\_1\le1,\quad\tau\ge\frac{128}{3\beta\_1\underline{\delta}},\quad 0<\eta\le\min\left\\{\frac{1}{4L},\sqrt{\frac{3\underline{\delta}\beta\_1^2}{80L^2}},\sqrt{\frac{3\underline{\delta}}{80\tau^2L^2}},\sqrt{\frac{3\beta\_1}{32\tau L^2}}\right\\},$$
> > > > >
> > > > >GaLore using small-batch stochastic gradients and MSGD with MP converges as
> > > > >
> > > > >$$\frac{1}{K\tau}\sum_{t=0}^{K\tau-1}\mathbb{E}[\|\nabla f(x)\|_2^2]\le\frac{16\Delta}{\underline{\delta}\eta K\tau}+\frac{32\beta_1\sigma^2}{3\underline{\delta}}$$
> > > > >
> > > > >for any $K\ge1$, where $\Delta=f(x^{(0)})-\inf_xf(x)$.
> > > >
> > > > >**Corollary 4 (Convergence complexity of GaLore under isotropic noise assumptions).** Under **isotropic noise** and other assumptions, if $T\ge2+256/(3\underline{\delta})+(256\sigma)^2/(9\sqrt{\underline{\delta}L\Delta})$ and we choose
> > > > >$$\beta_1=\left(1+\sqrt{\frac{\underline{\delta}^{3/2}\sigma^2T}{L\Delta}}\right)^{-1},\quad \tau=\left\lceil\frac{128}{3\underline{\delta}\beta_1}\right\rceil,\quad\eta=\left(4L+\sqrt{\frac{80L^2}{3\underline{\delta}\beta_1}}+\sqrt{\frac{80\tau^2L^2}{3\underline{\delta}}}+\sqrt{\frac{32\tau L^2}{3\beta_1}}\right)^{-1},$$
> > > > GaLore using small-batch stochastic gradients and MSGD with MP converges as
> > > > $$\frac{1}{T}\sum_{t=0}^{T-1}\mathbb{E}[\|\nabla f(x^{(t)})\|_2^2]=\mathcal{O}\left(\frac{L\Delta}{\underline{\delta}^{5/2}T}+\sqrt{\frac{L\Delta\sigma^2}{\underline{\delta}^{7/2}T}}\right),$$
> > > > where $\Delta=f(x^{(0)})-\inf_xf(x)$. Consequently, the computation complexity to reach an $\varepsilon$-accurate solution $x$ such that $\|\nabla f(x)\|_2^2\le\varepsilon$ is $\mathcal{O}\left(\frac{L\Delta\sigma^2}{\underline{\delta}^{7/2}\varepsilon^2}+\frac{L\Delta}{\underline{\delta}^{5/2}\varepsilon}+\frac{\sigma^2}{\underline{\delta}^{1/2}L\Delta}+\frac{1}{\underline{\delta}}\right)$.
> > > >
> > > > More details, including the additional assumptions, lemmas and proofs,  can be found in Appendix G in the latest revised version of our paper.
> > > >
> > > > This new finding significantly strengthens our theoretical contributions. Based on this result and our prior theoretical analysis, we conclude that GaLore's non-convergence arises from the interplay of the following factors: (1) reliance on **SVD projection**, (2) the presence of potentially **anisotropic noise**, and (3) **noise-dominated** scenarios. Addressing each of these factors individually leads to GaLore variants with convergence guarantees: (1) GaLore with random projection (GoLore), (2) GaLore with full-batch or large-batch gradients, and (3) GaLore under isotropic noise assumptions, respectively.
> > > >
> > > > Importantly, this new finding does not contradict our previous conclusions that GaLore may lack convergence guarantees under Assumptions 1–3. Our original analysis was based on general gradient noise assumptions (Assumption 3) without explicitly distinguishing between isotropic and anisotropic noise. To enhance clarity and reduce potential misunderstandings, we will revise the term "noise-dominated scenarios" to **"potentially anisotropic noise-dominated scenarios"** throughout the paper. Additional revisions can be found in our response to Point (1).
> > > >
> > > > We hope that this newly added theory under isotropic noise can address the reviewer’s concerns.
> > > >
> > > > **(3) Experimental improvements using GoLore.**
> > > >
> > > > We thank the reviewer for acknowledging our additional experiments. However, we respectfully disagree with the reviewer’s perspective and believe that our experimental results on the GLUE benchmark effectively demonstrate the utility of our method.
> > > >
> > > > We understand the concern that GoLore improves the performance recovery of full fine-tuning from 99.56% to 99.96%, which may seem marginal. However, it is worth noting that in GaLore's original paper, the improvement upon LoRA was even smaller, from 99.59% to 99.61%, under the same metric. Despite our disagreement, we greatly value the reviewer’s suggestion to highlight more demonstrative results in the main text.
> > > >
> > > > To address this, we will move Section 6 to Appendix F and bring the experimental results from Table 4 and Figure 7 into the main text, ensuring that the impact of our method is more prominently showcased.

---

> > > > ### Author Response · Authors · 2024-12-01
> > > > **Further Response to Reviewer YUoQ (Part 3/3)**
> > > >
> > > > **(4) Avoiding misleading terminology.**
> > > >
> > > > We appreciate the reviewer’s clarification regarding the inaccuracy of referring to GaLore as a "subspace learning algorithm." We have replaced this term with "subspace descent algorithm" throughout the manuscript, which we believe better aligns with the method’s nature and should effectively address the reviewer’s concern.
> > > >
> > > > Additionally, we would greatly appreciate it if the reviewer could provide specific examples of any other imprecise or confusing language in the paper. This feedback will help us make further improvements and ensure clarity for all readers.
> > > >
> > > > Once again, we thank the reviewer for the insightful feedback and are happy to engage in further discussions.

---

> > > > > ### Author Response · Authors · 2024-12-03
> > > > > **Follow-up on Your Concerns Regarding Our Submission**
> > > > >
> > > > > Dear Reviewer YUoQ,
> > > > >
> > > > > Thank you for your feedback on our paper. We have carefully addressed your comments in our rebuttal and provided detailed responses to the concerns you raised.
> > > > >
> > > > > As the discussion phase will conclude in less than 12 hours, we wanted to follow up to kindly ask if our responses have addressed your concerns. If there are any remaining points of clarification or further questions, we would be happy to respond promptly before the rebuttal period ends.
> > > > >
> > > > > Your feedback is very important to us, and we truly value your time and effort in reviewing our submission.
> > > > >
> > > > > Best regards,
> > > > >
> > > > > The authors of Submission 5448

---

> > > > ### Author Response · Authors · 2024-12-02
> > > > **Can we have your comments on our new rebuttal responses?**
> > > >
> > > > Dear reviewer YUoQ,
> > > >
> > > > We are grateful for your insightful comments and feedback. We have carefully addressed each of your concerns in our detailed responses. Given the impending rebuttal deadline, we would kindly request your review of our responses to ensure that all issues have been adequately resolved. If any further clarification or elaboration is required, we would be more than happy to provide additional explanations. Thank you for your time and consideration.
> > > >
> > > > Best regards,
> > > >
> > > > The authors of Submission 5448

---

### Official Review · Reviewer_FSJZ · 2024-11-04

**Soundness:** 3
**Presentation:** 3
**Contribution:** 3
**Rating:** 6
**Confidence:** 4

**Summary:**

The paper investigates into the convergence behaviour of GaLore and finds that GaLore only converges when their is no gradient noise or the batch size is large enough such that the gradient noise is small. For the stochastic setting without large batch, the paper provides one counter example where GaLore fails to converge. To address the issue of non-convergence, the paper proposes to use GoLore that replaces the projection matrix in GaLore with random matrices. Convergence analysis as well as experiments on language models are provided to support the effectiveness of the proposed GoLore algorithm.

**Strengths:**

Training language model requires large amount of resources. The research on efficient and cheaper algorithms for pretraining and fine-tuning language models is thus important and valuable to the entire community. Previous works that propose efficient methods like LoRA and GaLore focus on empirical performance, while this work aims at building theoretical foundations on the convergence analysis of GaLore. The theoretical results then guide the design of a new algorithm, GoLare. This paper is well written and clear, providing novel and important insights on a trendy and significant area on memory efficient methods for training language models.

**Weaknesses:**

1. There is a typo in the title. It should be optimization instead of optimiztion. At first impression, I feel the paper lacks careful proofreading and attention to detail. Given that most results are theoretical, I also have reasonable questions towards the overall quality and rigor of the work. Therefore, I recommend the authors to carefully reviewing the paper. Other typos: line 198 effiicent, line 347 large-batch is the same as stochastic. Other minor suggestions: (1) pre-training and pretraining are used interchangeably. It is better to be consistent. (2) I was very confused when first reading Figure 1. It might be good to at least explain what is the loss, MSGD, L.B., @50%, etc., in Figure 1's title. (3) line 335-338 and line 344-347 in Algorithm 1 are exactly the same. There is no need to repeat twice. It saves space to move these three lines outside the if-else block, e.g., $G\leftarrow$..., if ... then SVD ... else $P^t\leftarrow P^{t-1}$... end if. (4) I guess that the second moments $V$ in Algorithm 1 can also be implemented with projection in the same way as MP. Why not also consider it?

2. The theoretical results suggest that GaLore converges with large batch size while fails for the stochastic case. It is not clear how large this batch size should be. Theorem 3 only provides one case where batch size $B\sim\sqrt{T}$. What happens for other batch size? Would anything smaller lead to non-convergence? Theorem 1 is also not clear in this sense. The non-convergence happens when $B=1$. What happens for other choice of batch size? The specific question is that is there any condition, threshold, or phase transition for $B$ such that anything smaller leads to divergence and anything larger leads to convergence. The current large batch v.s. stochastic setting is not super clear.

3. I assume GoLore@100%'s performance is not comparable with GaLore based on the intuition the authors provide for using GaLore at the beginning and GoLore at the end. If the performance of GoLore@100% is comparable, the authors should also report it in the experiments. Otherwise, this seems to be contradictory to the theoretical results that GaLore diverges while GoLore converges.

4. Besides ReLoRA, FLoRA, and SIFT, there are also some similarity between GoLore and zeroth-order methods with directional derivatives, and GoLore and gradient sketching, e.g., [arXiv: 1809.03054] and [Can Gaussian Sketching Converge Faster on a Preconditioned Landscape? ICML 2024]. It is better to also include such comparisons. Paticularly, zeroth-order methods are also popular for LLMs fine-tuning because of its memory-efficiency, see [arXiv: 2305.17333], [arXiv: 2310.09639], and [arXiv: 2410.07698].

5. The language models used in the experiments are not "large" language models, as Figure 3 is only 60M and Table 2 is 125M. Figure 4 is the curve for fine-tuning LLaMA2-7B, but no test performance like Table 2 is given. Also, the improvement of GoLore in Table 2 is only marginal.

**Questions:**

1. In Theorem 1, what does it mean for any subspace optimizer $\rho$? What if $\rho$ just maps $P^\top G$ to the stochastic gradient $G$? In this case, the algorithm should still converge.

2. In Theorem 2, $\beta_1=1$ just reduces to SGD without momentum. What happens for other $\beta_1$? Also, in Theorem 3 and 4, what happens for other choice of $\beta_1$? I think $\eta\leq...$ should work instead of being the exact value. Is it the same for $\beta_1$ or the results only hold for only one specific choice?

3. The loss in LLMs training is usually very noisy. Are the curves in Figures 3 and 4 already smoothed?

4. What are the batch sizes used in Figure 1 for all four different algorithms?

---

> ### Author Response · Authors · 2024-11-23
> **Response to Reviewer FSJZ (Part 1/3)**
>
> We thank the reviewer for careful and valuable comments. All questions have been addressed as best as we can, and we are glad to address further comments or questions.
>
> **Weakness 1. Typos 'optimiztion', 'effiicent', 'pretraining', Algorithm 1 and whether we can apply projection to second-order momentum $V$.**
>
> (1) Clarification on typos
>
> We thank the reviewer for identifying these spelling errors, which we have corrected accordingly. Additionally, we have followed the suggestion to provide more details about Figure 1 in its captions. We sincerely appreciate the reviewer’s thorough and thoughtful review.
>
> However, the equation on line 347 in Algorithm 1 is not a typo. In the large-batch scenario, the large-batch gradient is used solely to compute a precise SVD projection matrix, as shown in lines 338–340. Once the SVD projection matrix is computed, the large-batch gradient is no longer required. Instead, we use the standard stochastic gradient (line 347) to accumulate momentum and update the model. Since **the gradients used in lines 338 and 347 differ**, we have placed them in separate if-else blocks for clarity.
>
> Since the above questions are only three spelling mistakes, we wish the reviewer could trust our sense of responsibility on the quality and rigor of our results, and we are open to answer any related questions.
>
> (2) Project second-order momentum $V$ into low-rank subspace
>
> We thank the reviewer for the insightful question. While it is feasible to project the second-order momentum $V$ into a low-rank subspace in practical experiments, establishing convergence guarantees for such an approach presents significant challenges, particularly when the projection matrix is updated lazily. We consider this an important direction for future work and aim to develop convergence guarantees for algorithms that project the second-order momentum $V$ into a low-rank subspace.
>
> **Weakness 2. Is there any condition, threshold, or phase transition for $B$ such that anything smaller leads to divergence and anything larger leads to convergence.**
>
> (1) Large-batch GaLore can converge with any batch-size $\mathcal{B}\sim T^{a}, a > 0$
>
> In Theorem 7 in Appendix B.4, we provide the convergence rates of the large-batch GaLore with any batch-size $\mathcal{B}$:
>
> **Theorem 7.** Given any $0<\beta\_1\le1$, $\tau\ge\frac{64}{3\beta\_1\underline{\delta}}$ and $0<\eta\le\min\left\\{\frac{1}{4L},\sqrt{\frac{3\underline{\delta}\beta\_1^2}{80L^2}},\sqrt{\frac{\underline{\delta}}{40\tau^2L^2}},\sqrt{\frac{3\beta\_1}{32\tau L^2}}\right\\}$, large-batch GaLore converges as
> $$\frac{1}{K\tau}\sum\_{t=0}^{K\tau-1}\mathbb{E}[\\|\nabla f(x^{(t)})\\|\_2^2]\le\frac{16\Delta}{\underline{\delta}\eta K\tau}+\left(\frac{160}{3\beta\_1\underline{\delta}\tau\mathcal{B}}+\frac{352}{3\underline{\delta}^2\mathcal{B}}+\frac{32\beta\_1}{3\underline{\delta}}\right)\sigma^2.$$
>
> According to Theorem 7, when we choose hyper-parameters as $\mathcal{B}\sim T^{a}$, $\tau\sim T^{b}$, $\beta\_1\sim T^{-b}$ for any $a>0$ and $0<b<1$, large-batch GaLore **can converge to the solution** at the convergence rate of $\mathcal{O}(\frac{1}{T^{\min\\{a,b,1-b\\}}})$. This result implies that **large-batch GaLore can converge for any batch size satisfying $\mathcal{B} \sim T^a$ with any positive $a$**, provided the other hyperparameters $\tau$, $\beta_1$, and $b$ are chosen appropriately.
>
> In the main body of our manuscript, we presented the convergence results for $\mathcal{B} \sim \sqrt{T}$ because:  a) Setting $\mathcal{B}, \tau \sim \sqrt{T}$ and $\beta_1 \sim 1/\sqrt{T}$ achieves the optimal convergence rate of $\mathcal{O}(1/\sqrt{T})$. Setting other values of $\mathcal{B}$ would result in a theoretically slower convergence rate. b) The total computational complexity with $\mathcal{B} \sim \sqrt{T}$ is $\Theta(1)$ per iteration, as it is utilized only every $\tau \sim \sqrt{T}$ iterations. This complexity is no higher than that achieved with $\mathcal{B} \sim T^a$ for smaller $a$.
>
> (2) Large-batch GaLore cannot converge with batch-size $\mathcal{B}=\mathcal{O}(1)$
>
> On the other hand, large-batch GaLore with any **constant batch size** $\mathcal{B} = \mathcal{O}(1)$ is not guaranteed to converge. Specifically, Theorem 1 establishes the non-convergence property of GaLore under an arbitrary constant gradient variance $\sigma^2$. While using a constant batch size $\mathcal{B} = \mathcal{O}(1)$ reduces the variance $\sigma^2$ to $\sigma^2 / \mathcal{B}$, this remains a constant variance, and Theorem 1 confirms that this does not ensure convergence for any constant $\mathcal{B}$.

---

> ### Author Response · Authors · 2024-11-23
> **Response to Reviewer FSJZ (Part 2/3)**
>
> **Weakness 3. GoLore@100% is assumed to be not comparable with GaLore according to the intuition, which seems contradictory with GaLore's non-convergence and GoLore's convergence.**
>
> In most of our experiments, we find that GoLore@100% is not comparable to GaLore. However, **this does not contradict our theoretical findings** regarding GaLore's non-convergence and GoLore's convergence. According to our theories, the SVD projection in GaLore is more efficient than the random projection in capturing the true gradient component in the early stages of pre-training, where the true gradient dominates the gradient noise. The advantage of using random projection becomes evident in the later stages of training, where the gradient noise dominates the true gradient. In short, **the SVD projection enables faster convergence in the initial training stages, while random projections facilitate more accurate convergence in the later stages**. This naturally leads to the hybrid GaLore + GoLore approach. This further explains why GaLore@100% fails to surpass GaLore in performance. The slower convergence of GaLore@100%, particularly during the initial training stages, adversely affects its fine-tuning performance.
>
> **Weakness 4. It is better to also include comparisons with zero-th order or gradient sketching methods.**
>
> (1) Connection with zero-th order methods.
>
> Zero-th order methods  (Malladi et al., 2023; Zhang et al., 2023; Chen et al., 2024)  are another line of works on memory-efficient training. While these algorithms randomly select a direction to estimate the directional derivatives by finite difference, GoLore computes subspace gradients via back propagation. The descent directions used in zero-th order methods change every iteration, while GoLore applies a more lazily strategy changing its subspace every $\tau$ iterations.
>
> (2) Connection with gradient sketching methods.
>
> Gradient sketching methods like  Hanzely et al.(2018) and Wang et al. (2024)  uses gradient sketches in algorithm iterates. These methods recover gradient estimates from projected gradients and retains **full-size** gradients and optimizer states. In comparison, GoLore directly updates with projected gradients and retains compressed gradients and optimizer states, which is more memory-efficient.
>
> Following the reviewer’s valuable suggestion, we have incorporated the related discussions into Appendix F in our revised manuscript.
>
> **Weaknesss 5. The models in the experiments are not "large". Improvement in Table 2 is marginal. Accuracy for Figure 4 is missing.**
>
> (1) Models are not "large".
>
> Given our limited computing resources, we were unable to conduct experiments on very large language models. The model sizes we tested are common in the literature. For instance, GaLore (ICML 2024 Oral) and Flora (ICML 2024) both report empirical results using 60M models. While GaLore and our paper use a 7B model as the largest size, Flora only mentions the memory usage for 7B models, noting that "the training takes months to complete." To further address the reviewer's concerns, we conducted additional experiments on the OPT-13B model, and the results can be found in Appendix E, Table 4, of our revised manuscript.
>
> (2) Improvement in Table 2.
>
> Since the main contribution of this paper lies in establishing convergence guarantees for memory-efficient algorithms, we believe our experimental results have validated our theoretical findings. Please refer to our response to Point 3 in the global response for further details.
>
> In particular, we have conducted new experiments to fine-tune OPT-13B on the BoolQ dataset. As shown in Table 4 of Appendix E, GoLore achieves an accuracy of 81.96%, notably higher than GaLore's accuracy of 79.79%.
>
> Furthermore, we would like to emphasize that GoLore retains the algorithmic structure of GaLore, differing only in replacing the SVD projection with a random projection during the later phase of pre-training. **This simple modification introduces no additional computational overhead while providing strong convergence guarantees and consistently improved performance**. Given these advantages, we believe random projection is a valuable technique with both theoretical benefits and practical utility.

---

> ### Author Response · Authors · 2024-11-23
> **Response to Reviewer FSJZ (Part 3/3)**
>
> **Question 1.** What does $\rho$ mean in Theorem 1? Can $\rho$ just map $P^\top G$ to $G$ so that the algorithm can still converge?
>
> We thank the reviewer for raising this valuable question. In our previous definition, we defined $\rho$ following the convention of the GaLore's paper as a stateful entry-wise gradient operator. This means that, the inputs and outputs of $\rho$ are in the $\mathbb{R}^{r_{\ell} \times n_{\ell}}$ subspace. Mapping $P^\top G$ to $G$ would violate the dimensions and lead to illegal operations within the GaLore algorithm. For the rigor of Theorem 1, we have clarified this point by adding the following: "... subspace optimizer $\rho$ takes as input the subspace gradient of shape $r_{\ell} \times n_{\ell}$ and outputs the subspace update direction of shape $r_{\ell} \times n_{\ell}$..."
>
> **Question 2. Convergence results related to other $\beta_1$ and $\eta$.**
>
> We have established convergence guarantees for a broad range of $\beta_1$ and $\eta$, as detailed in Theorem 6, Theorem 7 and Theorem 8 in Appendix B.2, B.3, B.4, respectively. Please also refer to our response to Weakness 2 raised by you, where we establish convergence for the following conditions: $0 < \beta_1 \le 1$, $\tau \ge \frac{64}{3 \beta_1 \underline{\delta}}$, and $0 < \eta \le \min\left\\{\frac{1}{4L}, \sqrt{\frac{3 \underline{\delta} \beta_1^2}{80 L^2}}, \sqrt{\frac{\underline{\delta}}{40 \tau^2 L^2}}, \sqrt{\frac{3 \beta_1}{32 \tau L^2}}\right\\}$.
>
> **Question 3. The loss in LLMs training is usually very noisy. Are the curves in Figures 3 and 4 already smoothed?**
>
> The curves in Figure 3 are smooth because we only present the validation loss, which is computed every 500 iterations. It is true that we smoothed the curves in Figure 4 by averaging every 15 consecutive loss values for better presentation.
>
> **Question 4. What are the batch sizes used in Figure 1 for all four different algorithms?**
>
> We use a batch size of 128 for large-batch GaLore and a batch size of 1 for the other algorithms, including GaLore, GoLore, and full-parameter training. We thank the reviewer for pointing out this issue and have addressed it in our updated manuscript.
>
> We thank the reviewer once again for their valuable comments. We hope that these responses help clarify the reviewer's questions, and we are happy to address any further comments or inquiries.

---

> > ### Comment · Reviewer_FSJZ · 2024-11-25
> >
> > Many thanks for the response and the effort trying to address my concerns. While I am not fully convinced by the reply to Weakness 3, my other concerns are properly addressed. I will increase my score to 6.
> >
> > I understand the explanation. I agree that if we can enter the second phase of training, then GoLore performs better than Galore. This is verified in both theory and experiments, and is explained by the intuition that noise dominates. However, according to experiments, GoLore is worse than GaLore on entering this phase. This is something not aligned with the theory and can only be explained by the intuition that SVD gives more information in the early stage when gradients dominate. Specifically, the convergence guarantees of GaLore (large batch) and GoLore are basically the same, and GaLore is even worse in the stochastic case.

---

> > > ### Author Response · Authors · 2024-11-29
> > >
> > > We sincerely thank you for acknowledging our response to the previous concerns and for raising your score. We are delighted to address your further concerns regarding Weakness 3.
> > >
> > > **1. Why our theory cannot prove GaLore's advantage in entering the second phase of training.**
> > >
> > > Currently, our theoretical analysis only establishes that GaLore may converge to a biased solution, regardless of the number of iterations. However, the theory does not provide insights into the speed at which it converges to this bias. As a result, we cannot theoretically compare the relative speed of GaLore and GoLore during the initial training stage. Importantly, the experimental results are consistent with our theoretical findings, as they do not contradict the established convergence behavior.
> > >
> > > **2. Why convergence guarantees of large-batch GaLore and GoLore are basically the same.**
> > >
> > > In **Lemma 1** we analyzed the error induced by deterministic / large-batch **GaLore's SVD** projection, where $$\\|PP^\top G-G\\|_F^2\le\left(1-\frac{r}{m}\right)\\|G\\|_F^2.$$ In **Lemma 5** we analyzed the error induced by **GoLore's random** projection, where
> > > $$\mathbb{E}[\\|PP^\top G-G\\|_F^2]=\left(1-\frac{r}{m}\right)\\|G\\|_F^2.$$
> > > The convergence guarantees for large-batch GaLore and GoLore are essentially the same because these results, derived under standard assumptions, reflect worst-case convergence rates. While SVD may retain more information by  capturing the principal components of the gradient in the early stage, its worst-case performance happens to align with that of GoLore's where all the $m$ singular values are equal.
> > >
> > > **3. Challenges in aligning theoretical results with the intuition that SVD is generally more beneficial in the early stage.**
> > >
> > > As highlighted above, the upper bound for SVD-induced error matches the expected error of random projection. Consequently, we believe better demonstrating the benefits of SVD in early-stage training requires characterizing additional conditions to rule out worst-case scenarios, where the singular values of the stochastic gradient are nearly identical. In other words, it is challenging to theoretically validate this advanatage by studying convergence results under standard assumptions. Identifying plausible conditions to exclude worst cases could represent an interesting open problem for future research.
> > >
> > > We deeply appreciate your thoughtful feedback and are more than happy to address any further questions or concerns.

---

### Official Review · Reviewer_7hSz · 2024-11-05

**Soundness:** 2
**Presentation:** 3
**Contribution:** 3
**Rating:** 6
**Confidence:** 3

**Summary:**

This paper investigates subspace optimization approaches for LLM pre-training and fine-tuning. The authors demonstrate that GaLore fails to converge to the desired solution under regular assumptions, as the SVD-based projection often generates noise-dominated subspaces when the true gradient is relatively small. Furthermore, the authors introduce a variant of GaLore - GoLore, and establish its convergence rate even with small-batch stochastic gradients. A limitation of this paper is that convergence guarantees for GoLore are currently provided only when using MSGD as the subspace optimizer.

**Strengths:**

1. The paper is logically structured and written in clear, accessible language.
2. The theorems presented are robust and well-supported.

**Weaknesses:**

1. The empirical advantages of GoLore appear marginal. Is the inability of GaLore to converge common in real-world scenarios?
2. The experiments only compare GoLore with GaLore. Do other memory-efficient fine-tuning algorithms also struggle with convergence? If not, what advantages does GoLore offer over these alternatives?
3. It is surprising that selecting components of the gradient matrix randomly provides better convergence guarantees than selecting the most informative components using SVD. A more intuitive explanation of this phenomenon would be beneficial. Additionally, since GoLore does not require SVD decomposition, does it run faster than GaLore?

**Questions:**

Please refer to the Weaknesses.

---

> ### Author Response · Authors · 2024-11-23
> **Response to Reviewer 7hSz**
>
> We thank the reviewer for the detailed comments. All questions have been clarified as best as we can, and we are glad to address any further comments or questions.
>
> **Weakness 1. The marginal advantages of GoLore and whether the inability of GaLore is common in real-world scenarios.**
>
> (1) The marginal advantages of GoLore
>
> For experimental comparisons between GaLore and GoLore, please check our response in Point 3 of the global response. In particular, we have added new expriments to fine-tune OPT-13B on the BoolQ dataset. It shows in Table 4 Appendix E that GoLare achieves an accuracy of 81.96%, notably higher than GaLore's accuracy of 79.79%.
>
> Furthermore, we would like to emphasize that GoLore retains the algorithmic structure of GaLore, differing only in replacing the SVD projection with a random projection during the later phase of pre-training. **This simple modification introduces no additional computational overhead while providing strong convergence guarantees and consistently improved performance**. Given these advantages, we believe random projection is a valuable technique with both theoretical benefits and practical utility.
>
> (2) Non-convergence of GaLore is common in real-world scenarios
>
> In real-world scenarios, GaLore often produces suboptimal results across various tasks. For instance, during LLaMA pre-training on the C4 dataset, GaLore exhibits a consistent performance gap of 0.08–0.82 compared to full-parameter training, as reported in its original paper. Similarly, fine-tuning RoBERTa on GLUE yields comparable outcomes, with GaLore's score being 0.36–0.41 lower than that of full-parameter fine-tuning. However, in our experiments, we observed that by incorporating random projection, GoLore can significantly reduce the performance gap to 0.12, demonstrating its effectiveness to improve the performance.
>
> **Weakness 2. Whether other memory-efficient fine-tuning algorithms also struggle with convergence.**
>
> Thank you for this insightful question! To the best of our knowledge, no existing memory-efficient pre-training or fine-tuning algorithms, including LoRA, DoRA, ReLoRA, SLTrain, Flora, ETHER+, and others, have theoretical convergence guarantees. It is not clear in the literature whether these algorithms encounter convergence issues. **This underscores the contribution of our work, as we establish the first convergence guarantees for a memory-efficient algorithm**.
>
> In our manuscript, we focus solely on GaLore and GoLore because: a) GaLore is a strong baseline algorithm in this field, and b) the only difference between GaLore and GoLore is the use of random projection. Therefore, it is natural to compare GaLore and GoLore in order to validate the effectiveness of the random projection.
>
> **Weakness 3. A more intuitive explanation of why random projection works and whether it is quicker than SVD.**
>
> (1) Intuition behind random projection
>
> Thanks for raising this question. We are happy to clarify it. The stochastic gradient matrix consists of two components: **the true gradient** and **the gradient noise**, as illustrated in Fig. 2. When the true gradient significantly outweighs the gradient noise, typically at the start of training, the low-rank subspace obtained via SVD effectively preserves the true gradient information. This is why the reviewer believes the SVD projection is beneficial. However, as training progresses toward the stationary solution, the true gradient diminishes to zero while the stochastic gradient noise persists. **In this scenario, where the gradient noise dominates the true gradient, SVD can only capture the noise information and may entirely miss the true gradient**. This is the fundamental reason why GaLore with SVD fails to converge to the desired solution.
>
> In contrast to the SVD projection, **our proposed random projection consistently captures the true gradient information in expectation, regardless of whether the gradient noise dominates the true gradient or how small the value of \(r\) is chosen**. This explains why random projection outperforms SVD projection in the later stages of the pre-training phase. For further details, please refer to our response in Point 3 of the global response.
>
> (2) Is random projection quicker than SVD projection?
>
> Using random projections instead of SVD may not result in faster computation, as the subspace projection matrix is computed every $\tau$ iterations, making the overhead of SVD relatively minor on average. We did not observe any speedup in our experiments.
>
> We thank the reviewer again for his careful comments. We hope these responses can clarify the reviewer's questions and are more than happy to address any further comments or questions.

---

> > ### Comment · Reviewer_7hSz · 2024-11-27
> >
> > Thank you for your detailed response. I decide to raise my score, as the authors have addressed my main concerns.

---

> > > ### Author Response · Authors · 2024-11-29
> > >
> > > We sincerely thank the reviewer for the thoughtful feedback and for considering our responses. We greatly appreciate the time and effort dedicated to reviewing our work and are glad that our clarifications addressed your main concerns. Thank you for your support and for raising your score.

---

### Author Response · Authors · 2024-11-23
**Global Response (Part 1/2)**

We sincerely thank all the reviewers for their valuable comments and suggestions. We are pleased that most reviewers recognize our theoretical results as **solid** and **well-motivated**. However, we have noted several common misunderstandings, particularly concerning the main intuition and contributions of this work. To address these, we provide clarifications in this global response.

**1. Intuition behind random projection.**

The intuition behind GaLore's failure to converge to the stationary solution is that the SVD projection represents a **greedy** strategy. As the algorithm approaches the stationary solution, i.e., first-order stationary points and gradient noise begins to dominate the true gradient, the SVD-derived subspace may exclusively capture the noise component (particularly when selecting a sufficiently small $r$) and **completely miss the true gradient information**. Consequently, when no meaningful gradient information can be utilized, GaLore naturally fails to converge. This is the primary intuition underlying our counter-example presented in Equation (1).

In contrast, **our proposed random projection consistently captures the true gradient information in expectation, regardless of whether the gradient noise dominates the true gradient or how small $r$ is chosen**. To see it, let $G$ denote the true gradient and $\epsilon$ represent the zero-mean random gradient noise. Additionally, we assume the random projection $P$ is independent of the gradient noise. By applying Lemma 5 in our manuscript, the gradient estimate satisfies

$$
\mathbb{E}[PP^T (G + \epsilon)] = \mathbb{E}[PP^T] G + \mathbb{E}[PP^T] \mathbb{E}[\epsilon] = \mathbb{E}[PP^T] G = \frac{r}{m} G.
$$

In summary, random projection consistently preserves gradient information, whereas SVD projection may completely lose it during the training process. This distinction highlights why GaLore fails to reach a stationary solution, while GoLore succeeds.

**2. Intuition behind the hybrid method of GaLore and GoLore.**

As illustrated in Section 3, LLM pre-training can be divided into two stages: (a) an initial stage where gradients are less noise-dominated, and (b) a final stage where gradients are highly noise-dominated. In the initial stage, the SVD strategy proves more effective than the random strategy, as it captures the principal components of the gradient. Consequently, employing GaLore during this phase is advantageous for quickly approaching a near-optimal solution. However, as our theoretical results suggest , the random strategy becomes more effective in the final stage for achieving a more accurate optimal solution. Therefore, we propose combining GaLore and GoLore to leverage the strengths of each: rapid progress in the initial stage and precise convergence in the final stage.

**3. Experimental comparison between GaLore and GoLore.**

a) Firstly, while we respect the reviewers' concerns regarding the "marginal" improvements observed in the experiments, we believe these results effectively validate our theoretical contributions. Our primary theoretical finding demonstrates that GaLore fails to converge to a first-order stationary solution, whereas GoLore successfully achieves this, as clearly illustrated in Figures 1 and 4. In these figures, the loss curve for GoLore closely matches that of full-parameter fine-tuning, whereas the loss curve for GaLore is noticeably worse. Moreover, the results in Table 2 indicate that GoLore@20% consistently outperforms GaLore across all 8 tasks in the GLUE benchmark, achieving a score only **0.12 lower** than full-parameter training. In contrast, GaLore's average score is **0.38 lower**, which is **more than three times** worse. This further validates the benefits of using random projection in approaching stationary solutions.

b) Secondly, we have included new experiments demonstrating significant improvements of GoLore over GaLore. For instance, when fine-tuning OPT-13B on the BoolQ dataset, GoLore achieves an accuracy of 81.96%, notably higher than GaLore's accuracy of 79.79%. These results are detailed in Table 4 of Appendix E in our revised manuscript.

c) Thirdly, we would like to emphasize that GoLore retains the algorithmic structure of GaLore, differing only in replacing the SVD projection with a random projection during the later phase of pre-training. **This simple modification introduces no additional computational overhead while providing strong convergence guarantees and consistently improved performance.** Given these advantages, we believe random projection is a valuable technique with both theoretical benefits and practical utility.

---

> ### Author Response · Authors · 2024-11-23
> **Global Response (Part 2/2)**
>
> **4. New experiments.**
>
> In our revised manuscripts, we have added more experiments to respond to reviewer's concerns, including:
>
> a) Figure 7 in Appendix E: pre-training tasks with more tokens;
>
> b) Table 2 in Section 7: fine-tuning RoBERTa-base on GLUE with Flora.
>
> c) Table 4 in Appendix E: fine-tuning OPT-13B on BoolQ with GaLore and GoLore@30%.
>
> We sincerely thank all reviewers for their valuable comments and are happy to address any further questions or concerns.

---

### Meta-Review · Area_Chair_ygaL · 2024-12-22

**Metareview:**

The paper makes two primary contributions. First, the authors study GaLore (a memory-efficient algorithm for pre-training LLMs by accumulating a sequence of low-rank updates), and show that this algorithm can diverge as training progresses via a carefully constructed counter-example. Second, the authors propose a modified algorithm (GoLore) which is a variant of GaLore, and which converges. The key technical ingredient is to change the way GaLore compresses gradients -- instead of SVD, use random projections. Not too surprisingly, this helps theory-wise, but in practice the authors advocate using GaLore most of the time, and switch to GaLore only towards the end of training.

The paper is generally well-written and the idea is sensible.

My main concerns are related to what I perceive as incremental novelty and significance, which prevent me from recommending acceptance. First, from the theory side, the top-line message of this paper is not quite precise. As the authors themselves argue, GaLore converges fine in large-batch settings and under isotropic noise. Second, from the practical side, the algorithm that the authors test in their experiment is not GoLore but GoLore@20%, which is really GaLore 80% of the time, followed by GoLore towards the end to gain some marginal improvements in GLUE scores. I suppose this is sensible but diminishes the impact of the newly proposed algorithm. Third (and most importantly, in my opinion), the idea of using random projections instead of SVD is not new; FLORA does something very similar (the only difference seems to be that FLORA uses iid Gaussian random sketches, while GoLore uses random orthogonal projections).

**Additional Comments On Reviewer Discussion:**

Initial review scores were tending towards a reject. The paper had several obvious typos, and reviewers mentioned the very limited improvement in GLUE performance, and the lack of comparisons with other low-rank methods. The authors provided an extensive response with additional theory and experiments. However, my concerns about novelty and significance unfortunately still persist.

---

### Decision · Program_Chairs · 2025-01-22

Reject